# TXNIP mediates LAT1/SLC7A5 endocytosis to limit amino acid uptake in cells entering quiescence

Jennifer Kahlhofer [1,2,21], Nikolas Marchet[1,2,21], Kristian Zubak[1], Brigitta Seifert [2], Madlen Hotze [3], Anna-Sophia Egger-Hörschinger [3], Lucija Kucej [1], Claudia Manzl[4], Yannick Weyer [1,2], Sabine Weys[2,5], Martin Offterdinger [6], Sebastian Herzog [7], Veronika Reiterer[8], Chiara Volani [9], Marcel Kwiatkowski [3], Saskia B Wortmann [10,11], Siamak Nemati[12], Johannes A Mayr [10], Johannes Zschocke [13], Bernhard Radlinger[14], Kathrin Thedieck[3,15,16,17,18], Leopold Kremser[19], Bettina Sarg[19], Lukas A Huber [2], Hesso Farhan [8], Mariana E G de Araujo [2], Susanne Kaser [14], Sabine Scholl-Bürgi[20], Daniela Karall [20] & David Teis [1,2✉]

## Abstract

Entry into and exit from cellular quiescence require dynamic adjustments in nutrient acquisition, yet the mechanisms by which quiescent cells downregulate amino acid (AA) transport remain poorly understood. Here we show that cells entering quiescence selectively target plasma membrane-resident amino acid transporters for endocytosis and lysosomal degradation. This process matches amino acid uptake with reduced translational demand and promotes survival during extended periods of quiescence. Mechanistically, we identify the α-arrestin TXNIP as a key regulator of this metabolic adaptation, since it mediates the endocytosis of the SLC7A5-SLC3A2 (LAT1-4F2hc) AA transporter complex in response to reduced AKT signaling. To promote transporter ubiquitination, TXNIP interacts with NEDD4L and other HECT-type ubiquitin ligases. Loss of TXNIP disrupts this regulation, resulting in dysregulated amino acid uptake, sustained mTORC1 signaling, and ultimately cell death under prolonged quiescence. The characterization of a novel TXNIP loss-of-function variant in a patient with a severe metabolic disease further supports its role in nutrient homeostasis and human health. Together, these findings highlight TXNIP's central role in controlling nutrient acquisition and metabolic plasticity with implications for quiescence biology and diseases.

**Keywords** Amino Acids Uptake; Quiescence; TXNIP; SLC7A5/LAT1; Endocytosis
**Subject Categories** Genetics, Gene Therapy & Genetic Disease; Membranes & Trafficking; Metabolism

## Introduction

How cells maintain a homeostatic pool of 20 proteinogenic amino acids (AA) is a fundamental question in biology. For the uptake and the release of AA across the plasma membrane (PM) and across organelle membranes, the human genome encodes at least 66 AA transporters, which belong to 11 individual solute carrier (SLC) families (Chidley et al, 2024; Kandasamy et al, 2018). The importance of controlling AA transport is exemplified by monogenic diseases caused by pathogenic variants in individual AA transporter and human pathologies associated with the dysregulation of AA transporters (Errasti-Murugarren and Palacin, 2022; White et al, 2021).

[1]Institute of Molecular Biochemistry, Biocenter, Medical University of Innsbruck, Innsbruck 6020, Austria. [2]Institute of Cell Biology, Biocenter, Medical University of Innsbruck, Innsbruck 6020, Austria. [3]Department of Biochemistry and Center for Molecular Biosciences Innsbruck, University of Innsbruck, Innsbruck 6020, Austria. [4]Institute of Neuropathology & Neuromolecular Pathology, Medical University of Innsbruck, 6020 Innsbruck, Austria. [5]Institute of Science and Technology Austria (ISTA), Klosterneuburg 3400, Austria. [6]Institute of Systems Neuroscience, BioOptics core facility, Biocenter, Medical University of Innsbruck, Innsbruck 6020, Austria. [7]Institute for Developmental Immunology, Biocenter, Medical University of Innsbruck, Innsbruck 6020, Austria. [8]Institute of Pathophysiology, Biocenter, Medical University of Innsbruck, Innsbruck 6020, Austria. [9]Department of Internal Medicine II, Medical University of Innsbruck, Innsbruck 6020, Austria. [10]University Children's Hospital, Salzburger Landeskliniken and Paracelsus Medical University, Salzburg 5020, Austria. [11]Amalia Children's Hospital, Department of Pediatrics, Pediatric Neurology, Nijmegen, The Netherlands. [12]Kinder- und Jugendheilkunde, Krankenhaus St. Vinzenz Zams, Zams 6511, Austria. [13]Institute of Human Genetics, Medical University of Innsbruck, Innsbruck 6020, Austria. [14]Department of Internal Medicine I, Medical University of Innsbruck, Innsbruck 6020, Austria. [15]Department Metabolism, Senescence and Autophagy, Research Center One Health Ruhr, University Alliance Ruhr & University Hospital Essen, University Duisburg-Essen, Essen, Germany. [16]German Cancer Consortium (DKTK), partner site Essen, a partnership between German Cancer Research Center (DKFZ) and University Hospital Essen, Heidelberg, Germany. [17]Freiburg Materials Research Center (FMF), University Freiburg, Freiburg, Germany. [18]Laboratory of Pediatrics, Section Systems Medicine of Metabolism and Signaling, University of Groningen, University Medical Center Groningen, Groningen, The Netherlands. [19]Institute of Medical Biochemistry, Protein Core Facility, Medical University of Innsbruck, Innsbruck, Austria. [20]Department of Child and Adolescent Health, Division of Pediatrics I-Inherited Metabolic Disorders, Medical University of Innsbruck, Innsbruck 6020, Austria. [21]These authors contributed equally: Jennifer Kahlhofer, Nikolas Marchet.✉E-mail: david.teis@i-med.ac.at

Cellular AA transport is coordinated with cell proliferation. Proliferating cells, in particular cancer cells, increase the abundance of AA transporters at the cell surface and thereby promote nutrient uptake for protein synthesis and other anabolic process that promote cell growth (Palm and Thompson, 2017). The upregulation of AA transporter levels is mainly mediated by transcriptional mechanisms (Hansen et al, 2015; Hayashi et al, 2012; Kandasamy et al, 2021; Yue et al, 2017).

Conversely, entry into quiescence, the reversible exit from the cell cycle, is accompanied by a reduction in cell size, lower rates of transcription, and reduced protein synthesis (Coller, 2011; Marescal and Cheeseman, 2020). Thus, quiescent cells must recalibrate AA uptake to align it with lower translation and metabolic homeostasis for cell survival, rather than for growth (Rathmell et al, 2000). How quiescent cells match AA flux across the PM to reduced cell size and to lower protein synthesis is unclear. This presents a knowledge gap in understanding cellular nutrient acquisition.

In budding yeast, *Saccharomyces cerevisiae*, we previously demonstrated that entry into quiescence induces the ubiquitin dependent endocytosis of AA transporters (Muller et al, 2015). The selective ubiquitination of AA transporters requires proteins of the α-arrestin family (Alvarez, 2008; Patwari and Lee, 2012). Activated α-arrestins function as adaptor proteins that recruit the HECT (homologous to the E6-AP carboxyl terminus) type ubiquitin ligase Rsp5 to different AA transporters for ubiquitination, endocytosis and lysosomal degradation (Ivashov et al, 2020; Lin et al, 2008; Nikko et al, 2008).

The human genome encodes six α-arrestin family members or ARRDCs (arrestin-domain-containing proteins). With the exception of ARRDC5, α-arrestins contain one or more PPxY motifs in their C-terminal regions and are able to interact with several HECT-type ubiquitin ligases (Lee et al, 2024; Liu et al, 2016; Nabhan et al, 2010; Rauch and Martin-Serrano, 2011; Zhang et al, 2010). The best characterized α-arrestin is TXNIP (thioredoxin interacting protein). TXNIP was first described as a negative regulator of thioredoxin (TXN) through the formation of an intermolecular disulfide bond that requires two cysteine residues of TXNIP (cysteine 32 and cysteine 247) (Hwang et al, 2014; Patwari et al, 2006; Polekhina et al, 2013).

TXNIP also regulates glucose uptake by mediating the selective endocytosis of SLC2A1 (GLUT1) and SLC2A4 (GLUT4), two major glucose transporters. TXNIP binds to these glucose transporters at the PM and interacts with a di-leucine motif in its C-terminal tail and with components of the endocytic machinery (Waldhart et al, 2017; Wu et al, 2013). The TXNIP dependent regulation of SLC2A1 and SLC2A4 is tightly coupled to intracellular glucose homeostasis, ATP availability and growth factor signaling. AMPK activation in response to low ATP levels or AKT activation in response to insulin signal results in the phosphorylation of TXNIP at serine 308, thereby inactivating it (Waldhart et al, 2017; Wu et al, 2013). This inactivation of TXNIP ensures SLC2A1 and SLC2A4 accumulation at the PM to increase glucose uptake (Waldhart et al, 2017; Wu et al, 2013). TXNIP inactivation can also involve ubiquitination by the HECT-type ubiquitin ligase ITCH, leading to its proteasomal degradation (Liu et al, 2016; Zhang et al, 2010). These processes appear to be largely independent of TXNIP's ability to interact with TXN. Consistent with the function in regulating glucose uptake, a biallelic loss-of-function variant in *TXNIP* leads to lactic acidosis in three siblings. These patients also showed lower serum methionine levels (Katsu-Jimenez et al, 2019).

Here, we show that in human cells entering quiescence, several AA transporters are selectively removed from the PM by endocytosis, including the heterodimeric AA transporter (HAT) SLC7A5-SLC3A2 (LAT1-4F2hc). The endocytic degradation of SLC7A5-SLC3A2 during entry into quiescence requires TXNIP. TXNIP uses its PPxY motif at position 331 to engage HECT-type ubiquitin ligases (foremost NEDD4L, but also others) for SLC7A5-SLC3A2 endocytosis and subsequent lysosomal degradation. The TXNIP-mediated endocytic degradation of SLC7A5-SLC3A2 reduces AA uptake, lowers free intracellular AA levels, and thereby helps to dampen mTORC1 signaling and translation. This process is important to allow cell survival over longer periods of quiescence. In growing cells, AKT phosphorylation of TXNIP restricts SLC7A5-SLC3A2 endocytosis. In addition, we identified a novel biallelic loss-of-function TXNIP variant in a boy with a severe inborn metabolic disease. The analysis of fibroblasts from this patient revealed that SLC7A5-SLC3A2 endocytosis was also blocked. This further supports the central role of TXNIP in adjusting cellular AA uptake, with important implications for human health.

# Results

## Selective endocytosis and lysosomal degradation of AA transporters in cells entering quiescence

To study the regulation of nutrient uptake during the transition from proliferation to quiescence, we used hTERT RPE1 cells (non-cancerous human telomerase-immortalized retinal pigmented epithelial, hereafter RPE1) as a model cell line (Barr et al, 2017; Hinze et al, 2021). First, we established and evaluated conditions for entry into quiescence.

The DNA content of cells growing with serum was analyzed by propidium iodide (PI) staining and fluorescence activated cell sorting (FACS). 46% (± 2%) of these cells were in the G0/G1 phase of the cell cycle, 30% (± 6%) were in S phase and 24% (± 4%) in the G2/M phase. 24 h after serum starvation (but in the presence of AA and glucose), the number of cells in G0/G1 increased to 67% (± 10%), while the number of cells in S-phase and G2/M phase decreased to 15% (± 6%) and 19% (± 7%), respectively (Fig. 1A). Cell viability was not affected by serum starvation during a period of 24 h (Appendix Fig. 1SA). Serum starved cells also decreased their volume by 20% from 2907 fl to 2332 fl (Fig. 1B). Consistent with an entry into quiescence in response to serum starvation, SDS-PAGE and Western Blot (WB) analysis of total cell lysates showed that the protein levels of the cyclin-dependent kinase inhibitor p27[Kip1] increased, while the phosphorylation dependent activation of mitogen activated kinases ERK1/2, and mTORC2-dependent phosphorylation of AKT on serine 473 decreased (Fig. 1C, lane 2).

Of note, in cells entering quiescence, the protein levels of the heterodimeric AA transporters (HAT) SLC7A5-SLC3A2 (LAT1-4F2hc) (Fig. 1C, lane 2; Appendix Fig. S1B) and SLC7A11-SLC3A2 (xCT-4F2hc) decreased (Fig. 1C). The protein levels of SLC1A5 (ASCT2) also decreased (Fig. 1C). The protein levels of the neutral AA transporter SLC38A2 (SNAT2), the glucose transporter SLC2A1 (GLUT1), the transferrin receptor TfR and epidermal growth factor receptor (EGFR) remained relatively unchanged over the course of the experiment (Fig. 1C). The downregulation of SLC7A5 and SLC1A5

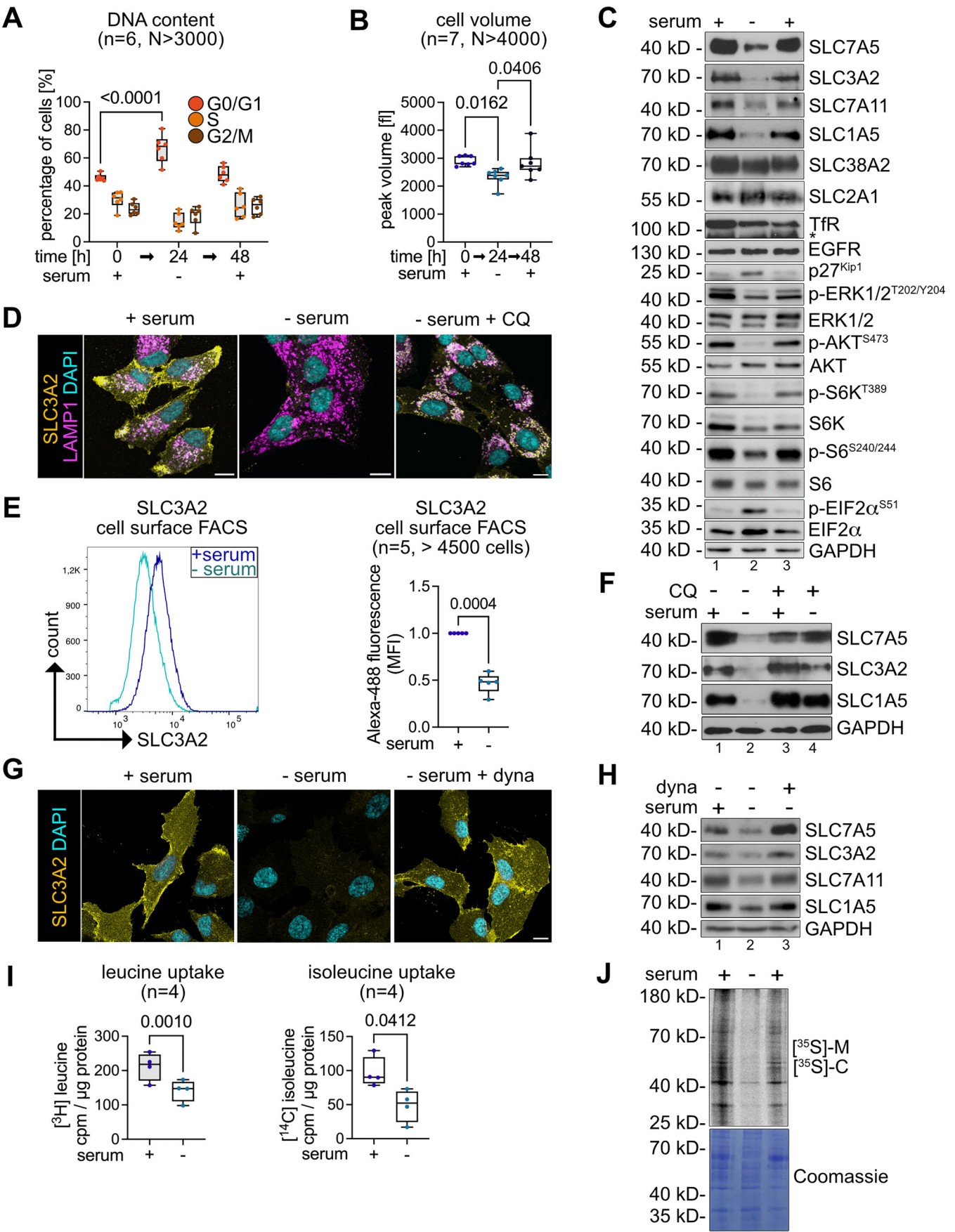

**Figure 1. Dynamin-dependent endocytosis and lysosomal degradation of SLC7A5 and SLC1A5.**

RPE1 cells were grown in DMEM supplemented with serum (0 h, + serum), cultured in serum free medium for 24 h (24 h, - serum) and refed with serum for 24 h (48 h, + serum). (A) Cells were harvested, permeabilized and fixed, and the DNA content was analyzed by propidium iodide (PI) staining to assess the cell cycle profile by FACS. Box plots represent the median (centre line) and the interquartile range (25th to 75th percentile box). The whiskers show the minimum and maximum values. ($n = 6$, $N > 3000$ cells, two-way ANOVA, Tukey's multiple comparisons test, G0/G1 t0[h] vs G0/G1 t24[h] $P = 6.7E-5$). (B) Peak volume (in fl) was assessed by CASY Cell Counter and Analyzer. Box plots represent the median (centre line) and the interquartile range (25th to 75th percentile box). The whiskers show the minimum and maximum values. ($n = 7$, $N > 4000$ cells, one-way ANOVA, Tukey's multiple comparisons test). (C) Total cells lysates were analyzed by SDS-PAGE and Western Blot (WB) with the indicated antibodies. (D) Indirect Immunofluorescence (IF) of PFA fixed cells stained for SLC3A2 (yellow), LAMP1 (magenta) and DAPI (cyan) were analyzed by confocal microscopy. The merged images show a single plane of a Z-stack. Scale bar = 10 µm. Incubation with 12.5 µM chloroquine (CQ) in absence of serum for 14 h (- serum, + CQ). (E) The cell surface was stained with anti-SLC3A2 and anti-mouse Alexa-488. FACS was used to detect SLC3A2 surface staining. A representative histogram is shown. Quantification of the fluorescence intensities of SLC3A2 was normalized to proliferating cells (+ serum). Box plots represent the median (centre line) and the interquartile range (25th to 75th percentile box). The whiskers show the minimum and maximum values. ($n = 5$, $N > 4500$ cells, paired $t$ test). (F) Total cells lysates of cells before and after serum starvation for 14 h and treatment with 12.5 µM chloroquine or vehicle (DMSO) analyzed by SDS-PAGE and WB with the indicated antibodies. (G) Indirect Immunofluorescence (IF) of PFA fixed cells stained for SLC3A2 (yellow), LAMP1 (magenta) and DAPI (cyan) were analyzed by confocal microscopy. The merged images show a single plane of a Z-stack. Scale bar = 10 µm. Incubation with 20 µM dynasore (dyna) in absence of serum for 14 h (− serum, + dyna). (H) Total cells lysates of cells before and after serum starvation for 14 h and treatment with 20 µM dynasore or vehicle (DMSO) analyzed by SDS-PAGE and WB with the indicated antibodies. (I) Cells were incubated with [3H]-leucine or [14 C]-isoleucine for 15 min, washed and lysed. Total cell lysates were analyzed by scintillation counting. Counts per minute (cpm) values were normalized to total protein content. Box plots represent the median (centre line) and the interquartile range (25th to 75th percentile box). The whiskers show the minimum and maximum values. (cpm/µg protein, $n = 4$, paired $t$ test). (J) Cells were incubated with [35S]-methionine, [35S]- cysteine for 2 min, before cycloheximide (CHX, 10 µg/ml) was added, cells were lysed and analyzed by SDS-PAGE and autoradiography. Source data are available online for this figure.

was observed in other non-cancerous cell lines, such as mouse embryonic fibroblasts (MEFs) (Appendix Fig. S1C) and primary human fibroblasts (Appendix Fig. S1D,E), when subjected to serum starvation. Nutrient transporter downregulation was not observed in a panel of seven different lung cancer cell lines (Appendix Fig. S1F).

Entry into quiescence in RPE1 cells was also accompanied by induction of autophagy and microautophagy (Mejlvang et al, 2018) (Fig. EV1A). The degradation of SLC7A5 protein levels occurred later than the turnover of LC3-I/II and the decrease of the protein levels of different autophagy adaptors, including NBR1, TAX1BP1, and NDP52 (Mejlvang et al, 2018) (Fig. EV1A).

To characterize how SLC7A5-SLC3A2 and SLC1A5 were downregulated in RPE1 cells entering quiescence, we examined the localization of these transporters using indirect immunofluorescence (IF) and confocal microscopy. In growing cells, endogenous SLC3A2 (Fig. 1D) and SLC1A5 (Fig. EV1B) were mainly detected at the PM and partially co-localized with LAMP1 positive lysosomes (Figs. 1D and EV1B). In serum starved cells, signals for SLC3A2 were barely detected (Figs. 1D and EV1C), which was in line with the downregulation of SLC7A5-SLC3A2 in quiescent cells. Also, cell surface FACS analysis measured a marked decrease of PM resident SLC3A2 upon serum starvation (Fig. 1E). Treatment of cells with chloroquine (CQ), which impairs lysosomal catabolic function, resulted in the accumulation of SLC3A2 (Fig. 1D) and SLC1A5 (Fig. EV1B) inside LAMP1 positive lysosomes and efficiently prevented the lysosomal degradation of these AA transporters (Figs. 1D, F and EV1D). Inhibition of the GTPase dynamin with dynasore (dyna), which blocked the scission of clathrin coated vesicles from the PM (Kirchhausen et al, 2008; Macia et al, 2006), impaired the endocytosis of SLC7A5-SLC3A2 (Fig. 1G, F) and SLC1A5 (Figs.1F and EV1E) and in consequence prevented their lysosomal degradation (Figs. 1F and EV1E–G).

In line with the quiescence induced endocytic degradation of SLC7A5-SLC3A2, the uptake of its AA substrates [3H]-leucine and [14C]-isoleucine was reduced by 33% and 49%, respectively (Fig. 1I). Similarly, the uptake of the SLC1A5 substrate [14C]-glutamine decreased by 20% (Fig. EV1H).

The decrease in AA uptake upon entry into quiescence correlated with a reduction of mTORC1 signaling towards S6K and with an increase of eIF2α phosphorylation at serine 51 (Fig. 1C), indicating a downregulation of translation. Consistently, the incorporation of [35S]-methionine and [35S]-cysteine into newly synthesized proteins was reduced in quiescent cells (Fig. 1J, lane 2).

Upon re-addition of serum to starved cells, SLC7A5-SLC3A2, SLC7A11-SLC3A2, and SLC1A5 protein levels were upregulated, ERK-, AKT- as well as mTORC1-signaling were re-activated, phosphorylation of eIF2α decreased as did p27^Kip1 protein levels (Fig. 1C, lane 3). Translation (Fig. 1J, lane 3) and cell size increased (Fig. 1B), and cells resumed proliferation (Fig. 1A).

These results showed that the selective endocytosis and degradation of SLC7A5-SLC3A2 coincided with a decrease in cell volume, with a drop in translation, and with a reduction in leucine and isoleucine uptake as cells entered quiescence in response to serum starvation.

## TXNIP is required for the selective downregulation of SLC7A5

To identify the molecular mechanism responsible for the quiescence induced endocytic degradation of SLC7A5-SLC3A2, we focused on the α-arrestin family of ubiquitin ligase adaptors. α-arrestins are required for the endocytic downregulation of AA transporters in budding yeast cells entering quiescence (Ivashov et al, 2020; Muller et al, 2015). It was not known if they had a similar function in human cells.

The human genome encodes six α-arrestins, called ARRDC1-5 (arrestin domain containing), and TXNIP (Alvarez, 2008). TXNIP protein levels (Fig. 2A,B) and mRNA levels (Fig. EV2A) increased in response to serum starvation.

To determine whether TXNIP contributed to SLC7A5-SLC3A2 endocytosis and lysosomal degradation, we used CRISPR/Cas9-mediated gene editing and introduced a 13 base pair deletion (c.644_656del), that resulted in a frameshift and a premature stop codon at amino acid 226 (p.Ile215Thrfs*11) (Fig. EV2B,C). This

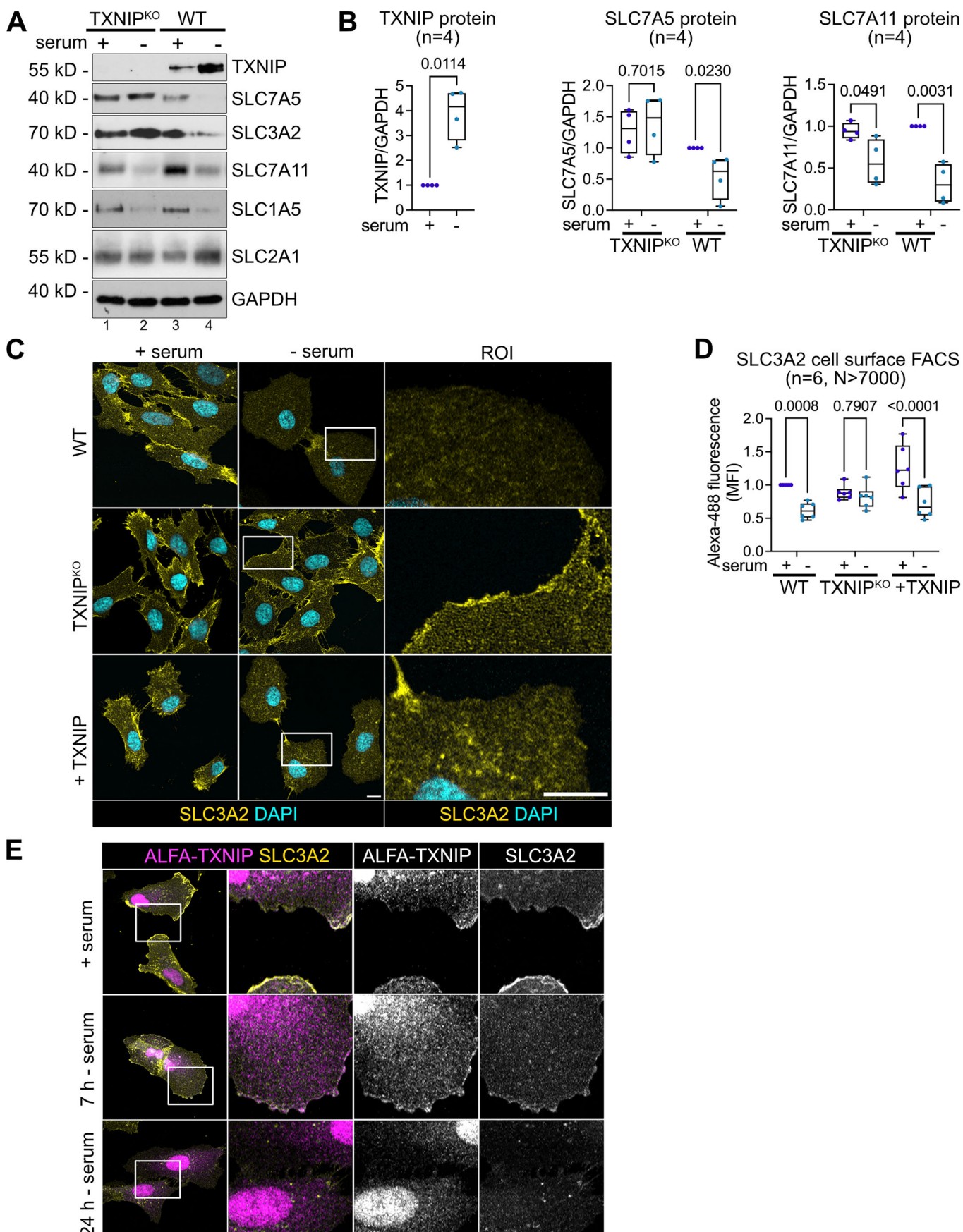

◄ **Figure 2. The importance of TXNIP for the selective degradation of SLC7A5.**

(A) TXNIP$^{KO}$ and WT cells were grown in growth medium ( + serum) or serum starved for 24 h (-serum). Total cell lysates were analyzed by SDS-PAGE and WB with the indicated antibodies. (B) WB quantification of TXNIP ($n = 4$, paired $t$ test), SLC7A5 ($n = 4$, two-way ANOVA, Sidak's multiple comparisons test) and SLC7A11 protein levels. Box plots represent the median (centre line) and the interquartile range (25th to 75th percentile box). The whiskers show the minimum and maximum values. ($n = 4$, two-way ANOVA, Sidak's multiple comparisons test). Values were normalized to GAPDH levels of proliferating WT cells ( + serum). (C) IF of PFA fixed cells (WT, TXNIP$^{KO}$ and TXNIP$^{KO}$ reconstituted with TXNIP) stained for SLC3A2 (yellow) and DAPI (cyan) were analyzed by confocal microscopy. The images show a single plane of a Z-stack. Regions of interest (ROI, white box) were magnified. Scale bar $= 10$ μm. (D) Quantification of SLC3A2 cell surface FACS of the indicated cells, normalized to proliferating WT cells. Box plots represent the median (centre line) and the interquartile range (25th to 75th percentile box). The whiskers show the minimum and maximum values. ($n = 6$, $N > 7000$, two-way ANOVA, Sidak's multiple comparisons test, +TXNIP +serum vs +TXNIP −serum: $P = 2.3E$-5). (E) IF of PFA fixed TXNIP$^{KO}$ reconstituted with ALFA-TXNIP. The localization of ALFA-TXNIP (magenta) and SLC3A2 (yellow) was analyzed by confocal microscopy. The images show a single plane of a Z-stack. Regions of interest (ROI, white box) were magnified. Scale bar $= 10$ μm. Source data are available online for this figure.

mutation disrupted the arrestin C-domain (Fig. EV2B). In single cell clones, the mutant protein was no longer detected (Fig. 2A). We refer to cells carrying this mutation as TXNIP knockout (TXNIP$^{KO}$) cells.

The loss-of-function mutation in TXNIP selectively prevented the endocytic downregulation of the SLC7A5-SLC3A2 heterodimer (Fig. 2A, lane 2, and Fig. 2B) but it did not inhibit the degradation of SLC7A11 and SLC1A5 (Fig. 2A) nor did it block the degradation of autophagy adaptors or LC3-II (Appendix Fig. S2A). In these TXNIP$^{KO}$ cells, SLC3A2 remained at the PM in response to serum starvation, as determined by IF and cell surface FACS (Figs. 2C,D and EV2D). Re-expression of TXNIP in TXNIP$^{KO}$ cells restored SLC7A5-SLC3A2 endocytosis and degradation in response to serum starvation (Figs. 2C,D and EV2D–F). Loss of TXNIP did not affect SLC7A5 mRNA levels (Fig. EV2G). Hence, in cells entering quiescence, TXNIP selectively targeted SLC7A5-SLC3A2 for endocytosis and lysosomal degradation.

To examine the localization of TXNIP, ALFA-tagged TXNIP (ALFA-TXNIP) was re-expressed in TXNIP$^{KO}$ cells. Using confocal microscopy, we detected a fraction of ALFA-tagged TXNIP at the PM (Fig. 2E), where it partially co-localized with SLC3A2 (Wu et al, 2013) in proliferating cells, and 7 h after serum starvation (Fig. 2E). 24 h after serum starvation, the majority of SLC3A2 was endocytosed and also ALFA-TXNIP was no longer detected at the PM (Fig. 2E). Under all conditions, ALFA-TXNIP also localized prominently to the nucleus and to the cytosol, as previously observed (Nishinaka et al, 2004; Saxena et al, 2010) (Fig. 2E).

TXNIP has been shown to be required for controlling cellular glucose uptake, by regulating clathrin-mediated endocytosis of the glucose transporters SLC2A1 and SLC2A4 in response to glucose depletion or insulin signaling (Kim et al, 2024; Qualls-Histed et al, 2023; Waldhart et al, 2017; Wu et al, 2013). In WT and in TXNIP$^{KO}$ RPE1 cells entering quiescence, SLC2A1 remained at the PM (Fig. EV2J) and the protein levels of SLC2A1 rather increased (Fig. EV2H,I). Consistently, the uptake of the fluorescent glucose analog 2-NBDG (2-[N-(7-nitrobenz-2-oxa-1,3-diazol-4-yl)amino]-2-deoxy-D-glucose) also increased in cell entering quiescence (Fig. EV2K) (Hytti et al, 2023). In the TXNIP$^{KO}$ cells, 2-NBDG uptake was moderately elevated already in growing cells (Fig. EV2K).

Our data suggested that TXNIP was required for the selective endocytic degradation of the SLC7A5-SLC3A2 heterodimer in cells entering quiescence. Since entry into quiescence did not trigger SLC2A1 endocytosis, it appeared that the TXNIP dependent downregulation of SLC7A5-SLC3A2 can be uncoupled from its role in SLC2A1 endocytosis.

## Identification of a novel *TXNIP* loss-of-function variant in a patient with an inborn metabolic disorder

TXNIP deficiency caused by a loss-of-function variant c.174_175delinsTT (resulting in a frameshift and premature stop codon p.Gln58Hisfs*2) has been previously reported in three siblings with congenital lactic acidosis, low serum methionine levels, variable hypoglycaemia and other metabolic alterations (Katsu-Jimenez et al, 2019). During the course of the study, we identified a novel TXNIP loss-of-function variant in another patient.

The patient is a boy, born in 2014 as the first child of healthy, consanguineous parents of Turkish origin. More details on his medical history are provided in the appendix (Appendix Patient Information). On the seventh day of life, he exhibited floppiness, recurrent hypoglycaemia, and lactic acidosis. A glucose infusion (10 mg/kg/min) stabilized his glucose and lactate concentrations. By day 20, glucose and lactate levels had stabilized with regular feeding but his muscular hypotonia persisted. During infancy, his blood glucose concentrations were within standard range (Appendix Table S1), but the boy experienced recurrent hypoglycaemia in response to metabolic stress, e.g., infections. He exhibited psychomotor developmental delays and, from 18 months of age, experienced increasing epileptic seizures (up to 3–4 per month). Currently, he remains metabolically stable but presents with significant developmental delay, muscular hypotonia, and autistic features (Appendix Fig. S2B).

Exome sequencing from peripheral blood of the patient detected a homozygous single nucleotide insertion c.642_643insT in exon 5 of 8 of the *TXNIP* gene. Both parents are healthy heterozygous carriers for this variant. This *TXNIP* variant was not recorded in the population genetic variant database gnomAD that lists *TXNIP* as likely haplosufficient (pLI = 0, LOEUF = 0,709: https://gnomad.broadinstitute.org accessed June 29, 2025). No other (likely) pathogenic variant in any other gene, with known function in metabolism was identified as explanation of the clinical features in the child. Potential pathogenic variants in genes required for mitochondrial functions were also not detected, although they were initially expected to cause the phenotype.

The *TXNIP* variant c.642_643insT caused a frameshift and a premature stop codon after 59 AA (p.Ile215Tyrfs*59), likely causing nonsense-mediated decay (NMD) or the synthesis of a severely truncated TXNIP protein (Fig. 3A). Serendipitously, this TXNIP variant was similar to the gene-edited version in the RPE1 TXNIP$^{KO}$ cells (p.Ile215Thrfs*11).

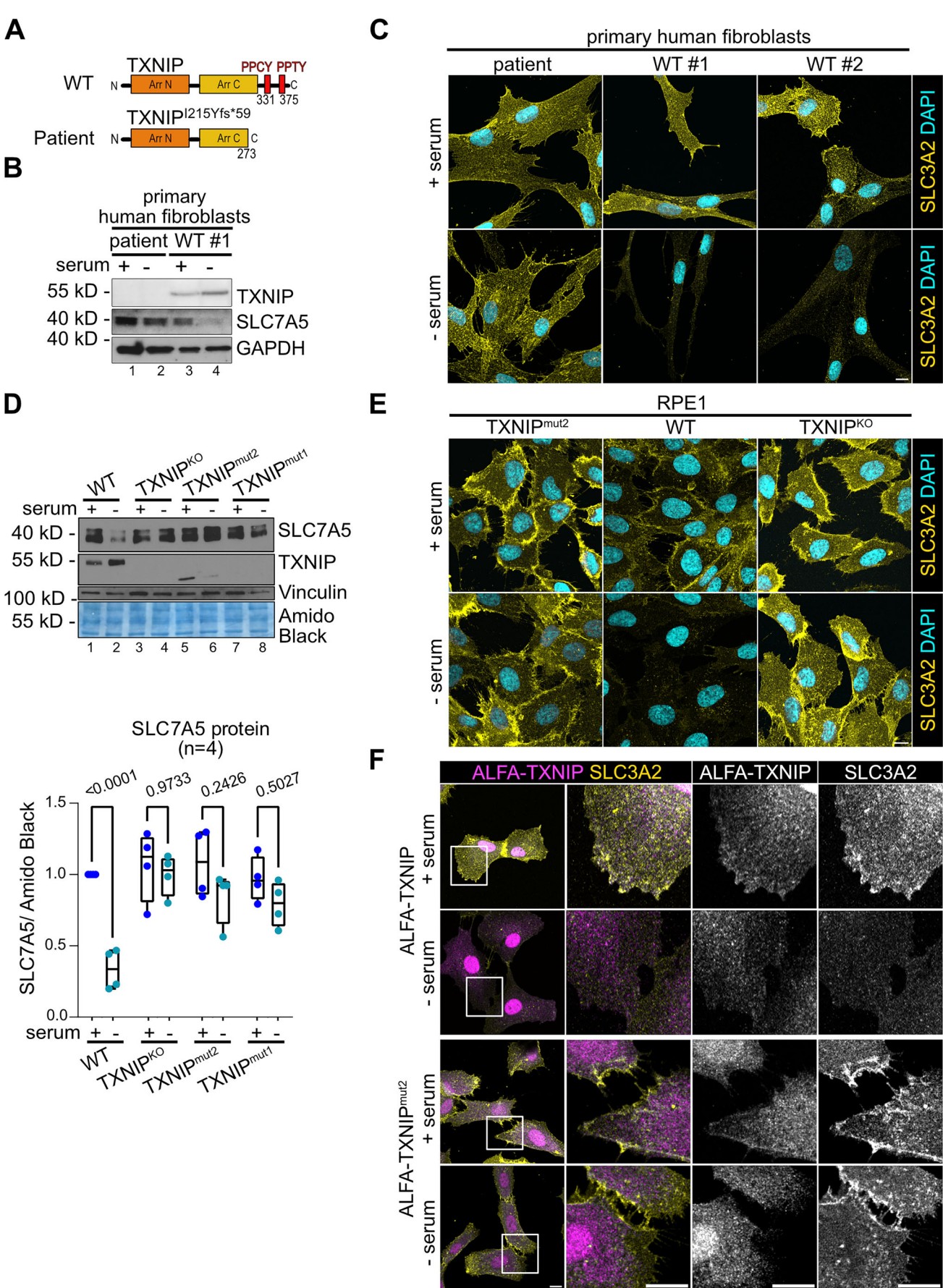

**Figure 3. Loss of function mutation in TXNIP causes a rare disease.**

(A) Schematic representation of WT TXNIP and mutant TXNIP harboring the patient mutation (c.642_643insT, patientmut2). (B, C) Control fibroblasts (WT#1 and WT#2) and patient-derived fibroblasts (patient) were grown in growth medium ( + serum) or serum starved for 24 h (-serum). (B) Total cell lysates were analyzed by SDS-PAGE and WB with the indicated antibodies. (C) Indirect IF of PFA fixed fibroblasts (patient, WT#1, WT#2) stained for SLC3A2 (yellow) and DAPI (cyan) was analyzed by confocal microscopy. The images show a single plane of a Z-stack. Scale bar = 10 μm. (D, E) RPE1 WT, TXNIP^KO and TXNIP^KO reconstituted with TXNIP harboring the patient mutation (c.174_175delinsTT (p.Gln58Hisfs*2), TXNIP^mut1) or the patient mutation (c.642_643insT (p.Ile215Tyrfs*59), TXNIP^mut2) were grown in growth medium ( + serum) or serum starved for 24 h (− serum). (D) Total cell lysates were analyzed by SDS-PAGE and WB with the indicated antibodies. WB quantification of SCL7A5 normalized to amido black. Box plots represent the median (centre line) and the interquartile range (25th to 75th percentile box). The whiskers show the minimum and maximum values. ($n$ = 4, two-way ANOVA, Sidak multiple comparison test, WT serum vs −serum: $P$ = 6.12E-5). (E) Indirect IF of PFA fixed cells (WT, TXNIP^KO and TXNIP^mut2) stained for SLC3A2 (yellow) and DAPI (cyan) was analyzed by confocal microscopy. The images show a single plane of a Z-stack. Scale bar = 10 μm. (F) ALFA-TXNIP (magenta) and SLC3A2 (yellow) was analyzed by confocal microscopy. The images show a single plane of a Z-stack. Regions of interest (ROI, white box) were magnified. Scale bar = 10 μm. Source data are available online for this figure.

The patient showed consistent metabolic alterations compatible with an AA transporter deficiency. Blood plasma concentrations of several large neutral amino acids (LNAAs, including L, I, V) were elevated throughout the years 2014 – 2025 (Appendix Table S1). The increased molar ratio of the LNAAs (L, I, V) relative to aromatic AAs (F, Y), resulted in a consistently elevated Fischer's ratio (FR, 2014: FR = 4.46; 2016: FR = 5.38; 2018: FR = 5.90; 2021; FR = 6.98; 2022: FR = 4.23; 2025 FR = 5.36; FR reference range = 2.10–4). The methionine levels were not dramatically altered.

To examine how this loss-of-function *TXNIP* variant affected SLC7A5 downregulation, we used primary patient-derived fibroblasts. WB analysis of the patient fibroblasts showed complete loss of the TXNIP protein (Fig. 3B). In these cells, SLC7A5 was no longer downregulated in response to serum starvation (Figs. 3B and EV3A). Indirect IF and confocal microscopy showed that SLC3A2 remained at the PM in response to serum starvation in the patient-derived fibroblasts, while SLC3A2 was no longer detected in the primary control fibroblasts (Figs. 3C and EV3B).

To directly compare the functional effects of this *TXNIP* variant (p.Ile215Tyrfs*59; hereafter TXNIP^mut2) with the previously identified *TXNIP* variant (p.Gln58Hisfs*2; hereafter TXNIP^mut1) (Katsu-Jimenez et al, 2019), we expressed their respective cDNAs in RPE1 TXNIP^KO cells to generate an isogenic cellular background. In cells expressing either TXNIP variant, SLC7A5-SLC3A2 degradation was blocked (Fig. 3D). Under these conditions (which do not trigger NMD), the expression of a truncated TXNIP^mut2 protein was detected (Fig. 3D, lanes 5 and 6). ALFA-TXNIP^mut2 localized to the PM, but failed to mediate SLC7A5-SLC3A2 endocytosis in response to serum starvation (Figs. 3E,F and EV3C).

Cells carrying the biallelic TXNIP^mut1 variant have been reported to have functional mitochondria; however the channeling of glucose and pyruvate into oxidative phosphorylation was inefficient (Katsu-Jimenez et al, 2019). We confirmed that RPE1 TXNIP^KO cells, as well as RPE1 TXNIP^KO cells expressing TXNIP^mut1 or TXNIP^mut2 cells exhibited reduced oxygen consumption rates compared to wild-type cells (Fig. EV3D).

We concluded that, *TXNIP* loss-of-function variants impaired SLC7A5 downregulation in cells entering quiescence. In growing cells, loss of TXNIP function was associated with reduced oxidative phosphorylation and increased glucose uptake. In human patients, these cellular defects converged into metabolic imbalances that underlie a wide spectrum of clinical manifestations.

## PPxY motifs of TXNIP were required for the endocytosis of SLC7A5

Next we analyzed if the two PPxY motifs in the C-terminal region of TXNIP (Fig. 4A) enabled the interaction with different HECT-type ubiquitin ligases to promote SLC7A5-SLC3A2 endocytosis (Alvarez, 2008; Lee et al, 2024; Liu et al, 2016; Nabhan et al, 2010; Qi et al, 2014; Rauch and Martin-Serrano, 2011; Shea et al, 2012; Zhang et al, 2010). Therefore, we introduced point mutations in the first PPxY motif (TXNIP^PPCY331AACA, hereafter TXNIP^PPxY331) or in both PPxY motifs (TXNIP^PPCY331AACA, PPTY375AATA, hereafter TXNIP^PPxY), and expressed these mutants in TXNIP^KO RPE1 cells. The protein levels of the PPxY mutants were higher compared to endogenous TXNIP (Figs. 4B and EV4A), and the endocytic downregulation of SLC7A5-SLC3A2 was readily blocked in TXNIP^PPxY331 cells and also in TXNIP^PPxY cells (Figs. 4B and EV4A,B). SLC3A2 remained at the PM in response to serum starvation (Figs. 4C,D and EV4C,D; Appendix Fig. S3A).

A fraction of ALFA-TXNIP^PPxY331 localized to the PM in proliferating cells, similar to wild-type ALFA-TXNIP. In response to serum starvation, the signal for ALFA-TXNIP^PPxY331 persisted at the PM, where it co-localized with SLC3A2, while ALFA-TXNIP was cleared from the PM together with SLC3A2 (Fig. 4D). These results showed that the PPxY motif at position 331 of TXNIP was required for the endocytic downregulation of SLC7A5-SLC3A2.

Next, we used label-free proteomics to determine which HECT-type E3 ubiquitin ligases interacted with the PPxY motifs of TXNIP. To this end, we immunoprecipitated ALFA-TXNIP, ALFA-TXNIP^PPxY331 or ALFA-TXNIP^PPxY from HEK293T cells, followed by quantitative interactome analysis (Figs. 4E,F and EV4E; Dataset EV1). Among the E3 ligases that co-immunoprecipitated with TXNIP, NEDD4L was by far the most abundant interactor (Fig. 4E). Its interaction with TXNIP was dependent on the first PPxY motif and was further reduced when both PPxY motifs were mutated (Figs. 4E,F and EV4E). Consistently, ALFA-TXNIP expressed at near-endogenous levels in RPE1 TXNIP^KO cells co-immunoprecipitated NEDD4L (Fig. 4G).

Other HECT-type E3 ubiquitin ligases—such as WWP2, NEDD4, WWP1, ITCH, HECW1, and HECW2—were also specifically co-immunoprecipitated with TXNIP in a PPxY motif dependent manner (Figs. 4E,F and EV4E). Yet, their abundance in the co-immunoprecipitation was significantly lower compared to NEDD4L: WWP2 was approximately eightfold less abundant,

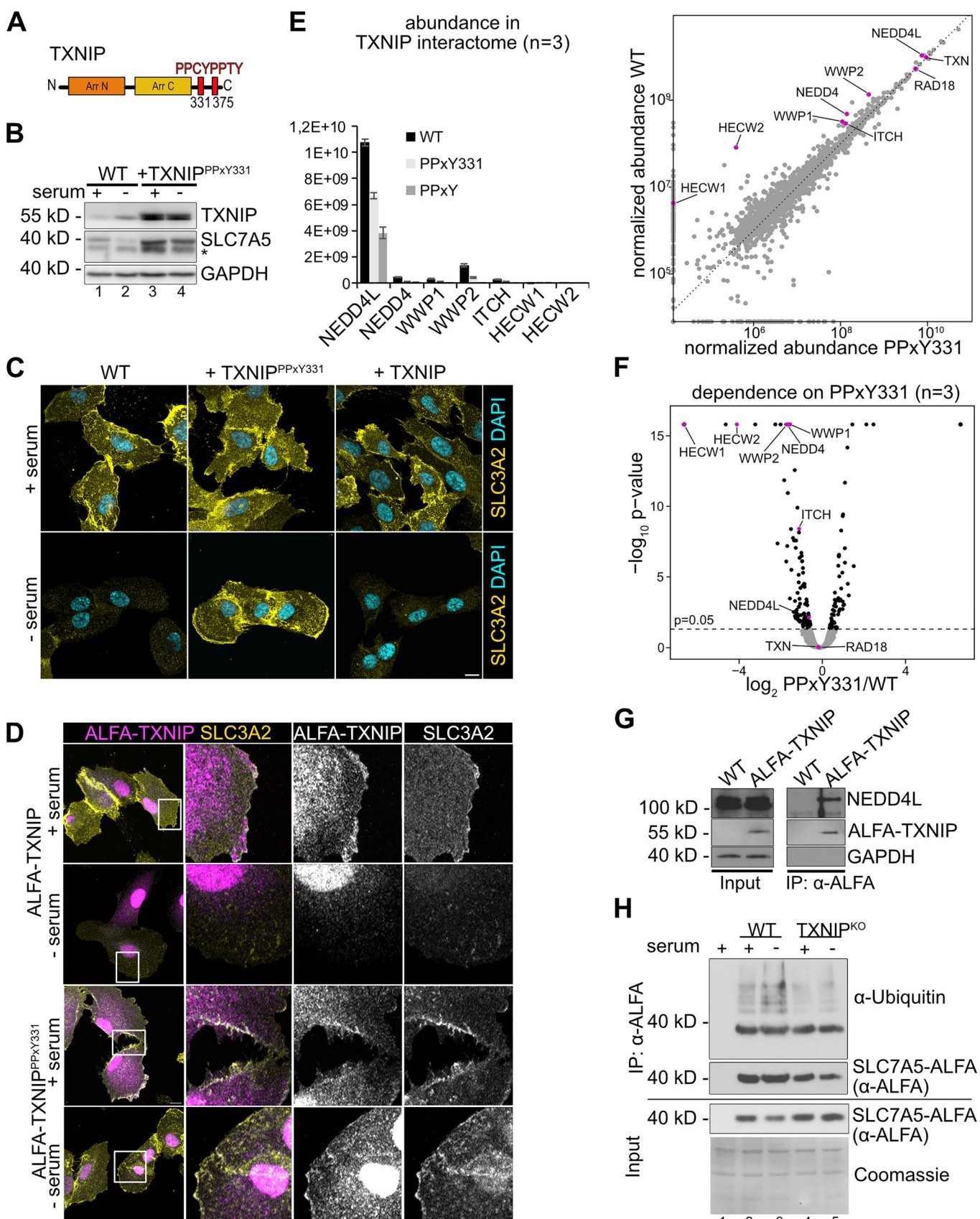

◄ **Figure 4. PPxY motifs of TXNIP in the downregulation of SLC7A5.**

(A) Schematic representation of TXNIP protein harboring PPxY motifs. (B) Total cell lysates from WT cells or TXNIP$^{KO}$ reconstituted with ALFA-TXNIP$^{PPxY331}$ mutants were analyzed by SDS-PAGE and WB with the indicated antibodies. The asterisk (*) labels a second band that was only detected using the monoclonal SLC7A5 antibody. (C, E) Indirect IF of PFA fixed cells (WT, TXNIP$^{KO}$ and TXNIP$^{KO}$ reconstituted with ALFA-TXNIP or ALFA-TXNIP$^{PPXY331}$) was analyzed by confocal microscopy. The images show a single plane of a Z-stack (C) Cells were stained for SLC3A2 (yellow) and DAPI (cyan). Scale bar = 10 μm. (D) Cells were stained for ALFA-TXNIP (magenta) and SLC3A2 (yellow). Regions of interest (ROI, white box) were magnified. Scale bar = 10 μm. (E) Label free interactome analysis (n = 3) of ALFA-TXNIP, ALFA-TXNIPPP$^{PPxY331}$ and ALFA-TXNIP$^{PPxY}$: The abundance of HECT type ubiquitin E3 ligases interacting in a PPxY motif dependent manner is presented as a histogram (for ALFA-TXNIP, ALFA-TXNIPP$^{PPxY331}$ and ALFA-TXNIP$^{PPxY}$) and as scatter plot comparing interactomes of ALFA-TXNIP with ALFA-TXNIP$^{PPxY331}$. The HECT type ubiquitin ligases are highlighted in magenta. The abundances of all proteins were normalized to unspecific background binding to empty beads. (F) Volcano plot of log2 transformed abundances from (E). HECT-type ubiquitin ligases are highlighted in magenta. The adjusted p-values were calculated using the Benjamin-Hochberg method (n = 3). (G) SDS-PAGE and WB analysis with the indicated antibodies from non-denaturing IPs of ALFA-TXNIP expressed in TXNIP$^{KO}$ cells or from WT cells (not expressing ALFA-TXNIP). (H) SDS-PAGE and WB analysis with the indicated antibodies from denaturing ALFA-SLC7A5 IP from SLC7A5$^{KO}$ RPE1 reconstituted with ALFA-SLC7A5. Source data are available online for this figure.

NEDD4 approx. 23-fold less abundant, and WWP1, ITCH, HECW1, and HECW2 were each reduced by at least 35-fold (Fig. 4E). These differences cannot be attributed to difference in cellular abundance: ITCH is estimated to be the most abundant of these proteins with $6.5 \times 10^4$ molecules/cell, and NEDD4L ($3.7 \times 10^4$ molecules/cell), WWP2 ($1.8 \times 10^4$ molecules/cell), and NEDD4 ($1.3 \times 10^4$ molecules/cell) are within a similar two- to threefold abundance range (Cho et al, 2022). The interaction of TXNIP with TXN (Thioredoxin) (Nishiyama et al, 1999), was independent of PPxY motifs (Figs. 4E,F and EV4E).

To better characterize the interaction of TXNIP with the less abundant HECT-type ubiquitin ligases we first immobilized equal amounts ALFA-tagged TXNIP, TXNIP$^{PPxY331}$or TXNIP$^{PPxYmut}$ on ALFA-beads (Fig. EV4F; Appendix Fig. S3B). These TXNIP beads were then incubated with cell lysates from HEK293T cells over-expressing YFP-tagged HECT-type ubiquitin ligases. The binding of the YFP-tagged HECT-type ubiquitin ligases to TXNIP on the beads was analyzed by fluorescence microscopy and quantified (Fig. EV4F; Appendix Fig. S3B). This binding-assay confirmed that TXNIP has the capacity to interact with these HECT-type ubiquitin ligases in a PPxY motif dependent manner. However, in contrast to the PPxY331 sensitive interaction with endogenous HECT-type ubiquitin ligases, the binding of the overexpressed ubiquitin ligases to TXNIP was only significantly reduced when both PPxY motifs were mutated (Fig. EV4F; Appendix Fig. S3B).

Taken together these experiments demonstrated that different HECT-type ubiquitin ligases interact with TXNIP in a PPxY motif dependent manner, with NEDD4L being the most abundant binding partner, followed by WWP2 and NEDD4. This suggests a ranked interaction profile, with possible redundancies, rather than a strict hierarchy.

To examine if entry into quiescence induced TXNIP-mediated ubiquitination of SLC7A5, we replaced endogenous SLC7A5 with C-terminally ALFA-tagged SLC7A5 (SLC7A5-ALFA). Therefore, we generated SLC7A5$^{KO}$ cells (using CRISPR/Cas9-mediated gene editing, Appendix Fig. S3C), as well as SLC7A5$^{KO}$ TXNIP$^{KO}$ double knock out cells and stably over-expressed SLC7A5-ALFA. To detect ubiquitination, SLC7A5-ALFA was immunoprecipitated under denaturing conditions (1% SDS) before and after 7 h of serum starvation. Polyubiquitination of SLC7A5-ALFA was detected under steady state conditions, and appeared to slightly increase 7 h after serum starvation. The polyubiquitination of SLC7A5-ALFA was markedly reduced in the TXNIP$^{KO}$ cells (Fig. 4H). Hence, TXNIP seemed to be required for the efficient polyubiquitination of SLC7A5-ALFA.

Taken together, these findings suggested that TXNIP used its first PPxY motif at position 331 to interact primarily with NEDD4L, but also with other HECT-type ubiquitin ligases, to promote the efficient ubiquitination of SLC7A5-SLC3A2, resulting in its endocytic downregulation.

## AKT signaling controls TXNIP-mediated downregulation of SLC7A5

Next, we addressed how the TXNIP dependent downregulation of SLC7A5 was regulated. Earlier work demonstrated that AKT directly phosphorylates TXNIP on serine 308 to inhibit endocytosis of SLC2A1 and increase glucose uptake (Waldhart et al, 2017). To examine if AKT also controlled the endocytic downregulation of SLC7A5-SLC3A2, we treated RPE1 cells with the AKT inhibitor MK2206. MK2206 treatment caused the endocytic downregulation of the SLC7A5-SLC3A2 heterodimer (Fig. 5A, lane 3; Appendix Fig. S4A), but not of SLC1A5 (Appendix Fig. S4B). Inhibition of MEK with PD0325901 triggered the selective endocytic down-regulation of SLC1A5, but not of SLC7A5-SLC3A2 (Fig. 5A, lane 4 and Appendix Fig. S4A,B). These results suggested that distinct mitogenic signaling cascades regulated the PM levels of different AA transporters, with AKT controlling SLC7A5-SLC3A2 endocytosis.

To test if AKT controlled SLC7A5 endocytosis via TXNIP, we treated RPE1 WT and TXNIP$^{KO}$ cells with MK2206. In the TXNIP$^{KO}$ cells, endocytosis and degradation of SLC3A2 was blocked in response to MK2206 treatment, suggesting that TXNIP functioned downstream of AKT to mediate SLC7A5-SLC3A2 endocytic degradation (Fig. 5B, C, lanes 1–3 and 13–15). Consistently, the phospho-mimetic TXNIP$^{S308D}$ mutant blocked SLC3A2 endocytosis in response to serum starvation or AKT inhibition, while the non-phosphorylatable TXNIP$^{S308A}$ mutant allowed SLC7A5-SLC3A2 endocytic degradation (Fig. 5B,C; Appendix Fig. S4C). These findings were corroborated by cell surface FACS analysis of SLC3A2 (Fig. 5D; Appendix Fig. S4D).

TXNIP forms a disulfide bound (S-S) via cysteine 247 to inhibit TXN (thioredoxin) activity (Hwang et al, 2014; Patwari et al, 2006). To test the role of this disulfide bridge for SLC7A5-SLC3A2 endocytosis, we generated mutant TXNIP$^{C247S}$, which disrupts the S-S bond with TXN (Patwari et al, 2009). The expression of TXNIP$^{C247S}$ in TXNIP$^{KO}$ cells rescued the endocytic degradation of SLC7A5-SLC3A2, similar to WT TXNIP (Fig. 5B–D; Appendix Fig. S4D). Together with the finding that the interaction of TXNIP with TXN was independent of PPxY motifs (Figs. 4E,F and EV4E),

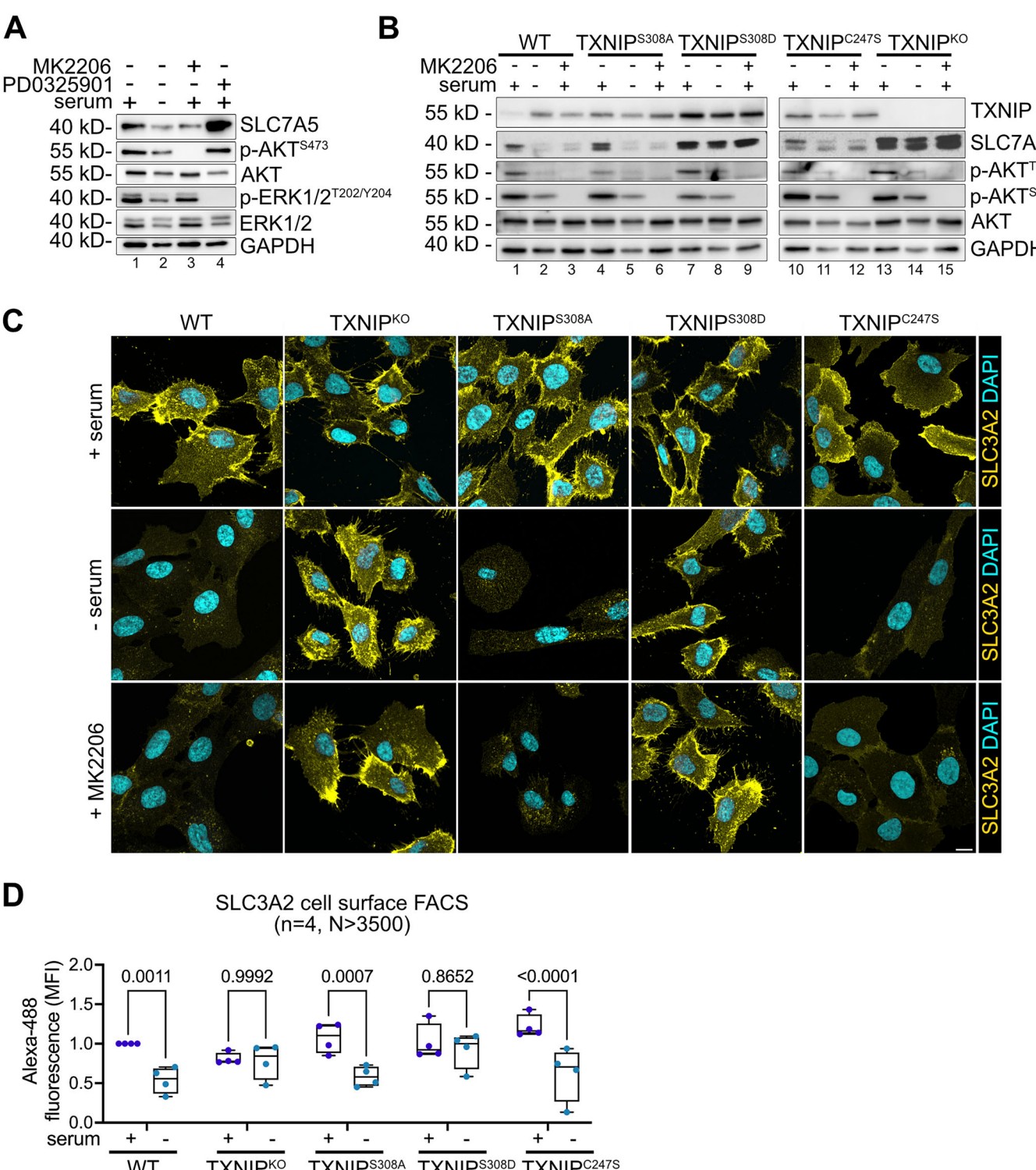

we concluded that SLC7A5 endocytosis was independent of the ability of TXNIP to form disulfide bridges with TXN.

These results demonstrated that AKT-mediated phosphorylation of TXNIP at serine 308 inhibited SLC7A5-SLC3A2 endocytosis, while reduction of AKT signaling enabled TXNIP-mediated SLC7A5-SLC3A2 endocytic degradation.

## TXNIP-mediated downregulation of SLC7A5 was essential to adjust intracellular AA levels and metabolic signaling in cells entering quiescence

Our results so far linked growth factor signaling via AKT with the TXNIP-mediated control of AA acquisition through SLC7A5-

**Figure 5. TXNIP and AKT signalling.**

(A) Total cell lysates from WT cells before or after 24 h of serum starvation or incubation with 1 μM MK2206 or 5 μM PD0325901. Cells were analysed by SDS-PAGE and WB with the indicated antibodies. (B) Indirect IF of PFA fixed cells (WT, TXNIP$^{KO}$ and TXNIP$^{KO}$ reconstituted with TXNIP$^{S308A}$, TXNIP$^{S308D}$ or TXNIP$^{C247S}$) before or after serum starvation or incubation with 1 μM MK2206 stained for SLC3A2 (yellow) and DAPI (cyan). Cells were analysed by confocal microscopy. The images show a single plane of a Z-stack. Scale bar = 10 μm. (C) Total cell lysates from cells treated as in (B) were analysed by SDS-PAGE and WB with the indicated antibodies. (D) Quantification of SLC3A2 cell surface FACS from the indicated cells. Box plots represent the median (centre line) and the interquartile range (25th to 75th percentile box). The whiskers show the minimum and maximum values. ($n = 4$, $N > 3500$ cells, two-way ANOVA, Sidak's multiple comparisons test, TXNIP$^{C247S}$ +serum vs TXNIP$^{C247S}$ –serum: $P = 7.8E-5$). Source data are available online for this figure.

SLC3A2. To directly test whether TXNIP-mediated endocytosis of SLC7A5-SLC3A2 adjusted AA uptake in cells entering quiescence, we measured the cellular uptake of [³H]-leucine. In response to serum starvation, WT cells downregulated [³H]-leucine uptake, while TXNIP$^{KO}$ cells did not reduce [³H]-leucine uptake (Fig. 6A). Upon JPH203 treatment, a selective inhibitor of SLC7A5 (Oda et al, 2010), the uptake of [³H]-leucine was blocked in WT cells and TXNIP$^{KO}$ cells alike (Fig. 6A). Also, in SLC7A5$^{KO}$ cells, the uptake of [³H]-leucine was strongly impaired (Appendix Fig. S5A).

To assess the role of SLC7A5 downregulation on intracellular free AA levels, we used quantitative mass spectrometry. We quantified free levels of 17 different proteinogenic AA in WT, TXNIP$^{KO}$ and SLC7A5$^{KO}$ RPE1 cells. In SLC7A5$^{KO}$ cells the intracellular levels of L, I, W, V, F, Y were low under all conditions (Fig. 6B), with exception for M, which was upregulated (Appendix Fig. S5B). These LNAAs are known substrates of SLC7A5 (Kandasamy et al, 2018) (Fig. 6B). In quiescent WT cells, the free levels of most these AAs (L, I, W, V, F, Y) decreased (Fig. 6B). In contrast, the levels of L, I, W, V, F, Y were lower in growing TXNIP$^{KO}$ cells, and they did not decrease when TXNIP$^{KO}$ cells entered quiescence (Fig. 6B). Neither loss of SLC7A5 nor loss of TXNIP affected the levels K, R, S, E, Q, P, G, A (Appendix Fig. S5B,C). Taken together, these data indicated that the loss of TXNIP affected the ability of cells to adjust SLC7A5 dependent AA transport.

To characterize the consequence resulting from the failure to adjust AA uptake in TXNIP$^{KO}$ cells, we determined how mTORC1 dependent translational control was affected. In quiescent WT cells, phosphorylation of S6K on threonine 389 was reduced compared to proliferating cells, as was phosphorylation of the S6K substrate S6 on serine 240/244, in keeping with lower mTORC1 activity (Fig. 6C, lane 3 and 4, Fig. 6D,E). In TXNIP$^{KO}$ cells, S6K and S6 phosphorylation also decreased, but the phosphorylation of S6K and S6 remained higher compared to WT cells (Fig. 6C, lane 1 and 2, Fig. 6D,E), indicating elevated mTORC1 activity. Another important mode for translational control is eIF2α phosphorylation at serine 51 by GCN2 (but also other kinases of the integrated stress response) in response to lower AA levels. In quiescent WT cells, eIF2α phosphorylation at serine 51 increased (Fig. 6C, lane 3 and 4, Fig. 6F). However, in growing TXNIP$^{KO}$ cells, eIF2α phosphorylation appeared to be slightly elevated, but it did not further increase, as these cells transitioned into quiescence (Fig. 6C, lane 1 and 2, Fig. 6F). It seemed that TXNIP-mediated SLC7A5-SLC3A2 endocytosis was required to decrease AA uptake in quiescence, which helped to adapt signaling towards the translation machinery, possibly to adjust translation to quiescence.

In line with this concept, [³⁵S]-methionine and [³⁵S]-cysteine incorporation into newly synthesized proteins remained higher in quiescent TXNIP$^{KO}$ cells, and appeared to increase faster after the re-addition of serum (Fig. 6G). The increased translation in the TXNIP$^{KO}$ cells required SLC7A5, because it was no longer observed in TXNIP$^{KO}$ SLC7A5$^{KO}$ double mutant cells (Fig. 6G).

We concluded that TXNIP-mediated endocytosis of SLC7A5-SLC3A2 lowered the uptake and the intracellular levels of LNAAs such as L, I, W, V, F, Y, thereby helping cells to reduce mTORC1 signaling and to dampen translational activity.

## TXNIP-mediated endocytic degradation of SLC7A5 restrains cell proliferation and is required for cell survival during extend quiescence

Finally, we determined how the TXNIP-mediated regulation of SLC7A5 mediated AA uptake contributed to cell growth and quiescence. Therefore, we first compared the DNA content of WT, TXNIP$^{KO}$ and TXNIP$^{KO}$ cells which re-expressed TXNIP during entry and exit from serum starvation induced quiescence (Fig. 7A).

In a proliferating cell population and 24 h after serum starvation, the DNA content of WT, TXNIP$^{KO}$ and rescued cells was similar (Fig. 7A, timepoint 0 h and 24 h). Interestingly, 24 h after serum re-addition (timepoint 48 h), a larger fraction of the TXNIP$^{KO}$ cells (35%, ± 1%) had already reached the G2/M phase of the cell cycle. In comparison, only 13% (± 10%) of WT cells or 17% (± 12%) of TXNIP$^{KO}$ cells re-expressing TXNIP reached G2/M phase and more cells (35% ± 10% of WT cell and 25% ± 11% of rescued TXNIP$^{KO}$) were still in S-Phase (Fig. 7A).

Consistent with a faster cell cycle progression of TXNIP$^{KO}$ cells during exit from quiescence, these cells resumed cell proliferation faster compared to WT cells, which was dependent on SLC7A5 (Fig. 7B). Moreover, the overall cell proliferation rate of TXNIP$^{KO}$ cells or of TXNIP$^{KO}$ cells expressing TXNIP$^{S308D}$ was increased compared to WT or TXNIP$^{KO}$ cells expressing TXNIP or TXNIP$^{S308A}$ (Fig. EV5A). The increased proliferation of TXNIP$^{KO}$ cells was dependent on SLC7A5, since the double knock out TXNIP$^{KO}$ SLC7A5$^{KO}$ grew slower, similar to SLC7A5$^{KO}$ cells (Fig. EV5A). When co-cultured, TXNIP$^{KO}$ cells expressing mCherry outcompeted TXNIP$^{KO}$ cells that re-expressed GFP-TXNIP (Fig. 7C).

In contrast to this growth advantage, quiescent TXNIP$^{KO}$ cells invariably began to die between 48 - 60 h after serum withdrawal (Figs. 7D and EV5B). All other cells survived extended periods of quiescence. Remarkably, deletion of SLC7A5 in TXNIP$^{KO}$ cells, or pharmacological inhibition of SLC7A5 with JPH203 prevented cell death during extended quiescence (Figs. 7D and EV5B). Also, the re-expression of GFP-TXNIP rescued cell death (Figs. 7D and EV5B).

Taken together, these findings indicated that the TXNIP-mediated downregulation of SLC7A5 was required for cell survival during prolonged quiescence. The concurrent reduction of AA transport helped to dampen translation and thereby supported cell survival during quiescence, restrained cell cycle re-entry, and

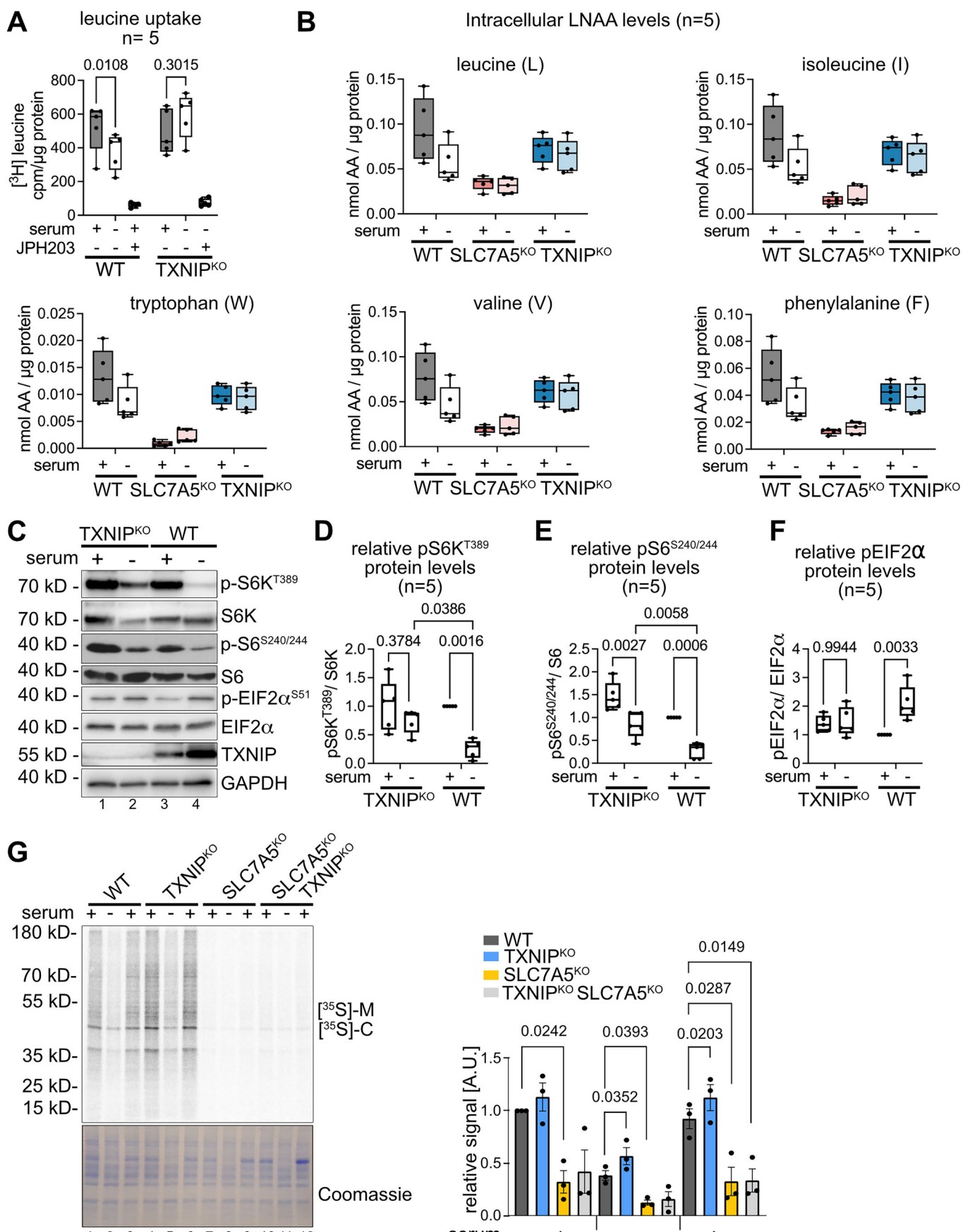

**Figure 6.   TXNIP mediated degradation of SLC7A5 contributed to metabolic signalling.**

(A) WT and TXNIP$^{KO}$ cells were grown in growth medium ( + serum), serum starved for 24 h (- serum) or treated with 10 µM JPH203. Cells were incubated with [3H]-leucine for 15 min, washed and lysed. Cell lysates were analysed by scintillation counting. cpm values were normalized to total protein content. Box plots represent the median (centre line) and the interquartile range (25th to 75th percentile box). The whiskers show the minimum and maximum values. ($n = 5$, two-way ANOVA, Tukey's multiple comparisons test). (B) WT, SLC7A5$^{KO}$ and TXNIP$^{KO}$ cells were grown in growth medium ( + serum) or serum starved for 24 h ( − serum). Mass spectrometry analysis of free AAs, normalized to total protein content. Box plots represent the median (centre line) and the interquartile range (25th to 75th percentile box). The whiskers show the minimum and maximum values. (nmol AA/µg protein, $n = 5$). (C) Total cell lysates from WT and TXNIP$^{KO}$ cells before or after 24 h serum starvation were analysed by SDS-PAGE and WB with the indicated antibodies. Box plots represent the median (centre line) and the interquartile range (25th to 75th percentile box). The whiskers show the minimum and maximum values. (D–F) WB quantification of pS6K$^{T389}$, normalized to total S6K. WB quantification of pS6$^{S240/244}$, normalized to total S6 and WB quantification of pEIF2 α$^{S51}$, normalized to total EIF2α protein levels ($n = 5$, two-way ANOVA). (G) Cells were incubated with [35S]-methionine, [35S]-cysteine for 2 min, before CHX (10 µg/ml) was added, cells were lysed and analysed by SDS-PAGE and autoradiography follow by densitometric analysis. The bar chart show mean values ($n = 3$, SEM, paired $t$ test). Source data are available online for this figure.

limited overall cell proliferation (Fig. 7E). This TXNIP dependent mechanism thus promoted metabolic flexibility and integrated with growth control via AKT signaling.

# Discussion

We made progress toward understanding how growth factor signaling regulates TXNIP-mediated endocytosis of SLC7A5-SLC3A2 to adapt the transport of essential AA (mainly LNAAs) for metabolic scaling in growing and quiescent cells. The framework of our model (Fig. 7E) is based on the following results: (1) Human cells that enter quiescence downregulate AA uptake, in part, by selective endocytosis of the AA transporters; (2) the α-arrestin TXNIP interacts with HECT-type ubiquitin ligases, foremost NEDD4L, to mediate the selective endocytosis of SLC7A5-SLC3A2; (3) TXNIP-mediated endocytosis is suppressed in growing cells by AKT signaling and (4) TXNIP-mediated SLC7A5-SLC3A2 endocytosis helps to lower AA uptake, translation and thereby promotes quiescent cell survival. (5) These processes contribute to metabolic flexibility in humans. A novel *TXNIP* loss-of-function variant, that impairs AA endocytosis, causes a rare metabolic disease.

Our data, together with a previous report on three siblings from a single consanguineous family who were homozygous for another *TXNIP* loss-of-function variant (c.174_175delinsTT) (Katsu-Jimenez et al, 2019), support TXNIP deficiency as a novel inherited metabolic disorder. Loss of TXNIP causes specific metabolic alterations on the cellular level that are associated with neonatal lactic acidosis, recurrent hypoglycaemia, and specific anomalies in AA metabolism. The full spectrum of associated clinical manifestations remains to be defined. The three siblings were treated with long-term dichloroacetic acid (DCA). Rather than directly clearing lactate from serum, DCA inhibits pyruvate dehydrogenase kinase and thus stimulates pyruvate dehydrogenase activity and oxidative decarboxylation of pyruvate, resulting in reduced serum lactate levels (Stacpoole, 2017) (Whitehouse et al, 1974). Two of the three affected children with the c.174_175delinsTT (p.Gln58Hisfs*2) *TXNIP* variant (a boy and a girl) had normal development during childhood; the clinical phenotype of one sibling (the youngest boy) was more severe. This boy suffered from recurrent hypoglycaemia, neonatal muscular hypotonia and failure to thrive, and was under investigation for autism spectrum disorder, similar to our patient with the c.642_643insT *TXNIP* (p.Ile215Tyrfs*59) variant. In addition, our patient has an increasing frequency of epileptic

seizures and developmental delay. The three siblings developed low plasma methionine concentrations. No other AA anomalies were specified in the previous report except for a transient general hyperaminoaciduria in one child (Katsu-Jimenez et al, 2019). The detailed metabolic investigations in our case revealed relative normal methionine levels, but identified other specific AA alterations in line with impaired SLC7A5 endocytosis in the case of complete TXNIP loss. How these cellular metabolic defects are linked to clinical manifestations such as possible autism and the severe epileptic developmental disorder, remains to be investigated. We speculate that these phenotypes could be related to roles of SLC7A5 in neurons and in the blood brain barrier (Knaus et al, 2023; Tarlungeanu et al, 2016). In the cerebral cortex, SLC7A5-mediated transport of large neutral amino acid appears interconnected with lipid metabolism and required for perinatal neuronal excitability and survival (Knaus et al, 2023; Tarlungeanu et al, 2016). Consistently, pathogenic variants in SLC7A5 lead to autism spectrum disorders (Tarlungeanu et al, 2016). Hence it is not unlikely that dysregulation of SLC7A5, caused by loss of TXNIP, might affect brain development. To address this point, additional experiments are required.

It is clear that complete loss of TXNIP disrupts metabolic homeostasis on the cellular level, possibly resulting in the failure to coordinate AA and glucose uptake with oxidative phosphorylation across different organs, which leads to metabolic rigidity on the organismal level. Interestingly, the phenotypic spectrum in patients is variable in severity. Thus, other yet unknown genetic and/or non-genetic modifiers may either exacerbate or partly compensate loss of TXNIP function (e.g. diet, differences in medical treatment) and likely play a role in establishing disease phenotypes. These modifiers remain to be identified.

On the molecular level, a partial functional redundancy among the PPxY motif containing α-arrestins (ARRDC1-4, TXNIP) cannot be excluded. Clustal omega multisequence alignment of TXNIP with human ARRDC1-4 revealed between 23 and 44% identify on amino acid level. Moreover, these α-arrestins use their PPxY motifs to interact with a similar set of HECT-type ubiquitin ligases (Lee et al, 2024; Rauch and Martin-Serrano, 2011). Therefore, it is not impossible that these α-arrestins are—at least in part—functionally redundant with TXNIP, perhaps even in a cell type specific manner. A similar argument for redundancy could be made for the HECT-type ubiquitin ligases. While NEDD4L was the most abundant interactor of TXNIP, also WWP2 and NEDD4 interacted with TXNIP in a PPxY motif specific manner. Hence different α-arrestins could interact with similar sets of HECT-type

ubiquitin ligase thereby forming a robust network to fine-tune nutrient acquisition in a cell type and tissue specific responses to hormonal or metabolic cues.

Several studies in mice show that TXNIP is important during the transition between fed and fasted states, which was attributed to its role in glucose and energy metabolism (Chutkow et al, 2008; Hui et al, 2008; Hui et al, 2004; Sheth et al, 2005). Probably many of these phenotypes relate to the role of TXNIP in regulating cellular glucose uptake by mediating the selective endocytosis of SLC2A1 (GLUT1) and SLC2A4 (GLUT4) (Kim et al, 2024; Qualls-Histed et al, 2023; Waldhart et al, 2017; Wu et al, 2013). Our findings imply that some of the described TXNIP dependent effects on cellular metabolism might also be linked to TXNIP dependent regulation of SLC7A5-SLC3A2 and the import of LNAAs. Perhaps, excess I, L, and V are shunt into branched chain AA catabolism and thereby help to fuel the TCA cycle as cells enter and exit quiescence. Thereby glucose could be spared for other metabolic processes (e.g. lipid synthesis) (Scholz et al, 2025).

While it was clear how proliferating cells benefit from the upregulation of SLC7A5-SLC3A2, it is now also apparent that quiescent cells profit from the downregulation of SLC7A5-SLC3A2. One rational for the selective endocytosis of this AA transporter could be that the cellular demand for essential AAs must be matched to lower rates of translation in quiescent cells. In line with this concept, also quiescent mammary epithelial cells reduced levels of V, L, I and Q (Coloff et al, 2016). Similar results were obtained in a non-malignant murine pro-B lymphocyte cell line (FL5.12), in which the abundance of SLC3A2 at the plasma membrane was regulated by the presence of growth factors (Edinger and Thompson, 2002). In those cells, the consumption rate of essential AAs was decreased in G0 arrest and increased upon re-entry into the cell cycle (Lee et al, 2017). Hence, the downregulation of AA transporters provide means to avoid an excess/imbalance of free intracellular AAs. Reducing free intracellular AA levels would facilitate the controlled dampening of mTORC1 signaling and the ensuing reduction of translation rates, possibly in conjugation with the integrated stress response. Thereby, the TXNIP-mediated selective downregulation of SLC7A5-SLC3A2 and the ensuing decreased import of essential AAs, helps to promote the switch from anabolic to catabolic metabolism. Moreover, the down-regulation of AA transporters upon entry into quiescence might also contribute to the barrier for cellular transformation (Palm and Thompson, 2017) (Kaira et al, 2013; Kalaany, 2023; Kanai, 2022).

TXNIP was specifically required for the endocytosis of SLC7A5-SLC3A2, but not for the endocytosis of SLC7A11-SLC3A2. Moreover, in RPE1 cells entering quiescence, SLC2A1 (GLUT1) was not downregulated. Hence, it seems that TXNIP can discriminate, in a context dependent manner, between targeting SLC7A5-SLC3A2 or SLC2A1 (GLUT1) or SLC2A4 (GLUT4) for endocytosis. Since AKT-mediated phosphorylation invariably appears to inactivate TXNIP, and dephosphorylation re-activates it, additional mechanisms must confer TXNIP selectivity towards these nutrient transporters. We consider it likely, that the exposure of sorting motifs in their cytosolic tails (Qualls-Histed et al, 2023; Waldhart et al, 2017; Wu et al, 2013) could regulate the binding of activated TXNIP and thus controls selective endocytosis to adapt nutrient uptake. The exposure of these sorting motifs could be dependent on the metabolic context / state of the cell. Indeed, yeast α-arrestins can detect N- or C-terminal acidic sorting motifs in amino acid transporters that are exposed in response to amino acid excess or starvation (Ivashov et al, 2020) (Guiney et al, 2016). Inspection of the SLC7A5 sequence indicates a possible N-terminal acidic sorting motif (17EEKEEAREK25). Two lysine residues (K19, K25) in this sequence can be ubiquitinated upon protein kinase C (PKC) activation and mTORC1 inhibition (Barthelemy and Andre, 2019; Rosario et al, 2016). The molecular details of how TXNIP interacts with SLC7A5-SLC3A2 remains unclear at the moment.

Taken together, we conclude that TXNIP plays a critical role in regulating the abundance of SLC7A5–SLC3A2 at the plasma membrane, and thereby ensures that AA uptake is matched to the metabolic demands of proliferating and quiescent cells. In the future, it will be important to understand how these functions of TXNIP integrate with its roles in regulating glucose uptake and oxidative phosphorylation, and how TXNIP loss-of-function variants cause the complex metabolic defects and the diverse phenotypic spectrum observed in affected patients.

# Methods

**Reagents and tools table**

| Reagent/resource | Reference or source | Identifier or catalog number |
| --- | --- | --- |
| **Experimental models** | | |
| hTERT RPE1 | Institute of Pathophysiology, MUI | ATCC CRL-4000 |
| Patient-derived primary fibroblasts | This study | – |
| Primary fibroblasts WT#1 | Institute of Cell Biology, MUI | – |
| Primary fibroblasts WT#2 | Institute of Cell Biology, MUI | – |
| Mouse embryonic fibroblasts | Teis et al, 2006 | – |
| HEK293T | Institute of Pathophysiology, MUI | ATCC CRL-3216 |
| A549 | Institute of Pathology, MUI | ATCC CCL-185 |
| H460 | Institute of Pathology, MUI | ATCC HTB-177 |
| HCC4006 | Institute of Pathology, MUI | ATCC CRL-2871 |
| H2228 | Institute of Pathology, MUI | ATCC CRL-5935 |
| H226 | Institute of Pathology, MUI | ATCC CRL-5826 |
| H520 | Institute of Pathology, MUI | ATCC HTB-182 |
| H1299 | Institute of Pathology, MUI | ATCC CRL-5803 |
| **Recombinant DNA** | | |
| pCCL-EIF1α-BlastiR-TXNIP | This study | |
| pCCL-EIF1α-BlastiR-ALFA-TXNIP | This study | |
| pCCL-EIF1α-BlastiR-SLC7A5-ALFA | This study | |
| pCCL-EIF1α-BlastiR-TXNIPS308A | This study | |
| pCCL-EIF1α-BlastiR-ALFA-TXNIPS308A | This study | |
| pCCL-EIF1α-BlastiR-TXNIPS308D | This study | |
| pCCL-EIF1α-BlastiR-ALFA-TXNIPS308D | This study | |
| pCCL-EIF1α-BlastiR-TXNIPC247S | This study | |

| Reagent/resource | Reference or source | Identifier or catalog number |
|---|---|---|
| pCCL-EIF1α-BlastiR-ALFA-TXNIPC247S | This study | |
| pCCL-EIF1α-BlastiR-TXNIPp.Ile215TyrfsTer59 | This study | |
| pCCL-EIF1α-BlastiR-ALFA-TXNIPp.Ile215TyrfsTer59 | This study | |
| pCCL-EIF1α-BlastiR-TXNIPp.Gln58HisGly59* | This study | |
| pCCL-EIF1α-BlastiR-mCherry | This study | |
| pCCL-EIF1α-BlastiR-GFP-TXNIP | This study | |
| pCCL-EIF1α-BlastiR-ALFA-TXNIPPPxY331AAxA,PPxY375AAxA | This study | |
| pCCL-EIF1α-BlastiR-ALFA-TXNIPPPCY331AACA | This study | |
| pVSV-G | Clonetech | #631530 |
| psPAX2 | Institute of Cell Biology, MUI | |
| pDONR-ALFA-MCS | Institute of Cell Biology, MUI | |
| pDONR-MCS-ALFA | Institute of Cell Biology, MUI | |
| pDONR-221 | Invitrogen | Cat. #12536017 |
| pCCL-EIF1α-BlastiR-DEST | Institute of Pathophysiology, MUI | |
| pSpCas9(BB)-2A-GFP (PX458) | Addgene | Cat. #48138 |
| pCR3.1 YFP-NEDD4 | Woelk et al, 2006, Gahlot et al, 2024 | |
| pCR3.1 YFP-ITCH | Woelk et al, 2006, Gahlot et al, 2024 | |
| pCR3.1 YFP-WWP1 | Woelk et al, 2006, Gahlot et al, 2024 | |
| pCR3.1 YFP-WWP2 | Woelk et al, 2006, Gahlot et al, 2024 | |
| pCR3.1 YFP-HECW1 | Woelk et al, 2006, Gahlot et al, 2024 | |
| pCR3.1 YFP-HECW2 | Woelk et al, 2006, Gahlot et al, 2024 | |
| **Antibodies** | | |
| SLC1A5/ASCT2 | Merck | #abn73, RRID:AB_10807715 |
| SLC1A5/ASCT2 (V501) | Cell Signaling | #5345, RRID:AB_10621427 |
| SLC1A5/ASCT2 | Sigma-Aldrich | #HPA035240, RRID:AB_10604092 |
| SLC7A5/LAT1 (polyclonal) | Cell Signaling | #5347S, RRID:AB_10695104 |
| SLC7A5/LAT1 (E9O4D) (monoclonal) | Cell Signaling | #32683 |
| SLC7A5/LAT1 (D10) | Santa Cruz | #sc-374232, RRID:AB_10988206 |
| SLC3A2/4F2hc (D6O3P) | Cell Signaling | #13180S, RRID:AB_2687475 |
| SLC3A2/4F2hc | BD Biosciences | #556074, RRID:AB_396341 |
| SLC2A1/Glut1 | Merck | #07-1401, RRID:AB_1587074 |
| SLC38A2/SNAT2 | MBL International | #BMP081, RRID:AB_10597880 |
| SLC7A11/xCT (D2M7A) | Cell Signaling | #12691, RRID:AB_2687474 |
| TXNIP (D5F3E) | Cell Signaling | #14715S, RRID:AB_2714178 |
| p-S6K$^{T389}$ (108D2) | Cell Signaling | #9234S, RRID:AB_2269803 |
| S6K | Cell Signaling | #9202S, RRID:AB_331676 |
| p-S6$^{S240/244}$ | Cell Signaling | #2215S, RRID:AB_331682 |
| S6 (54D2) | Cell Signaling | #2317S, RRID:AB_2238583 |
| p-EIF2alpha | Cell Signaling | #3398 RRID:AB_2096481 |
| EIF2alpha | Cell Signaling | #5324 RRID:AB_10692650 |
| p-ERK1/2$^{T202/Y204}$ (D13.14.4E) | Cell Signaling | #4370S, RRID:AB_2315112 |
| ERK1/2 | Cell Signaling | #9102S, RRID:AB_330744 |
| p27$^{Kip1}$ (D69C12) | Cell Signaling | #3686S, RRID:AB_2077850 |
| p-AKT$^{S473}$ (D9E) | Cell Signaling | #4060S, RRID:AB_2315049 |
| p-AKT$^{T308}$ | Cell Signaling | #9275, RRID:AB_329828 |
| AKT | Cell Signaling | #9272S, RRID:AB_329827 |
| ALFA | NanoTag Biotechnologies | #N1581, RRID:AB_3075997 |
| Transferrin Receptor (Tfr) | DSHB | #G1/221/12, RRID:AB_2201506 |
| EGF Receptor (EGFR) (1005) | Santa Cruz | #sc-03-G |
| GAPDH (D4C6R) | Cell Signaling | #97166S, RRID:AB_2756824 |
| LAMP1 | DSHB | #H4A3, RRID:AB_2296838 |
| LAMP1 (D2D11) | Cell Signaling | #9091, RRID:AB_2687579 |
| NEDD4L | Abcam | # ab46521, RRID:AB_2149325 |
| NEDD4 | Cell Signaling | #5344S, RRID:AB_10560514 |
| WWP1 | Cell Signaling | #70140S, RRID:AB_3662810 |
| WWP2 | Cell Signaling | #41182S, RRID:AB_3662809 |
| ITCH | Cell Signaling | #12117S, RRID:AB_2797822 |
| GFP | Roche Life Science | (Roche Cat# 11814460001, RRID:AB_390913) |
| Ubiquitin (P4D1) | Santa Cruz | sc-8017, RRID:AB_628423 |
| Goat anti-mouse IgG-peroxidase | Sigma-Aldrich | #A4416, RRID:AB_258167 |
| Goat anti-rabbit IgG-peroxidase | Sigma-Aldrich | #A0545, RRID:AB_257896 |
| Alexa Fluor 488 goat anti-rabbit IgG | Invitrogen | #A11008, RRID:AB_143165 |
| Alexa Fluor 488 goat anti-mouse IgG | Invitrogen | #A11001, RRID:AB_2534069 |
| Alexa Fluor 568 goat anti-mouse IgG | Invitrogen | #A11031, RRID:AB_144696 |
| Alexa Fluor 568 goat anti-rabbit IgG | Invitrogen | #A11011, RRID:AB_143157 |

| Reagent/resource | Reference or source | Identifier or catalog number |
|---|---|---|
| NBR1 | Santa Cruz Biotechnology | Cat# sc-130380, RRID:AB_2149402 |
| P62 | Cell Signaling | Cat# 23214, RRID:AB_2798858 |
| TAX1BP1 | Sigma-Aldrich | Cat# HPA024432, RRID:AB_1857783 |
| NDP52 | Sigma-Aldrich | Cat# HPA023195, RRID:AB_1845916 |
| Vinculin | Cell Signaling | Cat# 4650, RRID:AB_10559207 |
| SLC1A5/ASCT2 | Merck | #abn73, RRID:AB_10807715 |
| SLC1A5/ASCT2 (V501) | Cell Signaling | #5345, RRID:AB_10621427 |
| **Oligonucleotides and other sequence-based reagents** | | |
| ALFA-TXNIP | This study | Appendix Table S4 |
| attB-TXNIP | This study | Appendix Table S4 |
| SLC7A5-ALFA | This study | Appendix Table S4 |
| Gateway | This study | Appendix Table S4 |
| TXNIP$^{S308A}$ | This study | Appendix Table S4 |
| TXNIP$^{S308D}$ | This study | Appendix Table S4 |
| TXNIP$^{C247S}$ | This study | Appendix Table S4 |
| TXNIPp.Ile215TyrfsTer59 | This study | Appendix Table S4 |
| TXNIP$^{PPxY331AAxA}$ | This study | Appendix Table S4 |
| TXNIP$^{PPxY375AAxA}$ | This study | Appendix Table S4 |
| TXNIPp.Gln58HisGly59* | This study | Appendix Table S4 |
| mCherry | This study | Appendix Table S4 |
| GFP-TXNIP | This study | Appendix Table S4 |
| ALFA-TAG | This study | Appendix Table S4 |
| RPLP0 | Krzystek-Korpacka et al, 2016 | Appendix Table S5 |
| SLC7A5 | Kaira et al, 2013 | Appendix Table S5 |
| ARRDC1 | PrimerBank ID (65288282c1) | Appendix Table S5 |
| ARRDC2 | IDT | Appendix Table S5 |
| ARRDC3 | Krzystek-Korpacka et al, 2016 | Appendix Table S5 |
| ARRDC4 | Kaira et al, 2013 | Appendix Table S5 |
| TXNIP | PrimerBank ID (65288282c1) | Appendix Table S5 |
| **Chemicals, enzymes and other reagents** | | |
| Blasticidin | InvivoGen | Cat. #ant-bl-5b |
| High glucose Dulbecco's modified Eagle's medium (DMEM) | Sigma-Aldrich | Cat. #D6429 |
| RPMI 1640 Medium, GlutaMAX Supplement | Thermo Fisher Scientific | Cat. #61870044 |
| OptiMEM | Thermo Fisher Scientific | Cat. #31985062 |
| FBS | Sigma-Aldrich | Cat. #S0615 |
| Dialysed FBS | Thermo Fisher Scientific | Cat. #26400044 |
| Trypsin-EDTA solution | Sigma-Aldrich | Cat. #T4174 |
| DMSO | Sigma-Aldrich | Cat. #D2650 |
| MK2206 | Enzo Life Sciences | Cat. #ENZ-CHM164-0005 |
| PD0325901 | Absoure | Cat. #S1036 |
| JPH203 | Selleckchem | Cat. #S8667 |
| chloroquine | Sigma-Aldrich | Cat. #C6628 |
| dynasore hydrate | Sigma-Aldrich | Cat. #D7693 |
| penicillin/streptomycin | Sigma-Aldrich | Cat. #P0781 |
| WesternBright Chemiluminescence substrate solution (Advansta) | Biozym | Cat. #541005X |
| Lipofectamin LTX reagent | Thermo Fisher Scientific | Cat. #15338-100 |
| polyethylenimine (PEI) | Polyscience | Cat. #23966-100 |
| Polybrene | Sigma-Aldrich | Cat. #107689 |
| DAPI | Sigma-Aldrich | Cat. #D9542 |
| Mowiol | Sigma-Aldrich | Cat. #81381 |
| Saponin | Sigma-Aldrich | Cat. #84510 |
| Triton X-100 | Thermo Fisher Scientific | Cat. #28314 |
| RNAse A | Thermo Fisher Scientific | Cat. #EN0531 |
| [$^{14}C_5$]-L-glutamine | Hartmann Analytic | Cat. #MC1124 |
| [4,5-$^3$H]-L-Leucine | Hartmann Analytic | Cat. #MT672E |
| [$^{14}C_5$]-L-Isoleucine | Hartmann Analytic | Cat. #MC174 |
| [$^{35}$S]-L-methionine [$^{35}$S]-cysteine | Hartmann Analytic | Cat. #21635410 |
| LSC-Universal cocktail | Roth | Cat. #0016.3 |
| L-Glutamine | Gibco | Cat. #25030-024 |
| glutamine-free DMEM | Sigma-Aldrich | Cat. #D6546 |
| DMEM without L-methionine, L-cysteine and L-glutamine | Sigma-Aldrich | Car #D0422 |
| AA free DMEM | USBiological | Cat. #D9811-01 |
| 2-NBDG | AAT Bioquest | Cat. #36702 |
| phloretin | Sigma-Aldrich | Cat. #P7912 |
| Stable isotope-labeled canonical AA mix composition | Cambridge Isotope Laboratory | #MSK-CAA-1 |
| Cycloheximide | Sigma-Aldrich | Cat. #C7698 |
| Anti-ALFA selector ST magnetic beads | NanoTag Biotechnologies | Cat. #N1516 |
| Phusion™ High-Fidelity DNA Polymerase | NEB | M0530S |
| T7 DNA Ligase | NEB | M0318S |
| Micro BCA Protein Assay Kit | Thermo Fisher Scientific | Cat. #23235 |
| RNeasy Mini Kit | Quiagen | Cat. #74104 |
| LunaScript RT Super Mix Kit | NEB | Cat. # E3010 |
| Biozym Blue S'Green qPCR Kit | Biozym | Cat. # 331416S |
| XF DMEM Medium, pH 7.4 | Agilent Technologies | Cat. #103575-100 |
| 1.0 M Glucose Solution | Agilent Technologies | Cat. #103577-100 |
| 100 mM Pyruvate Solution | Agilent Technologies | Cat. #103578-100 |
| 200 mM Glutamine Solution | Agilent Technologies | Cat. #103579-100 |
| Cell Mito Stress Test Kit | Agilent Technologies | Cat. #103010-100 |
| PDL Poly-D-Lysine (0.1 mg/ml) | Gibco | Cat. #A38904-01 |
| XF Calibrant | Agilent Technologies | Cat. #103059-000 |
| Hoechst 33342 | Thermo Fisher Scientific | Cat. #62249 |
| Phalloidin 488 | Thermo Fisher Scientific | Cat. #A12379 |
| **Software** | | |

| Reagent/resource | Reference or source | Identifier or catalog number |
|---|---|---|
| ImageJ | | Version 1.53t |
| Affinity Designer | Serfi | Version 1.10.5 |
| GraphPad Prism 9 | | Version 9.4.1 |
| Benchling | https://benchling.com/ | |
| Flow Jo | BD Life Sciences | Software 10.8.1 |
| ModFit LT | Verity Software House | |
| Primer3 | https://primer3.ut.ee | Version 4.1.0 |
| PrimerBank | https://pga.mgh.harvard.edu/primerbank/ | |
| CHOPCHOP | https://chopchop.cbu.uib.no | |
| ZEISS ZEN Digital Imaging for Light Microscopy | ZEISS | Version 3.5 |
| PrimerQuestTM Tool | Integrated DNA Technologies, IDT | |
| ICE Analysis online tool | Synthego, USA | Synthego, v2.0 |
| IncuCyte S3 software | Sartorius | Version 2022B |
| Wave Pro software | Agilent Technologies | Version 10.2.1.4 |
| OLYMPUS cellSens Dimension | Olympus | Version 4.4 |
| Proteome Discoverer | Thermo Fischer Scientific | Version 3.1 |
| **Other** | | |
| CASY Cell Counter and Analyzer | OMNI Life Science | |
| IncuCyte S3 Live Cell Imaging System | Sartorius | |
| Fusion FX | Vilber Lourmat | |
| BD FACS Aria III Cell Sorter | BD Biosciences | |
| LSM980 AiryScan 2 | ZEISS | |
| Attune NxT Flow Cytometer | Life Technologies | |
| AccuFlex LSC-800 | HITACHI | |
| Typhoon FLA 9500 detection system | | |
| Olympus Microscope IX83 P2ZF | Olympus, Evident Scientific | |
| Seahorse XFp FluxPak (cartridge and plate) | Agilent Technologies | Cat. #103022-100 |

## Ethical statement

All patients' data were extracted from the medical routine records. Written informed consent for molecular genetic studies and publication of data and images was obtained from the legal guardians of the patient. This approach was approved by the ethics committee of the Medical University of Innsbruck (UN4501-MUI). The study was conducted in accordance with the principles of the Declaration of Helsinki.

## Cell culture

Patient-derived primary fibroblasts were cultured from skin biopsies. Control fibroblasts were obtained from a male, born in 2000 (WT#1) and from a male, born in 2015 (WT#2). RPE1, HEK293T, A549, H460, HCC4006, primary human fibroblasts and MEF (mouse embryonic fibroblasts) (Teis et al, 2006) were cultured in DMEM containing high glucose (4.5 g/L) (Sigma-Aldrich, #D6429), supplemented with 1% (vol/vol) penicillin/streptomycin (Sigma-Aldrich, #P0781) and 10% (vol/vol) FBS (fetal bovine serum) (Sigma-Aldrich, #S0615) at 37 °C, in 5% $CO_2$ and 98% humidity. H2228, H226, H520 and H1299 cells were cultured in RPMI 1640, glutamax supplement medium (Thermo Fisher, Gibco, # 61870044) supplemented with 1% (vol/vol) penicillin/streptomycin (Sigma-Aldrich, #P0781) and 10% (vol/vol) FBS (Sigma-Aldrich, #S0615) under the same culture conditions. Cells were passaged before reaching confluency by incubation in 1× PBS (phosphate-buffered saline, 8 g/L NaCl, 0.2 g/L KCl, 1.15 g/L $Na_2HPO_4$, 0.2 g/L $KH_2PO_4$) and 1× trypsin-EDTA solution (Sigma-Aldrich, #T4174). Cells were resuspended in growth medium and re-seeded at a 1:5 to 1:10 ratio. Cells were regularly tested negative for mycoplasma.

The AKT inhibitor MK2206 (Enzo Life Sciences, #ENZ-CHM164-0005), dissolved in DMSO (Sigma-Aldrich, #D2650), was used at a final concentration of 1 μM. The MEK inhibitor PD0325901 (Absource, #S1036), dissolved in DMSO, was used at a final concentration of 5 μM. The SLC7A5 inhibitor JPH203 (also KYT-0353, Selleckchem, #S8667), dissolved in DMSO, was used at a final concentration of 10 μM. Chloroquine (Sigma-Aldrich, #C6628) was dissolved in Millipore $H_2O$ and used at a final concentration of 12.5 μM. To inhibit dynamin-mediated endocytosis, dynasore hydrate (Sigma-Aldrich, #D7693) was used as previously described (Kirchhausen et al, 2008) at a final concentration of 20 μM.

To maintain cells in proliferation (control cells, defined as "+ serum"), they were cultured in DMEM containing 4.5 g/L glucose and all AAs, (Sigma-Aldrich, #D6429), supplemented with 1% (vol/vol) penicillin/streptomycin (Sigma-Aldrich, #P0781) and 10% (vol/vol) FBS (Sigma-Aldrich, #S0615). For all experiments cell confluency never exceeded 80% to avoid contact inhibition. To induce quiescence, cells were washed with 1× PBS and cultured in DMEM containing 4.5 g/L glucose and all AAs, supplemented with 1% penicillin/streptomycin but without FBS for 24 h (referred to as serum starvation, "− serum").

## Proliferation assay

A similar cell number (>20.000 cells) was seeded onto six-well plates (day 0). Cells were harvested daily by trypsinization. The cell number was measured daily using a CASY Cell Counter and Analyzer (OMNI Life Science) for 4 days. The cell proliferation index was calculated as described before (Cormerais et al, 2018). The fold increase was calculated by normalizing to the cell number obtained 24 h after seeding (day 1).

For measuring cell proliferation in an automated manner, 50.000 cells were seeded onto a six-well plate and cultured at standard cell culture conditions. Cells were set on serum free medium 24 h after seeding and refed after 48 h of serum starvation. Cell proliferation was tracked using a Sartorius IncuCyte® S3 Live Cell Imaging System and confluency measurement was performed every 4 h for 5 days. Data was acquired and analyzed using the Sartorius IncuCyte® S3 software version 2022B.

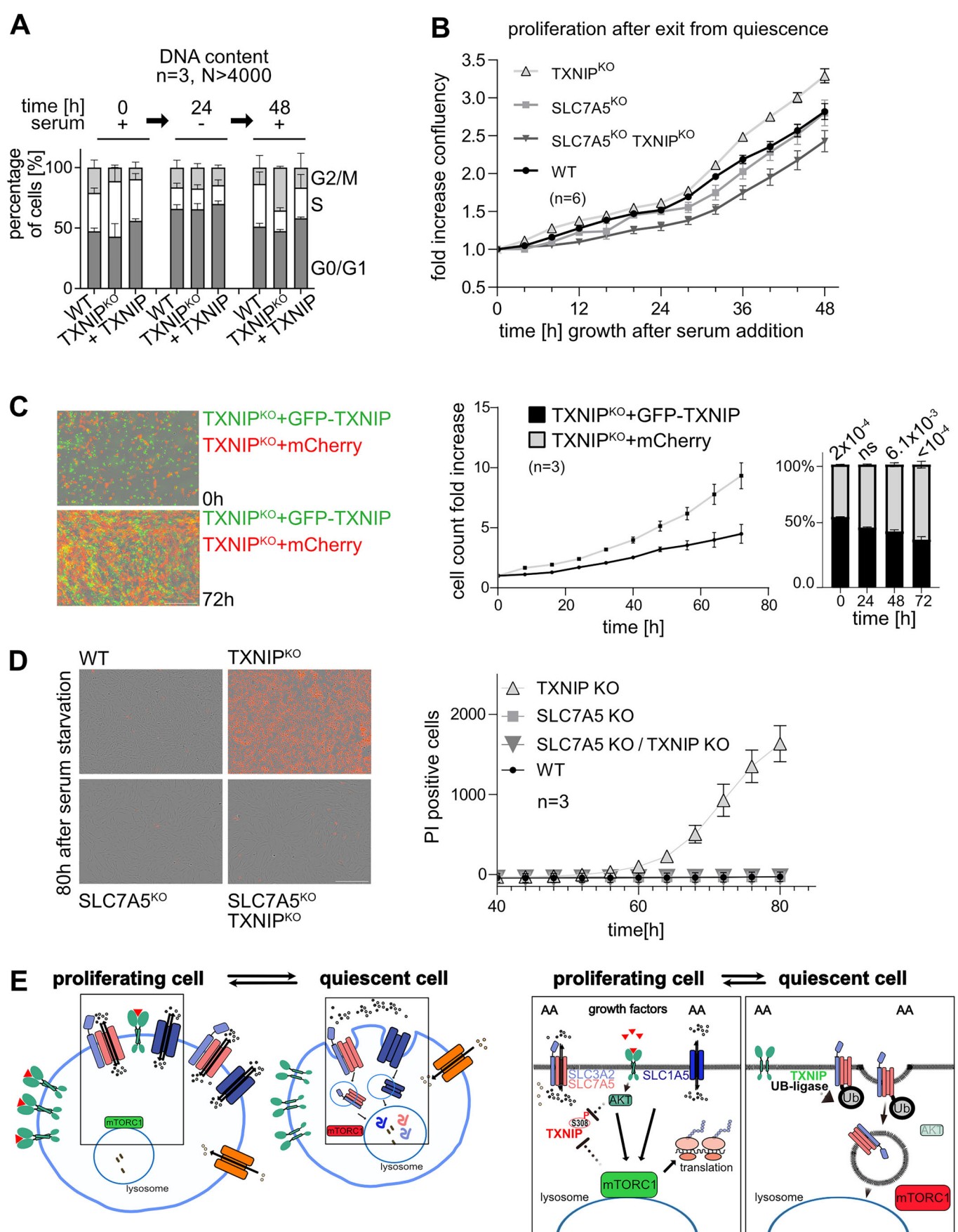

Figure 7.  TXNIP mediated degradation of SLC7A5 contributed to the regulation of cell proliferation.

(**A**) WT, TXNIP[KO] and TXNIP[KO] cells reconstituted with TXNIP were harvested, permeabilized and fixed. The DNA content was analysed by PI staining to assess the cell cycle profile by FACS ($n = 3$, $N > 4000$ cells, SD). (**B**) Increase in confluency of the indicated cells, after refeeding serum starved cells, was measured by continuous live cell microscopy using a Incucyte S3 system in an incubator for 48 h ($n = 6$, SEM). (**C**) Competition assay of TXNIP[KO] + mCherry and TXNIP[KO] + GFP-TXNIP. 100 000 TXNIP[KO] + mCherry and 100 000 TXNIP[KO] + GFP-TXNIP were co-seeded and the cell number was measured by continuous live cell microscopy in an Incucyte incubator for 72 h. Scale bar = 400 μm ($n = 3$, SEM). Quantification of TXNIP[KO] + mCherry and TXNIP[KO] + GFP-TXNIP cell number fold increase over time and quantification in % of total cells ($n = 3$, two-way ANOVA, Sidak's multiple comparisons test). (**D**) Analysis of cell death during quiescence of WT, TXNIP[KO] and SLC7A5[KO single] knock out cells, and SLC7A5[KO] TXNIP[KO] double knock out cells. 100,000 cells were seeded in growth medium ( + serum). After 24 h the medium was changed to starvation medium (-serum) containing propidium iodide (1 μg/ml). PI positive cells were measured by continuous live cell microscopy in an Incucyte incubator for 80 h. Scale bar = 400 μm. ($n = 3$, SEM). (**E**) Model of TXNIP mediated downregulation of SLC7A5 during entry into quiescence. Source data are available online for this figure.

## Cell competition assay

100.000 TXNIP[KO] expressing mCherry and TXNIP[KO] re-expressing GFP-TXNIP cells were mixed and co-seeded at standard cell culture conditions in a 6-well plate. Cell number increase was measured using a Sartorius IncuCyte® S3 Live Cell Imaging System for 72 h. Data were acquired and analyzed using the Sartorius IncuCyte® S3 software version 2022B.

## Cell death assay

In total, 100.000 WT cells, SLC7A5[KO], TXNIP[KO], SLC7A5[KO] TXNIP[KO], or TXNIP[KO] re-expressing GFP-TXNIP were seeded at standard cell culture conditions in a 24-well plate. After 24 h the medium was removed and the cells were washed 1x with PBS. Serum free medium (as described above) containing propidium iodide (1 μg/ml) was added. For SLC7A5 inhibition, cells were supplemented with 10 μM JPH203. The number of dead cells were measured by continuous live cell microscopy in a Sartorius IncuCyte® S3 Live Cell Imaging System for 80 h.

## Seahorse measurement

Mitochondrial functional analysis was performed using a Seahorse XF HS Mini Analyzer. For analysis, 5000 cells/well were seeded in a Seahorse XFp Cell Culture Miniplates, coated with 50 μl Poly-D-Lysine (0.1 mg/ml, Gibco A3890401). Prior to analysis, cells were washed and incubated with the Seahorse XF DMEM Medium, pH 7.4 (Agilent, 103575-100), containing Seahorse XF Glucose 10 mM (Agilent, 103577-100), Seahorse XF Pyruvate 1 mM (Agilent, 103578-100) and Seahorse XF L-Glutamine 2 mM (Agilent, 103579-100). A total of 180 μl of assay medium was present in all wells. Cells were incubated in a non-CO$_2$ 37 °C incubator for around 30 min. XFp sensor cartridges were hydrated overnight with the appropriate calibrant solution (Seahorse XF Calibrant Solution) and loaded with the reagents to perform MitoStress test. Briefly, respiration of intact cells was measured as follows: (1) ROUTINE/BASAL respiration corresponding to the physiological respiration, (2) LEAK respiration was determined by the addition of Oligomycin (final concentration 1 μM), (3) maximal respiratory capacity was determined by two additions of the protonophore Carbonyl cyanide 4-(trifluoromethoxy) phenylhydrazone (FCCP, 1 μM each step), (4) simultaneous injection of complex I inhibitor rotenone (final concentration 0.5 μM) and complex III inhibitor antimycin A (final concentration 0.5 μM) led to the assessment of the residual oxygen consumption. The oxygen consumption rate (OCR) was corrected for cell number. Briefly, after the analysis, plates were washed three times with PBS, fixed with 4%

PFA for 10 min at RT, washed three times with PBS and stored at 4 °C. Immunofluorescene analysis of the fixed cells was performed after 15 min of cell permeabilization with 0.1% Triton X-100 in PBS at room temperature, 1 h incubation with Phalloidin 488 (1:400) at room temperature, a quick washing step with PBS and an additional 10 min nuclei staining with Hoechst 33342 (1:5000). Seahorse plates were acquired using an Olympus Microscope (IX83 P2ZF) with a LUCPLFLN PH ×20 objective. Cells were then counted using the software OLYMPUS cellSens Dimension 4.4.

## Preparation of whole cell protein extracts, SDS-PAGE and western blot analysis

Cells were washed in 1x cold PBS, harvested using a cell scraper and pelleted at 13,000 rpm for 5 min at 4 °C. The cell pellets were resuspended in cold RIPA lysis buffer (50 mM Tris-HCl pH 7.5, 150 mM NaCl, 1% NP-40, 2 mM EDTA pH 8, 0.1% SDS, 0.5% sodium deoxycholate) or lysis buffer (50 mM Tris-HCl pH 7.5, 150 mM NaCl, 1% Triton X-100, 10% glycerol, 0.5 mM EDTA), supplemented with 50 mM NaF, 10 μg/mL leupeptin, 1 mM pefa-block, 1 μg/mL pepstatin, 10 μg/mL aprotinin, 2 mM Na$_3$VO$_4$. Cells were lysed for 30 min on ice, centrifuged at 13,000 rpm for 10 min at 4 °C and the cleared lysate was obtained. Protein concentration was measured using Micro BCA Protein Assay Kit (Thermo Fisher Scientific, #23235) according to the manufacturer's instructions. 20 μg protein was supplemented with 5x Urea sample buffer (8 M Urea, 5% SDS, 40 mM Tris pH 6.8, 0.1 mM EDTA, 0.4 mg/mL bromphenol blue, 1% final β-mercaptoethanol) at 1x final concentration. Lysates were separated by SDS-PAGE. Proteins were transferred onto polyvinylidene fluoride (PVDF) membranes (Amersham Hybond PVDF, #10600023 – 0.45 μm) using a wet transfer system (Bio-Rad) at constant 80 V for 2 h (1x transfer buffer: 20% methanol, 25 mM Tris-HCl, 192 mM glycine). Membranes were blocked in 5% milk in Tris-buffered saline buffer (TBS, 0.5 M NaCl, 20 mM Tris, pH 7.5) with 0.05% Tween 20 (TBS-T) and probed with the respective antibodies over night at 4 °C. The membranes were washed with TBS-T and probed with the respective horseradish peroxidase-linked secondary antibodies (1:5000 dilution) for 1 h at RT. After washing with TBS-T, the membrane was incubated with WesternBright Chemiluminescence substrate solution (Advansta) (Biozym, #541005X). The chemiluminescence signal was detected either on X-ray films or with a CCD-Camera (Fusion FX, Vilber Lourmat). The signal was quantified with ImageJ and normalized to GAPDH, or total protein content (stained with Coomassie, amido black or ponceau, as indicated). To quantify the phosphorylation of proteins, the signals were normalized to the unphosphorylated total protein levels.

## Denaturing SLC7A5-ALFA immunoprecipitation

Immunoprecipitation of SLC7A5-ALFA under denaturing lysis conditions was adopted from (Schmidt et al, 2019). Cells expressing SLC7A5-ALFA were either grown in growth medium (+ serum) or serum starved for 7 or 8 h (− serum). Cells were washed with 1x PBS and harvested. Cell pellets were resuspended in lysis buffer I (50 mM Tris pH 8, 150 mM NaCl, 50 mM NaF, 5 mM EDTA, 1% SDS, 10 nM NEM, 10 µg/mL leupeptin, 1 mM pefablock, 1 µg/mL pepstatin, 10 µg/mL aprotinin) and incubated 30 min at RT. Protein concentration was measured using Micro BCA Protein Assay Kit (Thermo Fisher Scientific, #23235) according to the manufacturer's instructions. Equal amounts of proteins were diluted by addition of 800 µl lysis buffer II (50 mM Tris pH 8, 150 mM NaCl, 50 mM NaF, 5 mM EDTA, 1% Triton X-100, 10 nM NEM, 10 µg/mL leupeptin, 1 mM pefablock, 1 µg/mL pepstatin, 10 µg/mL aprotinin). Samples were subjected to immunoprecipitation. Beads (ALFA Selector ST, NanoTag Biotechnologies, #N1516) were equilibrated in washing buffer I (50 mM Tris pH 8, 150 mM NaCl, 1% Triton X-100). Cell suspension and beads were incubated rotating at 4 °C for 3 h. Beads were recovered with a magnetic rack and washed twice with washing buffer I and three times with washing buffer II (50 mM Tris pH 8, 300 mM NaCl, 1% Triton X-100). Proteins were eluted with 5x SDS sample buffer (250 mM Tris-HCl pH 6.8, 10% SDS, 50% glycerol, 10% β-mercaptoethanol) and denatured for 10 min at 95 °C. For detection of ubiquitinated proteins, samples were separated by SDS-PAGE. Proteins were transferred onto PVDF membranes (Amersham Hybond PVDF, #10600023 – 0.45 µM (300 mm)) using a wet transfer system (Bio-Rad) at constant 80 V for 2 h. After transfer, the membrane was blocked with 10% BSA in 0.45% Tween 20 for at least 1 h and incubated with anti-ubiquitin antibody (Santa Cruz, #sc-8017) overnight in 10% BSA blocking solution.

## Non-denaturing ALFA-TXNIP immunoprecipitation

Non-denaturing immunoprecipitation of ALFA-TXNIP was adopted from (Wu et al, 2013). Briefly, cells were washed with 1x PBS and harvested by scraping. Cells were lysed in lysis buffer (30 mM Tris pH 7.5, 120 mM NaCl, 20 mM NaF, 0.5% octyl-β-D-glucopyranoside) (Sigma-Aldrich, #O8001) and protease inhibitors (10 µg/mL leupeptin, 1 mM pefablock, 1 µg/mL pepstatin, 10 µg/mL aprotinin, 2 mM $Na_3VO_4$). Cells were lysed for 30 min on ice, centrifuged at 13,000 rpm, at 4 °C for 10 min and cleared lysate was obtained. Protein concentration was measured using Micro BCA Protein Assay Kit (Thermo Fisher Scientific, #23235) according to the manufacturer's instructions. Equal amounts of proteins were subjected to immunoprecipitation. Beads (ALFA Selector ST, NanoTag Biotechnologies, #N1516) were equilibrated in lysis buffer. Cell suspension and beads were incubated rotating at 4 °C for 3 h. After incubation beads were washed three times for 10 min at 4 °C using wash buffer (30 mM Tris pH 7.5, 150 mM NaCl, 0.1% octyl-β-D-glucopyranosid). Proteins were eluted in 100 µl 2× Urea sample buffer at 42 °C for 30 min.

## Mass spectrometry analysis of ALFA-TXNIP immunoprecipitations

For label-free interactome analysis of ALFA-TXNIP and its PPxY variants, HEK293T cells (4x15cm dishes at 80% confluency per condition) were transfected with either ALFA-TXNIP, empty vector or the respective ALFA-TXNIP$^{PPXY331}$ or ALFA-TXNIP$^{PPXY}$

plasmid using PEI. Cells were harvested 24 h after transfection and were lysed in lysis buffer (30 mM Tris pH 7.5, 150 mM NaCl, 0,5% octyl-D-glucopyranosid) and the lysate was then incubated with 50 µl of 50% ALFA$^{ST}$-magnetic-bead slurry with head-over-tail rotation for 3 h at 4 °C. After incubation, the beads were collected using a magnetic rack and were then washed two times using wash buffer (30 mM Tris pH 7.5, 150 mM NaCl, 0.1% octyl-D-glucopyranosid), three times with 1 mL 100 mM ammonium bicarbonate (ABC) buffer pH 8, reduced by adding 50 µL 10 mM DTT in ABC buffer (30 min, 56 °C), digested with 0.5 µg trypsine (Promega, 6 h, 37 °C), and finally alkylated by adding 5 µL 550 mM Iodacetamide in ABC buffer (20 min, room temperature).

Samples were analyzed using an UltiMate 3000 nano-HPLC system coupled to a Orbitrap Eclipse mass spectrometer (Thermo Scientific, Bremen, Germany). The peptides were separated on a homemade fritless fused-silica microcapillary column (100 µm i.d. × 280 µm o.d. ×16 cm length) packed with 2,4 µm reversed-phase C18 material (Reprosil). Solvents for HPLC were 0.1% formic acid (solvent A) and 0.1% formic acid in 85% acetonitrile (solvent B). The gradient profile was as follows: 0–4 min, 4% B; 4–57 min, 4-35% B; 57–62 min, 35–100% B, and 62–67 min, 100% B. The flow rate was 300 nL/min.

The Orbitrap Eclipse mass spectrometer equipped with a field asymmetric ion mobility spectrometer (FAIMS) interface was operating in the data dependent mode with compensation voltages (CV) of -45, and -65 and a cycle time of one second. Survey full scan MS spectra were acquired from 375 to 1500 $m/z$ at a resolution of 60,000 with an isolation window of 1.2 mass-to-charge ratio ($m/z$), a maximum injection time (IT) of 50 ms, and automatic gain control (AGC) target 400,000. The MS2 spectra were measured in the Orbitrap analyzer at a resolution of 15,000 with a maximum IT of 22 ms, and AGC target or 50,000. The selected isotope patterns were fragmented by higher-energy collisional dissociation with normalized collision energy of 30%.

Data Analysis was performed using Proteome Discoverer 3.1 (Thermo Scientific) with search engine Sequest. The raw files were searched against the Uniprot human database. Precursor and fragment mass tolerance was set to 10 ppm and 0.02 Da, respectively, and up to two missed cleavages were allowed. Carbamidomethylation of cysteine was set as static modification, oxidation of methionine was set as variable modifications. Peptide identifications were filtered at 1% false discovery rate.

## Bead-immobilized prey assay (BIPA)

For the bead-immobilized prey assay (BIPA), ALFA-TXNIP or one of the two mutant forms ALFA-TXNIP$^{PPXY331}$ or ALFA-TXNIP$^{PPXY}$ (expressed in HEK293T) were first immobilized on ALFA$^{ST}$-beads by incubation of cell lysates with 50% ALFA$^{ST}$-bead-slurry with head-over-tail rotation for 3 h at 4 °C. The beads were collected by centrifugation at 1000×*g* for 60 s at 4 °C and were then washed two times using BIPA wash-buffer (100 mM HEPES pH = 7.5, 100 mM NaCl). The amount of bound ALFA-TXNIP was analyzed by SDS-PAGE and WB analysis. These beads were stored at −80 °C. In the meantime, YFP-Ubiquitin-ligase constructs were transfected into HEK293T. After 24 h the YFP-Ubiquitin-ligase expressing cells were harvested and lysed in 100 µl BIPA-Buffer (100 mM HEPES pH=7.5, 150 mM NaCl, 0.1% Triton-X100) with subsequent TCL-preparation as described. The thawed ALFA-TXNIP containing beads were

resuspended in 40 µl of BIPA wash-buffer and incubated with 10 µl of the YFP-Ubiquitin-ligase containing TCL for 30 min on ice, protected from light. After incubation, the beads were washed two times using BIPA wash-buffer, resuspended in 200 µl of BIPA wash-buffer and transferred to an eight-well microscopy glass slide. The recruitment of YFP-tagged HECT type ubiquitin ligases to ALFA-TXNIP beads was assessed using a LEICA DMi8 THUNDER Imager equipped with a Leica K8 camera. All pictures were taken using a 20x air objective with 100 ms of exposure time.

For quantification of the bead-assay, grey-values of all images were adjusted to a minimal grey-value of 80 and a maximal grey-value of 220, according to the brightest observed sample, using ImageJ. The fluorescence intensity of one individual bead was quantified by measuring four equidistant orthogonal line-scans across the bead-boundary. The raw-data was then processed in Microsoft Excel by subtracting the minimum grey-value from each line-scan (background correction), followed by averaging of the four line-scans of each bead. These bead averaged fluorescence intensity values are presented as individual dots in the corresponding graphs.

## Genetic modifications and cloning

Genetic modifications were performed by PCR using standard techniques or by using the Gateway cloning method (Thermo Fisher Scientific) according to manufacturer's instructions. All plasmids and primers used in this study are listed in Appendix Tables S3 and S4. The sequence-verified plasmids were then used for lentiviral transduction of RPE1 cells. To generate virus particles, HEK293T cells were seeded in a 6-well tissue culture plate and transfected with 2 µg lentiviral vector containing the gene of interest together with virus packing vectors (1 µg pVSV-G and 1 µg psPAX2) using polyethylenimine (PEI, Polyscience, #23966-100) as transfection reagent. In all, 4–8 h after transfection, the medium was changed. Virus particle-containing medium was collected from the HEK293T cells 48 h and 72 h after transfection and was used to infect RPE1 cells. Polybrene was added to the virus-particle-containing medium to help lentivirus integration (1:500 dilution, Sigma-Aldrich, #107689). After 10 days of selection with 1 µg/mL blasticidin (InvivoGen, #ant-bl-5b), the efficacy of the virus transduction was tested via SDS-PAGE and Western Blot analysis.

Genetic deletion using CRISPR/Cas9-mediated gene editing was performed as previously described (Ran et al, 2013). For CRISPR/Cas9-mediated gene editing, guide RNA (gRNA) targeting primers for SLC7A5 (5'-caccgCGTGAACTGCTACAGCGTGA-3' and 5'-aaacTCACGCTGTAGCAGTTCACGc-3') and TXNIP (5'-caccgG-TAAGTGTGGCGGGCCACAA-3' and 5'-aaacTTGTGGCCCGC-CACACTTACc-3'). RPE1 cells were transiently transfected with the gRNA-containing pSpCas9(BB)-2A-GFP plasmid (Addgene, #48138) using Lipofectamin LTX reagent (Thermo Fisher Scientific, #15338-100) according to the manufacturer's protocol. 24 h after transfection, GFP-positive, transfected cells were enriched by flow cytometry (BD FACSAria III Cell Sorter, BD Biosciences). The sorted cell suspension was used for the generation of single cell clones. Depletion efficiency was verified by PCR using specific primers and Western Blot analysis.

For the transient transfection of HEK293T Lipofectamine LTX & Plus reagent (Thermo Scientific, #15338100) was used according to the manufacturer's protocol. HEK293T cells were seeded in a 10 cm dish. At around 80% confluency, 16 µg of the respective plasmids were diluted in 500 µl of OptiMEM (Thermo Fisher Scientific, #31985062) in the presence of 48 µl of LTX- and 16 µl of Plus-reagent for the transfection. The transfected cells were incubated for 4 h before the transfection medium was discarded and replaced with 10 ml of fresh DMEM. After that, cells were incubated for 24 h under standard conditions.

## Immunofluorescence microscopy

Cells were grown on 12 mm glass coverslips (Hartenstein GmbH, #DKR1) and fixed with 4% paraformaldehyde (PFA) for 10–15 min at RT. Immunofluorescence staining of SLC3A2 and ALFA-tagged proteins was performed as described previously (Finicle et al, 2018). Briefly, cells were permeabilized and blocked in blocking buffer (10% FBS, 0.3% saponin (BioChemika, #84510), sodium azide (0.02% final) in 1× PBS) for 30 min at 37 °C. Cells were incubated with mouse anti-SLC3A2 (BD Pharmingen, #556074) or rabbit anti-ALFA (NanoTag Biotechnologies, #N1581) in a 1:100 dilution in blocking buffer overnight at 4 °C. For double-staining with LAMP1, cells were simultaneously incubated with rabbit anti-LAMP1 (Cell Signaling, #9091) in a 1:800 dilution in blocking buffer. Cells were washed once with wash solution (0.03% saponin, sodium azide (0.02% final) in 1x PBS), following three washing steps with PBS and incubated with Alexa Fluor 488 goat anti-mouse IgG (Invitrogen, #A11001) and Alexa Fluor 568 goat anti-rabbit IgG (Invitrogen, #A11011) in a 1:500 dilution in blocking buffer for 1 h at RT.

For immunofluorescence labeling of SLC1A5, cells were permeabilized in in 0.2% Triton X-100 (Thermo Fisher Scientific, #28314) in PBS for 5 min and blocked in blocking buffer (2% gelatin, 150 mM NaCl, 10 mM Pipes pH 6.8, 5 mM EGTA, 5 mM glucose, 5 mM MgCl$_2$, 50 mM NH$_4$Cl) for 1 h at RT. Cells were incubated with rabbit anti-SLC1A5 (Sigma-Aldrich, #HPA035240) in a 1:50 dilution in blocking buffer for 1 h at RT. For double-staining with LAMP1, cells were simultaneously incubated with mouse anti-LAMP1 (DSHB, #H4A3-S) in a 1:50 dilution in blocking buffer. Cells were washed with PBS and incubated with Alexa Fluor 488 goat anti-rabbit IgG (Invitrogen, #A11008) and Alexa Fluor 568 goat anti-mouse IgG (Invitrogen, #A11031) in a 1:500 dilution in blocking buffer for 1 h at RT. For immunofluorescence labeling of SLC2A1, cells were permeabilized and blocked in blocking buffer (2% gelatin, 150 mM NaCl, 10 mM Pipes pH 6.8, 5 mM EGTA, 5 mM glucose, 5 mM MgCl$_2$, 50 mM NH$_4$Cl) containing 0.025% saponin (Sigma-Aldrich, #84510) for 1 h at RT. Cells were incubated with rabbit anti-SLC2A1 (Merck, #07-1401) in a 1:200 dilution in blocking buffer for 1 h at RT. Cells were washed with PBS and incubated with Alexa Fluor 488 goat anti-rabbit IgG (Invitrogen, #A11008) in a 1:500 dilution in blocking buffer for 1 h at RT. Cells were washed with 1× PBS, incubated with DAPI (Sigma-Aldrich, #D9542) in a 1:20,000 dilution in 1× PBS and mounted in Mowiol (Sigma-Aldrich, #81381).

Confocal microscopy was performed on a LSM980 AiryScan 2 (ZEISS). Confocal Z-series stack (step size 0.22 µm) images were acquired using the AiryScan detector, a LD LCI Plan-Apochromat 63×/1.2 with glycerol immersion objective (ZEISS) and the 405 nm, 488 nm and 561 nm laser lines. To compare signals of a protein of interest in proliferating, quiescent or treated cells, laser settings were kept constant while imaging both conditions. The ZEISS Zen 3.5 software was used as recording software and for processing of

the raw images (AiryScan processing). Brightness and contrast were adjusted using ImageJ software. For merged images, the levels of red, green and blue channels were separately adjusted.

For quantification of the mean fluorescence intensity at the PM, single z-stack planes were analyzed in ImageJ. To quantify the mean fluorescence intensity at the PM, the region of interest was selected at five different positions along the membrane of a single cell.

## Cell surface staining of SLC3A2 and FACS analysis

To measure the difference in surface staining of SLC3A2 between proliferating and quiescent cells, cells were either grown in growth medium or serum starved for 24 h before the experiment. The staining was performed as previously described (Finicle et al, 2018). Cells were washed with PBS, trypsinized and resuspended in FACS buffer (10% FBS in 1× PBS) containing anti-SLC3A2 antibody (BD Pharmingen, #556074) at a dilution of 1:100. The antibody was incubated for 30 min on ice. Cells were centrifuged at 1000 rpm for 5 min at 4 °C and resuspended in FACS buffer containing Alexa Fluor 488 goat anti-mouse IgG (Invitrogen, #A11001) at a dilution of 1:500. The antibody was incubated for 30 min on ice in the dark. Cells were centrifuged at 1000 rpm for 5 min at 4 °C and resuspended in 200 µl FACS buffer. The fluorescence intensity was measured by flow cytometry (Attune NxT Flow Cytometer, Life Technologies). For every experiment, more than 3500 cells were analyzed. Flow cytometry analysis was performed with FlowJo software 10.8.1 (BD Life Sciences). Cells were gated by forward versus side scatter (FSC-A vs. SSC-A) to identify cell population, forward scatter width versus forward scatter area (FSC-W vs FSC-A) for doublet exclusion and fluorescence intensity (BL1-A) vs cell count to reflect Alexa-488 signal populations. Unstained cells were excluded from analysis.

## DNA content for cell cycle analysis of fixed cells stained with propidium iodide (PI)

Cells were washed with PBS and harvested by trypsinization. Cells were centrifuged at 1000 rpm for 5 min at 4 °C. The pellet was washed in cold PBS. Cells were fixed by resuspending the cells in 1 ml ice cold 70% vol/vol ethanol in PBS. Samples were stored at -20 °C for at least 30 min. The ethanol was washed away with ice cold PBS twice. Cells were then incubated with propidium iodide (PI) at a final dilution of 1:25 and RNAse A (Thermo Fisher Scientific, #EN0531) at a final dilution of 1:100 in PBS, for 30 min at 37 °C. The fluorescence intensity was measured by flow cytometry (Attune NxT Flow Cytometer, Life Technologies). For every experiment, more than 1500 cells were analyzed. Data were analyzed using ModFit LT (Verity Software House). Cells were gated by forward versus side scatter (FSC-A vs. SSC-A) to identify cell population. PI-positive cells were gated for doublet exclusion by width vs height (BL3-H vs BL3-W) and fluorescence intensity (BL3-A) vs cell count to reflect PI-positive signal populations.

## RT-PCR

RNA was extracted from cells using the RNeasy Mini Kit (Qiagen, #74104) according to the manufacturer's instructions. 800 ng RNA were reverse transcribed into cDNA using the LunaScript RT Super Mix Kit (NEB, #E3010) according to manufacturer's protocol. For mRNA fold regulation analysis, quantitative real-time PCR was performed using gene specific primers listed in Appendix Table S5. The Biozym Blue S'Green qPCR Kit (Biozym, #331416S) was used according to manufacturer's protocol. All primers were synthesized by Sigma-Aldrich and diluted in Millipore $H_2O$ to a concentration of 10 µM. Each condition was performed in technical triplicates from three biological replicates. Fold regulation was calculated using the $2^{-\Delta\Delta Ct}$ method and normalized to the housekeeping gene *ribosomal protein lateral stalk subunit P0 (RPLP0)*.

## Uptake of radioactive labeled AAs

To analyze the uptake of L-glutamine, cells were washed with PBS and incubated with 0.3 µCi/mL [$^{14}C_5$]-L-glutamine (Hartmann Analytic, #MC1124) in glutamine-free medium (Sigma-Aldrich, #D6546). To analyze the uptake of L-leucine, cells were washed with PBS and incubated with 0.5 µCi/mL [4,5-$^3$H]-L-Leucine (Hartmann Analytic, #MT672E) in AA free medium (USBiological, #D9811-01). The uptake of L-isoleucine was initiated by incubating the cells with 0.005 µCi/mL [$^{14}C_5$]-L-isoleucine (Hartmann Analytic; #MC174) in AA free medium. The uptake was performed for 15 min at 37 °C. To stop the reaction, cells were washed twice with ice cold PBS. Cells were harvested by trypsinization and the cell pellet was lysed in 100 µl lysis buffer (50 mM Tris-HCl pH 7.5, 150 mM NaCl, 1% NP-40, 2 mM EDTA pH 8, 0.1% SDS, 0.5% sodium deoxycholate) for 30 min on ice. Cells debris were removed by centrifugation (13,000 rpm, 4 °C, 10 min) and the supernatant was analyzed by liquid scintillation counting (LSC-Universal cocktail, Roth, #0016.3; HITACHI AccuFlex LSC-800). AA uptake was calculated as counts per min (cpm) normalized to the protein concentration (µg) of each cleared lysate. Protein concentration was determined by using Micro BCA Protein Assay Kit (Thermo Fisher Scientific, #23235) according to the manufacturer's instructions.

## Glucose uptake assay

The glucose uptake assay was performed as described previously (Zou et al, 2005). Briefly, glucose uptake was initiated by adding 2-NBDG (2-(N-(7-Nitrobenz-2-oxa-1,3-diazol-4-yl)amino)-2-deoxyglucose) (AAT Bioquest, #36702, dissolved in DMSO) at a final concentration of 100 µM in the respective medium. To inhibit glucose uptake, cells were simultaneously treated with 1 mM phloretin (Sigma-Aldrich, #P7912, dissolved in 50% ethanol) for 45 min. After 45 min incubation at 37 °C in the dark, cells were washed with ice cold PBS to stop the reaction, trypsinized and resuspended in PBS. Cells were centrifuged at 1000 rpm for 5 min at 4 °C and the pellet was resuspended in 200 µl PBS and 10% FBS. The fluorescence intensity was measured by flow cytometry (Attune NxT Flow Cytometer, Life Technologies). For every experiment, more than 5000 cells were analyzed. Flow cytometry analysis was performed with FlowJo software 10.8.1 (BD Life Sciences). Cells were gated by forward versus side scatter (FSC-A vs. SSC-A) to identify cell population, forward scatter width versus forward scatter area (FSC-W vs FSC-A) for doublet exclusion and fluorescence intensity (BL1-A) vs cell count to reflect the uptake of 2-NBDG.

## Incorporation of ³⁵S labeled L-cysteine and L-methionine

In all, 150,000 cells were seeded and after two days, these cells were left untreated or serum starved for 24 h or serum starved and then refeed for 24 h. Prior to radiolabeling the cells, the serum starved cells were washed with PBS and incubated for 30 min in L-cysteine, L-methionine, L-glutamine free medium (Sigma-Aldrich #D0422), supplemented with 2 mM L-glutamine (Gibco #25030-024) and 1% (vol/vol) penicillin/streptomycin. The proliferating cells were incubated for 30 min in L-cysteine, L-methionine, L-glutamine free medium, supplemented with 2 mM L-glutamine, 10% dialysed FBS (Thermo Fisher Scientific #26400044) and 1% (vol/vol) penicillin/ streptomycin. Cells were washed with PBS, brought into suspension using 1× trypsin-EDTA solution and resuspended in the corresponding L-cysteine, L-methionine free medium. 17.6 µCi/mL [³⁵S]-methionine [³⁵S]-cysteine (Hartmann Analytic #21635410) was added to the cell suspension for 2 min and quenched with 10 µg/ml cycloheximide. Cell pellets were lysed for 30 min on ice, with RIPA lysis buffer, supplemented with protease inhibitors and 10 µg/ml cycloheximide. Equal amounts of protein were subjected to SDS-PAGE, (de)stained with Coomassie and dried for 1 h at 70 °C. The dried gel was exposed to a phosphor-imager for 48 h. Detection of a [³⁵S]-methionine [³⁵S]-cysteine radiogram was performed using a Typhoon FLA 9500 detection system.

## Mass spectrometry analysis of intracellular free AAs

The analysis of intracellular AAs by mass spectrometry was performed as described previously (van Pijkeren et al, 2023) (Kipura et al, 2024). Briefly, $10^6$ cells were seeded and serum starved for 24 h prior the extraction. Then, cells were washed three times with ice cold 1× PBS and extracted by adding 500 µl ice cold methanol (99,9% vol/vol, Fisher Chemicals) and 500 µl cold Millipore $H_2O$, containing the internal standard (1:100), a stable isotope-labeled canonical AA mix composition (Cambridge Isotope Laboratory, #MSK-CAA-1). Cells were scraped and 500 µl ice cold chloroform (99,9% vol/vol, Sigma-Aldrich, #650498) was added to the cell extracts. Cells were agitated in a tube shaker for 20 min at 1400 rpm at 4 °C, followed by centrifugation for 5 min at 16,100×g at 4 °C. The upper, polar phase was separated, evaporated by using a SpeedVac (Savant SPD111V SpeedVac Concentrator, Thermo Fisher Scientific) and stored at −80 °C. The interphase was used to determine the protein concentration. Therefore, the interphase was evaporated and the pellet was resuspended in denaturing buffer (1.6 M Urea, 100 mM AmBiCa, pH 8.2) and sonicated for 10 s to break up protein aggregates. Protein determination was done by using Micro BCA Protein Assay Kit (Thermo Fisher Scientific, #23235) according to the manufacturer's instructions. For absolute quantification, AAs were analyzed by targeted LC-MS using mixed-mode chromatography (RP-AEX, reversed phase–anion exchange chromatography) and ultra-high sensitive Orbitrap technology (Orbitrap Q Exactive HF-X and Orbitrap Fusion Lumos, both Thermo Fisher Scientific), operating in parallel reaction monitoring (PRM) mode. Bioinformatics data processing and statistical analysis was performed using TraceFinder (Thermo Fisher Scientific) and in-house R and Python scripts.

## Statistical analysis and software

Software used in this study, if not already specified, were Affinity Designer Version 1.10.5 (Serfi, RRID:SCR_016952), ImageJ Version 1.53t (RRID:SCR_003070), GraphPad Prism 9 Version 9.4.1 (458) (RRID:SCR_002798), Benchling (RRID:SCR_013955), FlowJo Version 10.8.1 (RRID:SCR_008520), ModFit LT (RRRID:SCR_016106), Primer3 Version 4.1.0 (RRID:SCR_003139), PrimerBank (RRID:SCR_006898), CHOPCHOP (RRID:SCR_015723), ZEISS ZEN Digital Imaging for Light Microscopy Version 3.5 (RRID:SCR_013672), and online tools TIDE (tracking of indels by Decomposition) (Brinkman et al, 2014), ICE Analysis online tool (Synthego, v2.0; Synthego, USA) and PrimerQuest™ Tool (Integrated DNA Technologies, IDT, USA)

Results were represented as mean with standard deviation (SD) analyzed by GraphPad Prism. Statistical significance between different biological samples was determined by either paired Student's $t$ test or one-way/two-way ANOVA with Tukey test for multiple comparisons, if not described otherwise.

# Data availability

The datasets produced in this study are available in the following databases: The mass spectrometry proteomics data have been deposited to the ProteomeXchange Consortium via the PRIDE (Perez-Riverol et al, 2025) partner repository (https:// www.ebi.ac.uk/pride) with the dataset identifier PXD066527. The confocal imaging data have been deposited to figshare and are accessible here: https://figshare.com/s/8a55f66a3a17781bcdfd.

The source data of this paper are collected in the following database record: biostudies:S-SCDT-10_1038-S44318-025-00608-9.

# Peer review information

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

## Acknowledgements

We thank the patient and his family. We are grateful to Hemmo Meyer and Simona Polo for providing the YFP-tagged HECT-type ubiquitin ligases and to our protein core facility for excellent support. This research was funded in part by the Austrian Science Fund (FWF) (10.55776/P35874, 10.55776/P34907 to DT, 10.55776/P35832, 10.55776/P36600 to HF, 10.55776/P36925 to VR, 10.55776/P30196 to SH, 10.55776/FG20 to HF, BS, DT, LAH, KT and MA, 10.55776/DOC82 to DT, SK, LAH). JK is a recipient of a DOC Fellowship of the Austrian Academy of Sciences. KT acknowledges support from the DFG (German Research Foundation, project No TH 1358/3-2), the MESI-STRAT project (grant agreement No 754688) which has received funding from the European Union's Horizon 2020 research and innovation programme, and from the European Union European Research Council (ERC AdG BEYOND STRESS, grant agreement No 101054429) which has received funding from the European Union's Horizon Europe research and innovation programme. Views & opinions are those of the authors. For open access purposes, the author has applied a CC BY public copyright license to any author accepted manuscript version arising from this submission.

## Author contributions

**Jennifer Kahlhofer**: Data curation; Software; Formal analysis; Funding acquisition; Validation; Investigation; Methodology; Writing—original draft;

Writing—review and editing. **Nikolas Marchet**: Data curation; Formal analysis; Supervision; Validation; Investigation; Visualization; Methodology; Writing—review and editing. **Kristian Zubak**: Data curation; Formal analysis; Validation; Investigation; Methodology. **Brigitta Seifert**: Data curation; Investigation; Methodology. **Madlen Hotze**: Data curation; Formal analysis; Investigation; Methodology. **Anna-Sophia Egger-Hörschinger**: Investigation; Methodology. **Lucija Kucej**: Data curation; Formal analysis; Investigation; Methodology. **Claudia Manzl**: Data curation; Formal analysis; Investigation; Methodology. **Yannick Weyer**: Data curation; Formal analysis; Validation; Investigation; Methodology. **Sabine Weys**: Data curation; Formal analysis; Investigation; Methodology. **Martin Offterdinger**: Data curation; Investigation; Methodology. **Sebastian Herzog**: Methodology. **Veronika Reiterer**: Investigation; Methodology. **Chiara Volani**: Investigation; Methodology. **Marcel Kwiatkowski**: Data curation; Validation; Investigation; Methodology. **Saskia B Wortmann**: Investigation; Methodology. **Siamak Nemati**: Investigation. **Johannes A Mayr**: Investigation. **Johannes Zschocke**: Formal analysis; Investigation. **Bernhard Radlinger**: Data curation; Methodology. **Kathrin Thedieck**: Data curation; Investigation; Methodology. **Leopold Kremser**: Data curation; Validation; Investigation; Methodology. **Bettina Sarg**: Data curation; Formal analysis; Investigation; Methodology. **Lukas A Huber**: Investigation; Methodology. **Hesso Farhan**: Investigation. **Mariana EG de Araujo**: Investigation; Methodology. **Susanne Kaser**: Investigation; Methodology. **Sabine Scholl-Burgi**: Investigation; Methodology. **Daniela Karall**: Data curation; Validation; Investigation; Methodology; Writing—original draft; Writing—review and editing. **David Teis**: Conceptualization; Supervision; Funding acquisition; Investigation; Writing—original draft; Project administration; Writing—review and editing.

Source data underlying figure panels in this paper may have individual authorship assigned. Where available, figure panel/source data authorship is listed in the following database record: biostudies:S-SCDT-10_1038-S44318-025-00608-9.

## Disclosure and competing interests statement
The authors declare no competing interests.

# Expanded View Figures

**Figure EV1.   Cells entering quiescence selectively downregulate amino acid transporters.**

(**A**) WT cells were grown in growth medium (+ serum) or serum starved for 2 h, 4 h, 6 h and 24 h (− serum). Total cell lysates were analyzed by SDS-PAGE and WB with the indicated antibodies. WB quantification of SLC7A5, TAX1BP1 normalized to Vinculin and NBR1, NDP52, LC3 normalized to Ponceau (*n* = 3, SEM, paired *t* test). (**B, E**) Indirect IF of PFA fixed cells stained for SLC1A5 (yellow) LAMP1 (magenta) and DAPI (cyan) were analyzed by confocal microscopy. The merged images show a single plane of a Z-stack. Incubation with 12,5 µM chloroquine (CQ) in absence of serum for 14 h (− serum, + CQ). Scale bar=10 µm) (**C**) Quantification of SLC3A2 IF signal at the plasma membrane under serum-supplemented (+ serum) and starved (-serum) conditions with chloroquine (+ CQ) or without it. Box plots represent the median (centre line) and the interquartile range (25th to 75th percentile box). The whiskers show the minimum and maximum values. (*n* = 35, one way anova, Tukey's multiple comparison test, +serum –CQ vs –serum –CQ: *P* = 5.7E-14, +serum –CQ vs –serum +CQ: *P* = 8.5E-13). (**D**) WB quantification of SLC7A5 under serum-supplemented (+ serum) and starved (-serum) conditions starved (-serum) conditions with chloroquine (+ CQ) or without it. Box plots represent the median (centre line) and the interquartile range (25th to 75th percentile box). The whiskers show the minimum and maximum values. (*n* = 3, one way ANOVA, Tukey's multiple comparison test) (**E**) Incubation with 20 µM dynasore (dyna) in absence of serum for 14 h (− serum, + dyna). Scale bar=10 µm. (**F**) Quantification of SLC3A2 IF signal at the plasma membrane under serum-supplemented (+ serum) and starved (-serum) conditions with dynasore (+ dyna) or without it. Box plots represent the median (centre line) and the interquartile range (25th to 75th percentile box). The whiskers show the minimum and maximum values. (*n* = 50, one-way anova, Tukey's multiple comparison test, +serum –dyna vs –serum –dyna: *P* = 8.7E-14, +serum –dyna vs –serum +dyna: *P* = 7.6E-14). (**G**) WB quantification of SLC7A5 under serum-supplemented (+ serum) and starved (−serum) conditions in the presence (+ dyna) or absence (−dyna) of dynasore. Box plots represent the median (centre line) and the interquartile range (25th to 75th percentile box). The whiskers show the minimum and maximum values (*n* = 3, one-way ANOVA, Tukey's multiple comparison test). (**H**) Cells were incubated with [14 C]-glutamine for 15 min, washed and lysed. Total cell lysates were analyzed by scintillation counting. Cpm values were normalized to total protein content. Box plots represent the median (centre line) and the interquartile range (25th to 75th percentile box). The whiskers show the minimum and maximum values. (cpm/µg protein, *n* = 4, paired *t* test).

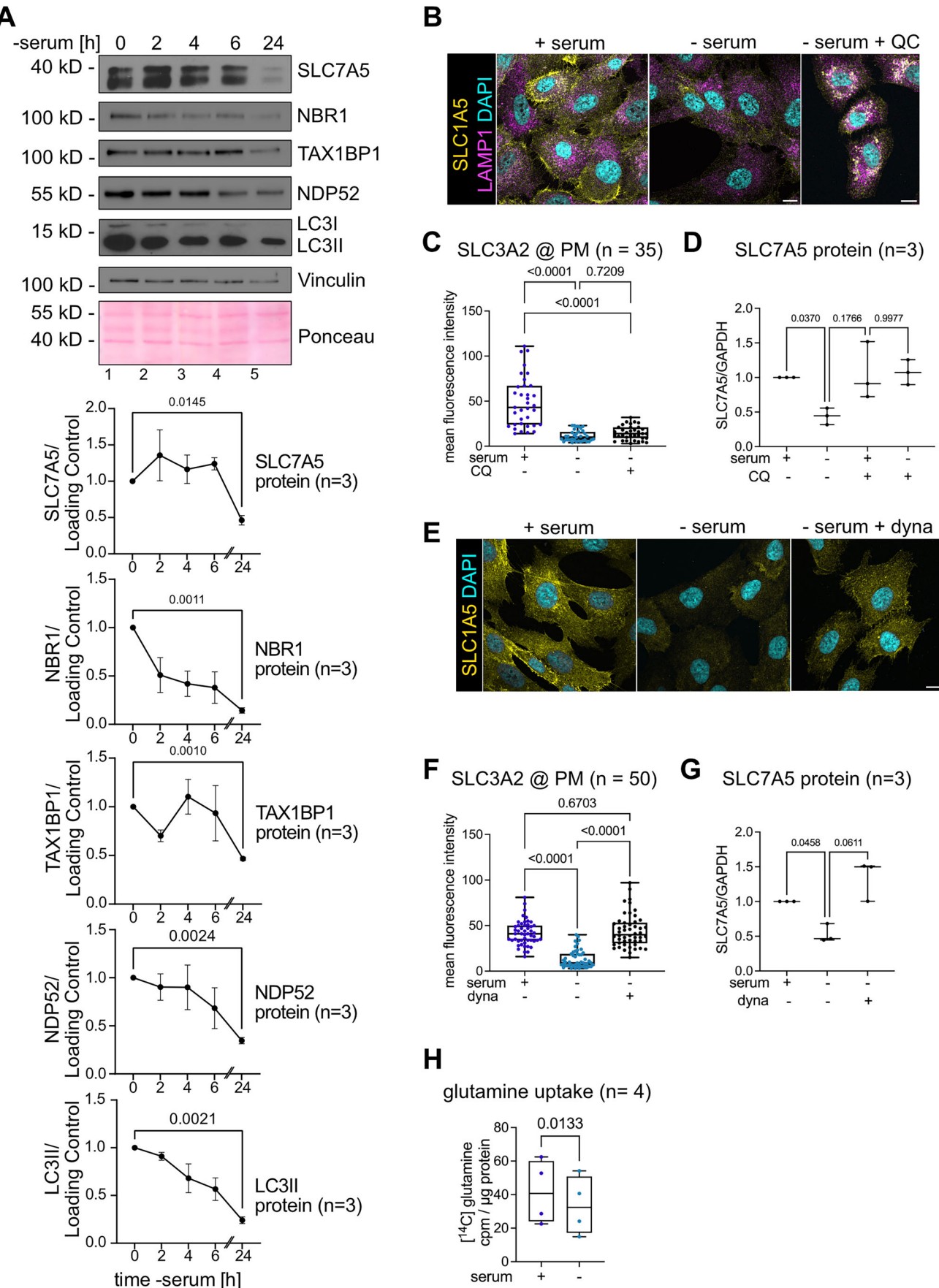

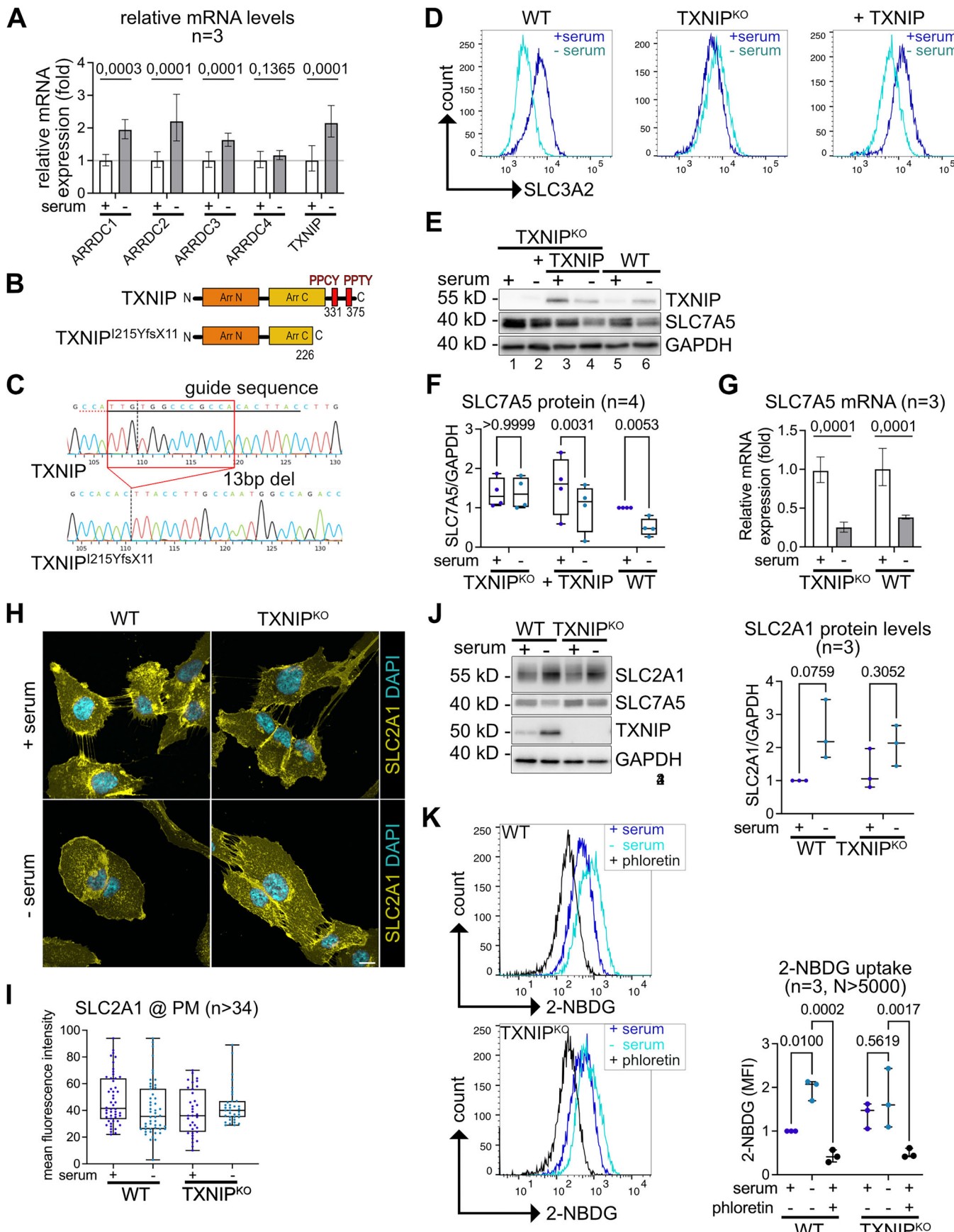

◀

**Figure EV2. TXNIP is required for the endocytosis of SLC7A5.**

qPCR analysis of (**A**) α-arrestins mRNA in growing (+ serum) or serum starved cells (− serum). Expression levels were normalized to the housekeeping gene *RPLP0* ($n = 3$, paired *t* test). (**B**) Schematic presentation of the mutation introduced into TXNIP by gene editing. (**C**) Representative chromatogram of Sanger sequencing from the genome amplicon of WT TXNIP and gene edited TXNIP cells. The guide RNA and the protospacer adjacent motif (PAM) sequence are indicated. The vertical lane shows the Cas9 editing site causing a 13 base pair deletion in *TXNIP* (red box). Image was generated by the ICE Analysis online tool (Synthego, v2.0; Synthego, USA). (**D**) Representative histograms of SLC3A2 cell surface FACS from WT, TXNIP[KO] and TXNIP[KO] reconstituted with TXNIP cells. (**E**) WT, TXNIP[KO] and TXNIP[KO] reconstituted with TXNIP were grown in growth medium (+ serum) or serum starved for 24 h (− serum). Total cell lysates were analysed by SDS-PAGE and WB with the indicated antibodies. (**F**) WB quantification of SLC7A5 protein levels, normalized to GAPDH. Box plots represent the median (centre line) and the interquartile range (25th to 75th percentile box). The whiskers show the minimum and maximum values. ($n = 4$, two-way ANOVA, Sidak's multiple comparisons test). (**G**) qPCR analysis of SLC7A5 mRNA in growing (+ serum) or serum starved cells (- serum). Expression levels were normalized to the housekeeping gene *RPLP0*. The bar charts show mean values ($n = 3$, paired *t* test). (**H**) Indirect of PFA fixed WT and TXNIP[KO] cells stained for SLC2A1 (yellow) and DAPI (cyan) was analysed by confocal microscopy. The merged images show a single plane of a Z-stack. Scale bar = 10 µm. (**I**) Quantification of SLC2A1 IF signal at the PM. Box plots represent the median (centre line) and the interquartile range (25th to 75th percentile box). The whiskers show the minimum and maximum values. ($n > 34$). (**J**) Total cell lysates were analysed by SDS-PAGE and WB with the indicated antibodies. For WB quantification, SLC2A1 protein levels were normalized to GAPDH. Box plots represent the median (centre line) and the interquartile range (25th to 75th percentile box). The whiskers show the minimum and maximum values. ($n = 3$, two-way ANOVA, Sidak's multiple comparisons test). (**K**) WT and TXNIP[KO] cells were assessed for glucose uptake by incorporation of 100 µM 2-NBDG for 45 min. Simultaneously, cells were treated with 1 mM phloretin. The fluorescence intensity was analysed by FACS. Box plots represent the median (centre line) and the interquartile range (25th to 75th percentile box). The whiskers show the minimum and maximum values ($n = 3$, $N > 5000$ cells, two-way ANOVA, Sidak's multiple comparisons test).

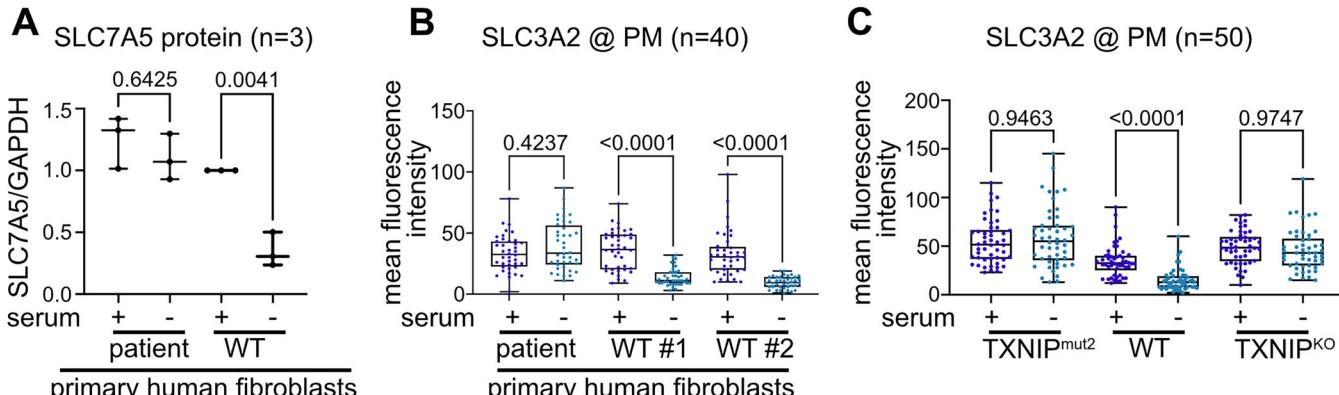

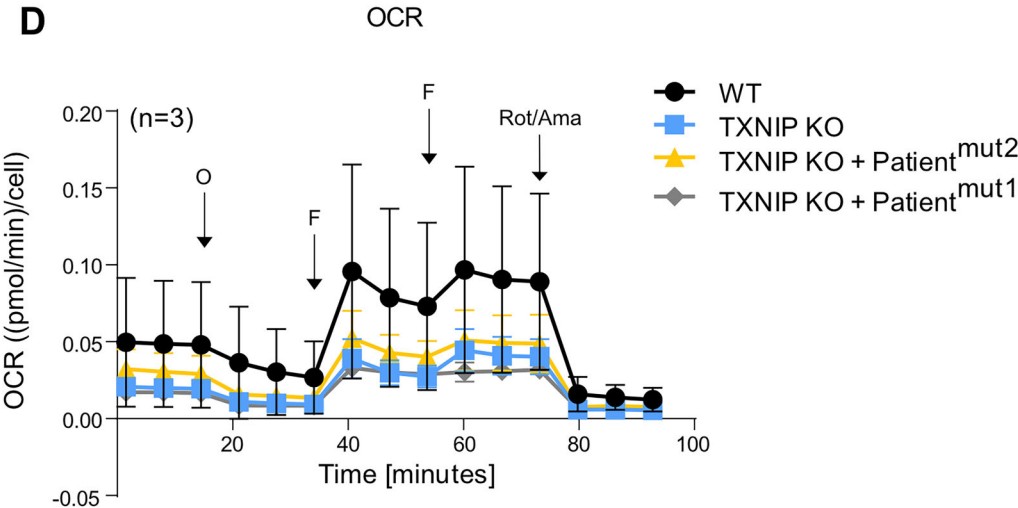

**Figure EV3. The role of TXNIP in SLC7A5 endocytosis and mitochondrial function.**

(A) WB quantification of SCL7A5 from primary human fibroblast normalized to GAPDH Box plots represent the median (centre line) and the interquartile range (25th to 75th percentile box). The whiskers show the minimum and maximum values ($n = 3$, one-way anova, Tukey's multiple comparison test). (B) Quantification of SLC3A2 IF signal at the PM of primary human fibroblast. Box plots represent the median (centre line) and the interquartile range (25th to 75th percentile box). The whiskers show the minimum and maximum values. ($n = 40$). (C) Quantification of SLC3A2 IF signal at the PM of the indicated RPE1 cells. Box plots represent the median (centre line) and the interquartile range (25th to 75th percentile box). The whiskers show the minimum and maximum values. ($n = 40$). (D) Mitochondrial oxygene consumption rate (OCR) of the indicated RPE1 cells using a Seahorse XF HS Mini Analyzer. The inhibitor injection time points are indicated by O (Oligomycin), F (FCCP), Rot/Ama (Rotenone/ Antimycin A). Dots represent mean values ($n = 3$, SD).

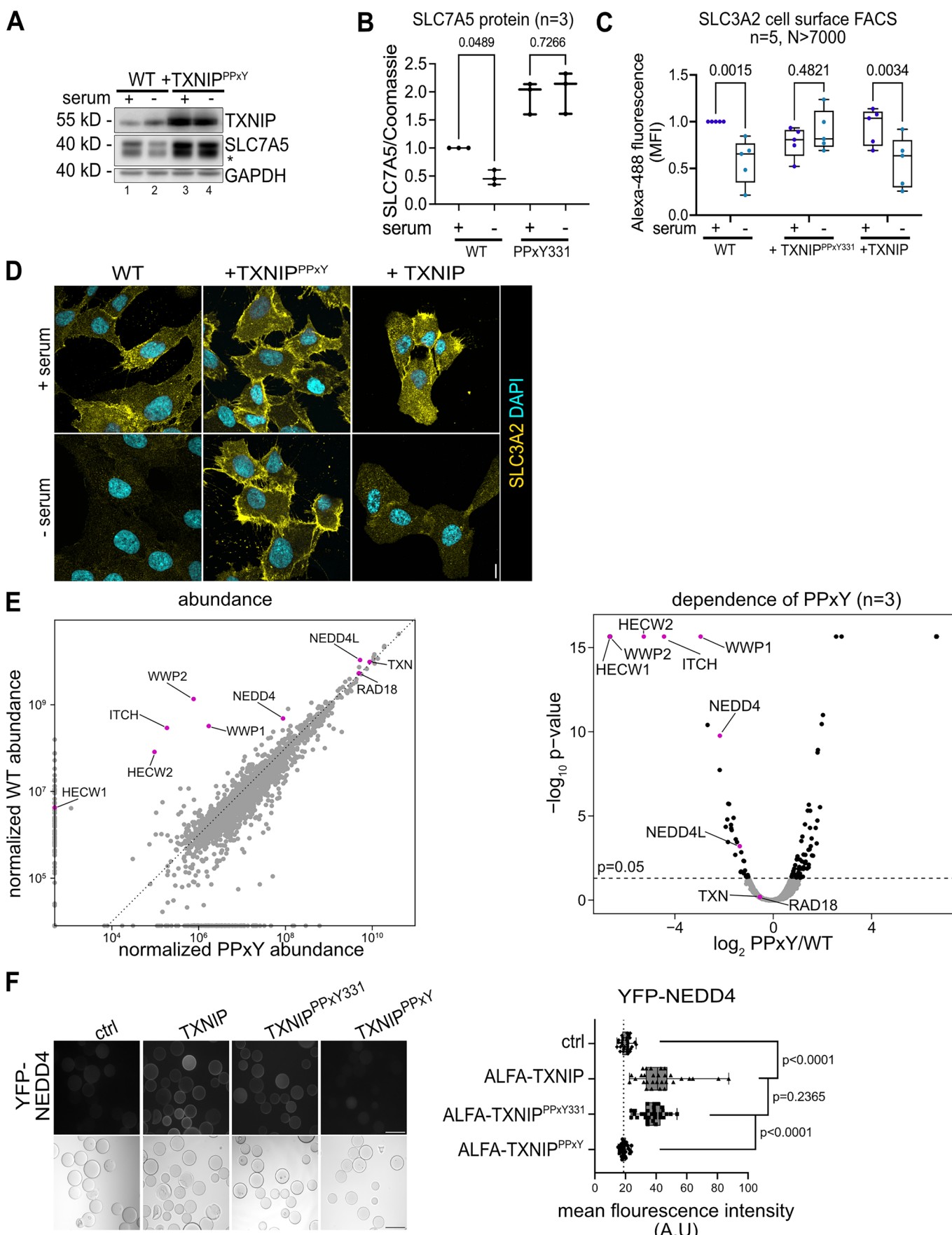

◄

**Figure EV4.   TXNIP interacts with HECT type ubiquitin ligases via its PPxY motif.**

(A) Total cell lysates of RPE1 WT and TXNIP[PPxY] cells that were grown in growth medium (+ serum) or were serum starved for 24 h (− serum) were analyzed by SDS-PAGE and WB with the indicated antibodies. (B) WB quantification of SLC7A5 normalized to the Coomassie-stained membrane as loading control Box plots represent the median (centre line) and the interquartile range (25th to 75th percentile box). The whiskers show the minimum and maximum values. ($n = 3$, one-way ANOVA, Tukey's multiple comparison test). (C) Quantification of SLC3A2 cell surface FACS of the indicated cells, normalized to proliferating WT cells. Box plots represent the median (centre line) and the interquartile range (25th to 75th percentile box). The whiskers show the minimum and maximum values. ($n = 5$, $N > 7000$ cells, two-way ANOVA, Sidak's multiple comparisons test). (D) IF of PFA fixed cells (WT, TXNIP[KO] reconstituted with TXNIP[PPxY] or TXNIP) under + serum or − serum conditions stained for SLC3A2 (yellow) and DAPI (cyan) were analyzed by confocal microscopy. The images show a single plane of a Z-stack. Scale bar $= 10\,\mu m$. (E) Scatter plot comparing interactomes of ALFA-TXNIP with ALFA-TXNIPP[PPxY] ($n = 3$). The HECT type ubiquitin ligases are highlighted in magenta. The abundances of all proteins were normalized to unspecific background binding to empty beads. Volcano plot of log2 transformed abundances. (F) Representative bright-field and fluorescence microscopy images of YFP-NEDD4 bound to ALFA-TXNIP (and the indicated PPxY mutants) immobilized to beads and image quantifications of the YFP-NEDD4 fluorescence signal. Box plots represent the median (centre line) and the interquartile range (25th to 75th percentile box). The whiskers show the minimum and maximum values. ($n = 40$ beads).

    

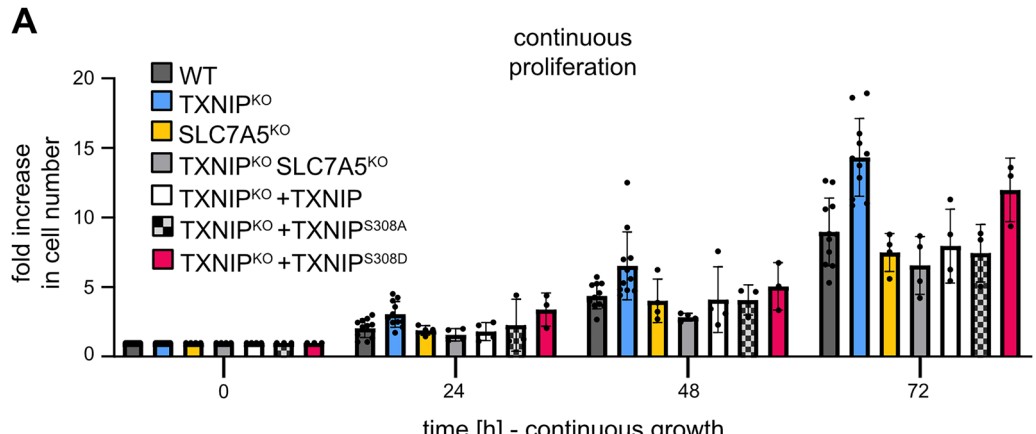

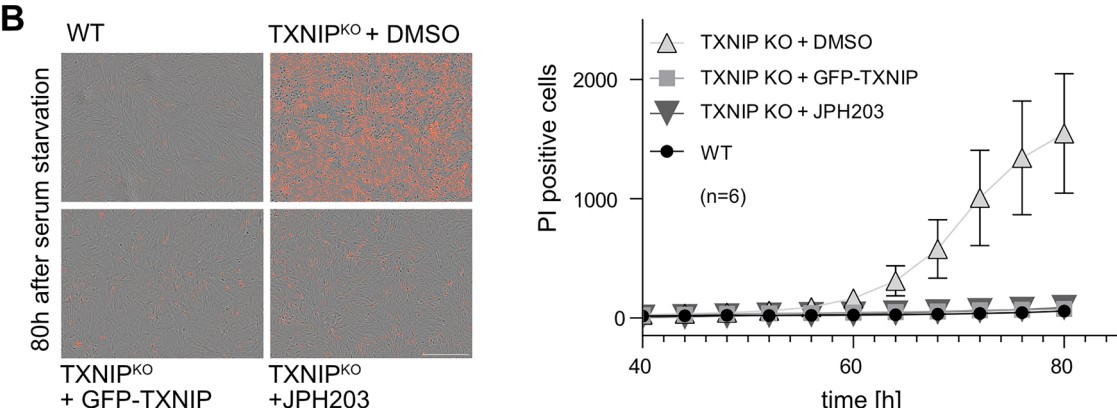

**Figure EV5.  TXNIP helps to control cell growth and survival during quiescence.**

(A) The proliferation of the indicated cells was determined using CASY cell counting during the indicated time. The relative increase in cell number is normalized to 0 h and show as fold increase. Bar charts represent the mean value ($n > 3$, SD). (B) Analysis of cell death during quiescence of WT, TXNIP[KO] TXNIP[KO] ($+$ DMSO), TXNIP[KO] reconstituted with GFP-TXNIP and TXNIP[KO] treated with JPH203 (10 μM). 100.000 cells were seeded in growth medium ($+$ serum). After 24 h the medium was changed to starvation medium (-serum) containing propidium iodide (1 μg/ml). At the same time JPH203 (10 μM) was added. Cells and PI positive cells were continuously monitored for 80 h using live cell microscopy in an Incucyte incubator. Scale bar $=$ 400 μm ($n = 6$, SEM).

