## [Peer Review File · The EMBO Journal]

TXNIP mediates LAT1/SLC7A5 endocytosis to limit amino acid uptake in cells entering quiescence

Jennifer Kahlhofer, Nikolas Marchet, Kristian Zubak, Brigitta Seifert, Madlen Hotze, Anna-Sophia Egger, Lucija Kucej, Claudia Manzl, Yannick Weyer, Sabine Weys, Martin Offterdinger, Sebastian Herzog, Veronika Reiterer, Chiara Volani, Marcel Kwiatkowski, Saskia Wortmann, Siamak Nemati, Johannes Mayr, Johannes Zschocke, Bernhard Radlinger, Kathrin Thedieck, Leopold Kremser, Bettina Sarg, Lukas Huber, Hesso Farhan, Mariana de Araujo, Susanne Kaser, Sabine Scholl-Burgi, Daniela Karall, and David Teis

Corresponding author(s): David Teis (david.teis@i-med.ac.at)

Review Timeline:

Transfer from Review Commons:	10th Mar 25
Editorial Decision:	14th Mar 25
Revision Received:	23rd Jul 25
Editorial Decision:	26th Sep 25
Revision Received:	8th Oct 25
Accepted:	10th Oct 25

Editor: Hartmut Vodermaier

Transaction Report: This manuscript was transferred to The EMBO JOURNAL following peer review at Review Commons.

Review #1

1. Evidence, reproducibility and clarity:

Evidence, reproducibility and clarity (Required)

****Summary:****

In their study, Kahlhofer et al. investigate the mechanism by which cells downregulate amino acid (AA) uptake while entering quiescence. Using western blotting, immunohistochemistry and KO cell lines, the authors show that the α -arrestin family protein TXNIP acts as a regulator of specific membrane-resident AA transporters. They demonstrate that TXNIP promotes the endocytosis and degradation of SLC7A5-SLC7A3 in serum-starved cells as a result of reduced AKT signalling. They further show that the molecular mechanism involves a direct interaction between a PPCY motif in TXNIP and HECT-type ubiquitin ligases which promote AA transporter ubiquitination. Additionally, they identify a novel TXNIP loss-of-function in a patient and show that patient-derived fibroblasts fail to downregulate SLC7A5-SLC7A3 upon starvation. This dysregulation likely contributes to persistent alterations in serum AA levels observed in the patient.

Experiments are well designed and important controls have been performed. Overall, the claims and the conclusions are supported by the data.

****Minor comments:****

Authors should indicate how often western blot experiments were repeated with similar results. Ideally band quantification (as in Fig. 2b) for the most relevant proteins should be provided for all shown Western blots.

For confocal images no n number of experiments/analysed cells is stated. Often only 2-3 cells are shown in these images. In some figures, conclusions from these confocal images are additionally supported by cell surface FACS. For panels with missing cell surface FACS quantifications, the authors should consider using the existing imaging data to perform quantifications of the membrane signal. In this way the reader can get the right impression of the reproducibility of the phenotype described.

I appreciate that the authors have also investigated SLC2A1 endocytosis in their experimental setup. Interestingly, they found that TXNIP mediated downregulation of SLC7A5-SLC3A2 was not linked to TXNIP mediated SLC2A1 endocytosis. Since the role of

TXNIP in glucose metabolism has been studied in more detail in the past, it would be interesting if the authors could further comment on the differences/similarities in the molecular mechanism of glucose and AA transporter downregulation in the discussion.

I would recommend a colour blind-friendly colour palette for the confocal images

2. Significance:

Significance (Required)

The study is well-executed, and the claims are supported by appropriate experiments. As introduced by the authors in their introduction, ubiquitin-dependent endocytosis of AA transporters has been previously shown in *S. cerevisiae* and TXNIP has previously been identified as a regulator of glucose uptake by promoting endocytosis of GLUT1 and GLUT4. Here, the authors identify the molecular mechanism by which TXNIP promotes the endocytosis, and degradation of amino acid transporters (SLC7A5-SLC7A3) through its interaction with HECT-type ubiquitin ligases. This is an advance in the field and will be of interest for researchers in the fields of quiescence, metabolism and cell biology.

3. How much time do you estimate the authors will need to complete the suggested revisions:

Estimated time to Complete Revisions (Required)

(Decision Recommendation)

Less than 1 month

Yes

Review #2

1. Evidence, reproducibility and clarity:

Evidence, reproducibility and clarity (Required)

Summary

Cells entering quiescence must recalibrate metabolism to match lower energy demands, yet the role of endocytosis in this process remains poorly defined. In yeast, amino acid transporters undergo rapid endocytic degradation upon entry into quiescence, but whether a similar mechanism exists in human cells is unknown. Kahlhofer and colleagues demonstrate that human quiescent cells selectively degrade plasma membrane-resident amino acid (AA) transporters, particularly SLC7A5-SLC3A2 (LAT1-4F2hc) and SLC1A5 (ASCT2). TXNIP facilitates LAT1 endocytosis and lysosomal degradation, thereby limiting AA uptake and intracellular AA levels to attenuate mTORC1 signaling and protein translation. In TXNIP-deficient cells, LAT1 remains at the plasma membrane, leading to persistent AA uptake, sustained mTORC1 activation, and accelerated proliferation upon exiting quiescence. In proliferating cells, AKT phosphorylates TXNIP at Ser308, inactivating it and preventing LAT1 degradation, a process that is reversed upon entering quiescence. Notably, the authors identify a biallelic TXNIP loss-of-function variant in a patient with severe metabolic disease, recurrent hypoglycemia, and amino acid imbalances. Patient-derived fibroblasts exhibit defective LAT1 internalization, a phenotype that cannot be rescued by complementation with the pathogenic TXNIP variant, supporting an important role in disease pathology. Functionally, TXNIP-deficient cells have elevated AA levels that sustain mTORC1 activation, enhancing translation, and accelerate exit from quiescence. This study establishes TXNIP as a key regulator of amino acid transporter endocytosis in quiescent cells, linking metabolic adaptation, mTORC1 signaling, and cell cycle control through a previously unrecognized mechanism.

Major comments

Overall, this is a very interesting study indeed. The use of TXNIP knockout models and a loss-of-function patient variant strengthens the conclusion that TXNIP is required for LAT1 degradation. The functional consequences of TXNIP deficiency (elevated intracellular aa, sustained mTORC1 activation, and accelerated quiescence exit) are well-supported by the data. The major concerns are as follows:

1. The identification of a biallelic TXNIP loss-of-function variant in a patient with metabolic disease and neurological dysfunction is highly significant. However, it is problematic that the manuscript effectively presents a case report but does not explicitly frame it as such, and the clinical details are very superficial (lack of pedigree, genetics, structured disease

timeline, differential diagnosis, any histology/scans/photography and broader metabolic profiling - please see best practices for case reports). Although whole-exome sequencing identified the TXNIP variant, it remains unclear whether other genetic or metabolic contributors were systematically excluded. At first glance, the clinical discovery strengthens the physiological significance of the cell biology. However, a discrepancy remains between the clear neurological presentation of the patient (intellectual disability, autism and epilepsy) and the fibroblast-based TXNIP-LAT1 mechanism described in the study. Furthermore, the metabolic phenotype described in this manuscript is significantly more severe than that reported in a previous Swedish study of TXNIP deficiency in humans, where the clinical presentation was milder. This discrepancy suggests that different TXNIP mutations may lead to a spectrum of clinical outcomes, which is highly novel (i.e. metabolic and neurological in terms of loss of function, and carcinogenesis with respect to association studies, reviewed in PMID: 37794178). Of course, this could be influenced by mutation type, genetic background, compensatory mechanisms or environmental factors - it is noteworthy that the previous siblings had mitochondrial dysfunction, and this remains unknown in the present individual. Addressing this variability and discussing potential reasons for the pronounced phenotype observed in this patient would strengthen the manuscript overall. It is noteworthy that LAT1 is highly expressed in brain endothelial cells, which can also adopt a quiescent state (PMID: 33627876), and the authors should expand beyond the single sentence in their discussion. In the absence of the above details, the title and conclusions of Figure 3 and in the discussion greatly overstate causality, implying a direct relationship between TXNIP loss and metabolic dysfunction, despite data from only one patient. This may indeed be the case, but the claims should be carefully revised to reflect an association rather than definitive causation until additional patients are identified. Additionally, while it is assumed that the authors have obtained ethical approval and informed consent, this needs to be explicitly stated for transparency, with dedicated details in the methods sections. Addressing these issues will improve the rigor and mechanistic coherence of the study - otherwise it is quite disjointed.

2. The authors report that TXNIP interacts with HECT E3 ligases to regulate substrate degradation, yet this conclusion is drawn from overexpression-based immunoprecipitation studies, which do not confirm interaction under endogenous conditions. Without direct evidence of TXNIP-HECT E3 binding at native expression levels, this mechanistic link remains unresolved. Given that the authors have already generated antibody-validated TXNIP KO models, endogenous validation should be feasible if the interactions are not super-transient.

3. What are the temporal dynamics of TXNIP-associated degradation, and is this process distinct from endosomal microautophagy (as reported in PMID: 30018090)? The authors present convincing, high-quality FACS-based data supporting TXNIP-mediated turnover. If

this pathway is mechanistically separate from endosomal microautophagy, it suggests a hierarchy of degradation pathways leading to quiescence. Live cell imaging studies that define the temporal dynamics of this process using the tools the authors have created may reveal the relationship between these processes and refine the broader implications of TXNIP in homeostatic adaptation.

****Minor comments****

In the discussion, the authors might briefly speculate on the implications of any functional redundancy that might exist between other arrestins.

2. Significance:

Significance (Required)

This study establishes TXNIP as a regulator of LAT1 endocytosis and metabolic homeostasis in quiescence. The integration of KO models and a TXNIP-deficient patient strengthens the findings, though clinical characterization remains underdeveloped relative to the mechanism reported, and biochemical interactions require endogenous validation. The work expands our understanding of TXNIP beyond association studies, positioning it as a key player in nutrient sensing and metabolic regulation. Addressing the concerns will enhance its relevance across fields - particularly metabolism, cell biology, and disease research.

****Referees cross-commenting****

The comments raised by Reviewer #1 are reasonable, well-founded and align well with the concerns I have raised.

Expertise: Organelle dynamics/degradation, metabolism, biochemistry, tissue homeostasis/disease.

3. How much time do you estimate the authors will need to complete the suggested revisions:

Estimated time to Complete Revisions (Required)

(Decision Recommendation)

Between 1 and 3 months

Yes

Revision Plan

Manuscript number: RC-2024-02821

Corresponding author(s): David, Teis

1. General Statements [optional]

Ruby Ponnudurai, PhD
Scientific Editor, Review Commons,
r.ponnudurai@reviewcommons.org

Hartmut Vodermaier, PhD
Senior Editor | *The EMBO Journal*
h.vodermaier@embojournal.org

Dear Dr. Ponnudurai, dear Dr. Vodermaier, dear Hartmut,

We are pleased to transfer our peer reviewed manuscript '**TXNIP mediates LAT1/SLC7A5 endocytosis to reduce amino acid uptake in cells entering quiescence**' from *Review Commons* to *the EMBO Journal* for your consideration.

How cells maintain a homeostatic pool of amino acids is a central question in biology, and relevant for growth, mTOR signaling (e.g.: Nature Cell Biology, 2024) and metabolic cross-talk (e.g.: Cell Metabolism, 2024). Much is known about how growing cells upregulate amino acid transporters to scale anabolism with growth and how cancer cells hijack these regulatory circuitries to fuel their metabolism. **How cells that enter quiescence reduce the uptake of exogenous amino acids was not understood and the molecular mechanism have remained elusive.**

In our study, we discover that quiescent cells use α -arrestin – ubiquitin ligase complexes to select specific plasma membrane-resident amino acid transporters for endocytosis and lysosomal degradation. This process aligns amino acid uptake with decreased translational demands. We characterized the underlying molecular mechanism, and identified a patient with a severe inborn metabolic disease that fails in quiescence induced endocytosis.

Our findings put forward a novel paradigm for amino acid homeostasis with significant implications for human health. The **key discoveries** of our study include:

1. **Selective Downregulation of Amino Acid Transporters in Quiescence:** Non-cancerous cells entering quiescence selectively downregulate a subset of amino acid transporters, notably the essential amino acid transporter complex SLC7A5-SLC3A2 (LAT1-4F2hc).
2. **TXNIP-Dependent Endocytosis and Degradation of SLC7A5-SLC3A2:** The downregulation of SLC7A5-SLC3A2 is mediated by the α -arrestin TXNIP, which promotes transporter

Revision Plan

ubiquitination via **HECT ubiquitin ligases** for selective endocytosis and lysosomal degradation.

3. **Regulation by AKT:** In growing cells, TXNIP function is inhibited by AKT-mediated phosphorylation, thereby promoting amino acid uptake. AKT inhibition triggers SLC7A5-SLC3A2 endocytosis to reduce amino acid uptake.
4. **Impact of TXNIP Deficiency:** TXNIP deficiency disrupts amino acid homeostasis, enhances mTORC1 signaling, increases translation, and accelerates exit from quiescence, ultimately leading to increased cellular proliferation.
5. **Clinical Significance:** We have identified a novel homozygous loss-of-function mutation in TXNIP in a human patient, which is associated with a severe metabolic disorder and also disrupts SLC7A5-SLC3A2 endocytosis.

Our manuscript has received **favorable reviews from two reviewers at 'Review Commons'**. Both reviewers consider our work an **interesting advance for researchers studying quiescence, metabolism, and cell biology**.

We are now submitting a partially revised version of our manuscript, in which we have already addressed many of the reviewers' comments. In particular, we have **addressed all points raised by Reviewer #1**, including the quantification of Western blot and immunofluorescence data. Additionally, we **now provide a detailed description (case study) of the patient** with the novel TXNIP variant that we identified and characterized—this was the major point raised by Reviewer #2.

We remain fully committed to resolving any remaining open issues, as outlined in our separate revision plan (RC-2024-02821).

We believe that our work is **an excellent match for the EMBO Journal**, since it has broad implications for understanding nutrient homeostasis, metabolic signaling, and their intersections with quiescence biology and metabolic diseases.

Thank you for considering our manuscript.

Best regards,

David

2. Description of the planned revisions

Reviewer 1: Reviewer 1 did not raise major concerns. **The majority of the minor comments have been already addressed in the partially revised manuscript** (please see below - Description of the revisions that have already been incorporated in the transferred manuscript), with one exception:

Minor comment 1: The authors should indicate how often western blot experiments were repeated with similar results. Ideally band quantification (as in Fig. 2b) for the most relevant proteins should be provided for all shown Western blots.

Response: Each Western Blot (WB) experiment has been performed at least 3 times and each WB result for SLC7A5 is complemented by immunofluorescence and/or additionally by FACS analysis, across the manuscript. In addition, **we incorporated WB quantifications of SLC7A5 protein levels for Figures 1c, f, h, Figure 3b, Figure 4b, and Figure 5a, c** (please see below - Description of the revisions that have already been incorporated in the transferred manuscript).

During the revision, we will address how the other known TXNIP variant (TXNIP p.Gln58His; p.Gly59*; PMID: 30755400) affects nutrient transporter endocytosis. This TXNIP variant will be expressed in TXNIP^{KO} RPE1 cells to analyze its effect on quiescence induced SLC7A5 downregulation. The results of this experiment will allow comparing directly the effect of both known TXNIP variants (p.Gln58His; p.Gly59* and p.Ile215TyrfsTer59) on SLC7A5 downregulation in an identical genetic background. **These new results will replace Figure 3d, and therefore we do not yet provide a quantification of Figure 3d.**

Reviewer 2: Reviewer 2 had three major concerns.

Concern 1: The identification of a biallelic TXNIP loss-of-function variant in a patient with metabolic disease and neurological dysfunction is highly significant. However, it is problematic that the manuscript effectively presents a case report but does not explicitly frame it as such, and the clinical details are very superficial (lack of pedigree, genetics, structured disease timeline, differential diagnosis, any histology/scans/photography and broader metabolic profiling - please see best practices for case reports). Although whole-exome sequencing identified the TXNIP variant, it remains unclear whether other genetic or metabolic contributors were systematically excluded. At first glance, the clinical discovery strengthens the physiological significance of the cell biology. However, a discrepancy remains between the clear neurological presentation of the patient (intellectual disability, autism and epilepsy) and the fibroblast-based TXNIP-LAT1 mechanism described in the study. Furthermore, the metabolic phenotype described in this

manuscript is significantly more severe than that reported in a previous Swedish study of TXNIP deficiency in humans, where the clinical presentation was milder. This discrepancy suggests that different TXNIP mutations may lead to a spectrum of clinical outcomes, which is highly novel (i.e. metabolic and neurological in terms of loss of function, and carcinogenesis with respect to association studies, reviewed in PMID: 37794178). Of course, this could be influenced by mutation type, genetic background, compensatory mechanisms or environmental factors - it is noteworthy that the previous siblings had mitochondrial dysfunction, and this remains unknown in the present individual. Addressing this variability and discussing potential reasons for the pronounced phenotype observed in this patient would strengthen the manuscript overall. It is noteworthy that LAT1 is highly expressed in brain endothelial cells, which can also adopt a quiescent state (PMID: 33627876), and the authors should expand beyond the single sentence in their discussion. In the absence of the above details, the title and conclusions of Figure 3 and in the discussion greatly overstate causality, implying a direct relationship between TXNIP loss and metabolic dysfunction, despite data from only one patient. This may indeed be the case, but the claims should be carefully revised to reflect an association rather than definitive causation until additional patients are identified. Additionally, while it is assumed that the authors have obtained ethical approval and informed consent, this needs to be explicitly stated for transparency, with dedicated details in the methods sections. Addressing these issues will improve the rigor and mechanistic coherence of the study - otherwise it is quite disjointed.

Response: We have addressed many these valid concerns and **provide a detailed description of the patient in the partially revised manuscript** (please see below - Description of the revisions that have already been incorporated in the transferred manuscript).

In addition, we are currently collecting video- and photo material of the patient, to support our description of the clinical phenotype.

During the revision, we will additionally address how the other known TXNIP variant (TXNIP p.Gln58His; p.Gly59*; PMID: 30755400) affects nutrient transporter endocytosis. This TXNIP variant will be expressed in TXNIP^{KO} RPE1 cells to analyze its effect on quiescence induced SLC7A5 downregulation. The results of this experiment will allow comparing directly the effect of both known TXNIP variants (p.Gln58His; p.Gly59* and p.Ile215TyrfTer59) on SLC7A5 downregulation in an identical genetic background. In addition, we will compare how both TXNIP variants affect mitochondrial function (using Seahorse technology).

Three siblings have been reported to carry the TXNIP p.Gln58His; p.Gly59* variant. The two siblings (a boy and a girl) show a relatively mild phenotype, while one sibling (the youngest boy) has a more severe phenotype with neonatal muscular hypotonia and failure to thrive, and possible autism spectrum disorder, similar to our patient. Despite the differences in overall patient phenotypes, there are also similarities. All patients suffered from neonatal lactic acidosis and hypoglycaemia, as well as alterations in amino acid metabolism. The Gln58His; p.Gly59* variant caused methionine deficiency over time, while the Ile215TyrfTer59 variant resulted in increased

large neutral amino acids (LNAAs, including L, I, V). It seems that both TXNIP variants disrupt metabolic homeostasis, albeit to varying extents. These variations may stem from variations in genetic background, compensatory mechanisms, diet, or medical treatment.

By comparing both TXNIP variants in an identical genetic background (RPE1 TXNIP^{KO} cells), we can at least firmly determine if SLC7A5 downregulation is subject to additional genetic or non-genetic modifiers.

Concern 2: The authors report that TXNIP interacts with HECT E3 ligases to regulate substrate degradation, yet this conclusion is drawn from overexpression-based immunoprecipitation studies, which do not confirm interaction under endogenous conditions. Without direct evidence of TXNIP-HECT E3 binding at native expression levels, this mechanistic link remains unresolved. Given that the authors have already generated antibody-validated TXNIP KO models, endogenous validation should be feasible if the interactions are not super-transient.

Response: While the manuscript was under review, we have improved the stringency of our TXNIP-HECT type ubiquitin ligase interaction experiments and developed additional biochemical experiments that strengthen our original conclusions. In the course of these experiments, we found that the interaction of TXNIP with NEDD4, WWP2 and HECW1/2 (but not with WWP1 or ITCH) were particularly dependent on the PPxY331 motif (Figure 1a, NEDD4 is shown).

Revision Plan

During the revision, we will conduct additional experiments to substantiate these findings and **to narrow down the list possible ubiquitin ligases that are required for the downregulation of SLC7A5**. In particular, we will test if endogenous TXNIP co-immunoprecipitates (in a PPxY motif dependent manner) NEDD4, HECW1/2 or another HECT type ubiquitin ligase.

Furthermore, we will include a newly developed 'Bead-Immobilized Prey Assay (BIPA)', where protein-protein interactions can be analyzed by microscopy in a fast in straight forward manner. In the BIPA, ALFA-TXNIP (or mutant variants) are first captured on ALFA-beads (Bead immobilized). These TXNIP beads are then incubated with cell lysates from HEK293 expressing GFP-tagged HECT type ubiquitin ligases (Prey). The binding of the GFP-tagged ubiquitin ligases to the TXNIP beads is analyzed by fluorescence microscopy and quantified (Figure 1b, a BIPA with YFP-NEDD4). This efficient assay will also be conducted with NEDD4, WWP1, WWP2, HECW2, and ITCH to analyze how they bind to TXNIP, TXNIP-PPxY331 and the PPxY double mutant.

Together we are confident that our experiments establish that TXNIP must interact with a specific subset of HECT type ubiquitin ligase (our prime candidate are NEDD4 and HECW1/2) to trigger SLC7A5-SLC3A2 ubiquitination, endocytosis and lysosomal degradation.

Comment 3: What are the temporal dynamics of TXNIP-associated degradation, and is this process distinct from endosomal microautophagy (as reported in PMID: 30018090)? The authors present convincing, high-quality FACS-based data supporting TXNIP-mediated turnover. If this pathway is mechanistically separate from endosomal microautophagy, it suggests a hierarchy of degradation pathways leading to quiescence. Live cell imaging studies that define the temporal dynamics of this process using the tools the authors have created may reveal the relationship between these processes and refine the broader implications of TXNIP in homeostatic adaptation.

Response: Thank you for this interesting suggestion.

During the revision, we will first investigate a potential temporal correlation of endosomal microautophagy of p62/SQSTM1, NBR1, TAX1BP1, NDP52, and NCOA4 (PMID: 30018090) and the downregulation of SLC7A5 as cells enter quiescence. For these experiments, we will follow the turn-over of the above-mentioned autophagy adaptors and compare it to the turnover of SLC7A5, using either WB analysis, or microscopy or both.

Next, we will test if SLC7A5-SLC3A2 endocytosis and lysosomal degradation is required to initiate endosomal micro-autophagy of p62/SQSTM1, NBR1, TAX1BP1, NDP52, and NCOA4 in TXNIP^{KO} cells.

Together, these experiments will address if the endosomal micro-autophagy and TXNIP mediated downregulation of SLC7A5 are mechanistically linked during entry into quiescence.

3. Description of the revisions that have already been incorporated in the transferred manuscript

Please insert a point-by-point reply describing the revisions that were already carried out and included in the transferred manuscript. If no revisions have been carried out yet, please leave this section empty.

Reviewer 1:

Minor comment 1: The authors should indicate how often western blot experiments were repeated with similar results. Ideally band quantification (as in Fig. 2b) for the most relevant proteins should be provided for all shown Western blots.

Response: Each Western Blot (WB) experiment has been performed at least 3 times and each WB result for SLC7A5 is complemented by immunofluorescence and/or additionally by FACS analysis, across the manuscript.

We already **incorporated WB quantifications of SLC7A5 protein levels** for Figures 1c, f, h, Figure 3b, Figure 4b, and Figure 5a, c in **Supplementary Figure 1b, Supplementary Figure 2c, f, Supplementary Figure 4a, e, and in Supplementary Figure 5a, c**, respectively.

Minor comment 2: For confocal images no n number of experiments/analyzed cells is stated. Often only 2-3 cells are shown in these images. In some figures, conclusions from these confocal images are additionally supported by cell surface FACS. For panels with missing cell surface FACS quantifications, the authors should consider using the existing imaging data to perform quantifications of the membrane signal. In this way the reader can get the right impression of the reproducibility of the phenotype described.

Response: Each immunofluorescence experiment has been performed at least 3 times. Line-scan quantification of immunofluorescence (IF) of SLC3A2 at the plasma membrane (PM) is now provided for immunofluorescence experiments in Figure 1e, g, Figure 3c, e in **Supplementary Figure 2b, e, Supplementary Figure 4b, c**, and for SLC2A1 in **Supplementary Figure 3i**, where FACS data was missing. In addition, WB experiments complement the results of each IF experiment.

Minor comment 4: I appreciate that the authors have also investigated SLC2A1 endocytosis in their experimental setup. Interestingly, they found that TXNIP mediated downregulation of SLC7A5-SLC3A2 was not linked to TXNIP mediated SLC2A1 endocytosis. Since the role of TXNIP in glucose metabolism has been studied in more detail in the past, it would be interesting if the authors could further comment on the differences/similarities in the molecular mechanism of glucose and AA transporter downregulation in the discussion.

Revision Plan

Response: Thank you for bringing up this point. We now have added the following paragraph to the discussion to speculate about the differences/similarities in the molecular mechanism of glucose and AA transporter downregulation in the discussion:

Moreover, in RPE1 cells entering quiescence, GLUT1/4 was not downregulated. Hence, it seems that TXNIP can discriminate, in a context dependent manner, between targeting SLC7A5-SLC3A2 or GLUT1/4 for endocytosis. Since AKT mediated phosphorylation invariably appeared to inactivate TXNIP, and dephosphorylation re-activated it, additional mechanism must confer TXNIP selectivity towards SLC7A5-SLC3A2 or GLUT1/4. We consider it likely, that the exposure of sorting motifs in cytosolic tails of SLC7A5 or GLUT1/4 could regulate the binding of activated TXNIP and thus controls selective endocytosis to adapt nutrient uptake. The exposure of these sorting motifs could be dependent on the metabolic context / state of the cell. Indeed, yeast α -arrestins can detect n- or c-terminal acidic sorting motifs in amino acid transporters, respectively, that are alternatively exposed in response to amino acid excess or starvation (Ivashov *et al.*, 2020a) (Guiney *et al.*, 2016). Inspection of the SLC7A5 sequence indicates a possible n-terminal acidic sorting motif (17EEKKEEAREK25). Two lysine residues (K19, K25) in this sequence have been found to be ubiquitinated in an earlier study upon protein kinase C (PKC) activation and mTORC1 inhibition (Barthelemy & Andre, 2019; Rosario *et al.*, 2016).

Minor comment 5: I would recommend a colour blind-friendly colour palette for the confocal images

Response: Thank you for pointing this out – we have changed the color palette accordingly.

reviewer 2:

Comment 1: The identification of a biallelic TXNIP loss-of-function variant in a patient with metabolic disease and neurological dysfunction is highly significant. However, it is problematic that the manuscript effectively presents a case report but does not explicitly frame it as such, and the clinical details are very superficial (lack of pedigree, genetics, structured disease timeline, differential diagnosis, any histology/scans/photography and broader metabolic profiling - please see best practices for case reports). Although whole-exome sequencing identified the TXNIP variant, it remains unclear whether other genetic or metabolic contributors were systematically excluded. At first glance, the clinical discovery strengthens the physiological significance of the cell biology. However, a discrepancy remains between the clear neurological presentation of the patient (intellectual disability, autism and epilepsy) and the fibroblast-based TXNIP-LAT1

mechanism described in the study. Furthermore, the metabolic phenotype described in this manuscript is significantly more severe than that reported in a previous Swedish study of TXNIP deficiency in humans, where the clinical presentation was milder. This discrepancy suggests that different TXNIP mutations may lead to a spectrum of clinical outcomes, which is highly novel (i.e. metabolic and neurological in terms of loss of function, and carcinogenesis with respect to association studies, reviewed in PMID: 37794178). Of course, this could be influenced by mutation type, genetic background, compensatory mechanisms or environmental factors - it is noteworthy that the previous siblings had mitochondrial dysfunction, and this remains unknown in the present individual. Addressing this variability and discussing potential reasons for the pronounced phenotype observed in this patient would strengthen the manuscript overall. It is noteworthy that LAT1 is highly expressed in brain endothelial cells, which can also adopt a quiescent state (PMID: 33627876), and the authors should expand beyond the single sentence in their discussion. In the absence of the above details, the title and conclusions of Figure 3 and in the discussion greatly overstate causality, implying a direct relationship between TXNIP loss and metabolic dysfunction, despite data from only one patient. This may indeed be the case, but the claims should be carefully revised to reflect an association rather than definitive causation until additional patients are identified. Additionally, while it is assumed that the authors have obtained ethical approval and informed consent, this needs to be explicitly stated for transparency, with dedicated details in the methods sections. Addressing these issues will improve the rigor and mechanistic coherence of the study - otherwise it is quite disjointed.

Response: thank you for pointing this out. We now provide a detailed description of the patient, that addresses the reviewers concerns regarding the patient description.

The patient is a boy, born in 2014 as the first child of healthy, consanguineous parents of Turkish origin. During pregnancy, the mother was diagnosed with polyhydramnios. At 38 + 6 weeks of gestation, the baby was in a breech position, leading to a cesarean section. At birth, he weighed 3880 g (P90), measured 55 cm in length, and had a head circumference of 38 cm.

On the seventh day of life, he exhibited floppiness, recurrent hypoglycaemia, and lactic acidosis, prompting his transfer from the birth hospital to a tertiary care centre. During the first three days there, his lowest recorded blood glucose level was 30 mg/dl, lactate levels were approximately 6.5 mmol/l, and pH was 7.11. Subsequently, he developed hypertriglyceridemia, with triglyceride levels reaching 364 mg/dl. Initially stable, he began experiencing elevated pCO₂ levels (up to 70 mmHg due to bradypnea) and metabolic acidosis on day 10. A glucose infusion (10 mg/kg/min) stabilized his glucose and lactate concentrations, though lactate remained elevated at around 3-4 mmol/l. Regardless, his muscular hypotonia persisted. On day 12, a skin punch biopsy for a fibroblast culture was performed.

Revision Plan

By day 20, glucose and lactate levels had stabilized with regular feeding, allowing his transfer back to a peripheral hospital. During infancy, his blood glucose concentrations were within standard range (Supplementary Table 1), but the boy experienced recurrent hypoglycaemia in response to metabolic stress, e.g., infections. He exhibited psychomotor developmental delays and, from 18 months of age, experienced increasing epileptic seizures (up to 3-4 per month), which were managed with levetiracetam, topiramate, and lamotrigine. Currently, he remains metabolically stable but presents with significant developmental delay, muscular hypotonia, and autistic features.

Whole-exome sequencing from peripheral blood of the patient detected a homozygous single nucleotide insertion c.642_643insT in exon 5 of 8 of the TXNIP gene. This variant was not recorded in the population genetic variant database gnomAD that lists TXNIP as likely haplosufficient (pLI = 0, LOEUF = 0,709: <https://gnomad.broadinstitute.org> accessed Sept. 10, 2024). No other (likely) pathogenic variant in any other gene, with known function in metabolism was identified as explanation of the clinical features in the child. Potential pathogenic variants in genes required for mitochondrial functions were also not detected, although they were initially expected to cause the phenotype of the boy.

The TXNIP variant c.642_643insT caused a frameshift and a premature stop codon after 59 AA (denoted p.Ile215TyrfsTer59), likely causing nonsense-mediated decay (NMD) or the synthesis of a severely truncated TXNIP protein (Figure 3a). Both parents are healthy heterozygous carriers for the TXNIP variant. Serendipitously, this TXNIP variant was similar to the gene-edited version in the RPE1 TXNIP^{KO} cells (p.I215TfsX11).

The patient showed consistent metabolic alterations compatible with an AA transporter deficiency. Blood plasma concentrations of several large neutral amino acids (LNAAs, including L, I, V) were elevated throughout the years 2014 – 2022 (Supplementary Table 1). The increased molar ratio of the LNAAs (L, I, V) to aromatic AAs (F, Y), resulted in an elevated Fischer's ratio (FR, 2014: FR = 4.46; 2016: FR = 5.38, 2018: FR = 5.90; 2021; FR= 6.98; 2022: FR = 4.23; FR reference range = 2.10 - 4). The methionine levels are not dramatically altered (Supplementary Table 1).

We also provide the following ethical statement:

Ethical statement

All patients' data were extracted from the medical routine records. Written informed consent for molecular genetic studies and publication of data was obtained from the legal guardians of the patient. This approach was approved by the ethics committee of the Medical University of Innsbruck (UN4501-MUI). The study was conducted in accordance with the principles of the Declaration of Helsinki.

4. Description of analyses that authors prefer not to carry out

Please include a point-by-point response explaining why some of the requested data or additional analyses might not be necessary or cannot be provided within the scope of a revision. This can be due to time or resource limitations or in case of disagreement about the necessity of such additional data given the scope of the study. Please leave empty if not applicable.

reviewer 2

Comment 1: At first glance, the clinical discovery strengthens the physiological significance of the cell biology. However, a discrepancy remains between the clear neurological presentation of the patient (intellectual disability, autism and epilepsy) and the fibroblast-based TXNIP-LAT1 mechanism described in the study.

Response: The reviewer is correct and we acknowledge this point in the discussion:

... Two of the three affected children (a boy and a girl) had normal development during childhood; the clinical phenotype of one sibling (the youngest boy) was more severe. This boy suffered from recurrent hypoglycaemia, neonatal muscular hypotonia and failure to thrive, and was under investigation for autism spectrum disorder, similar to our patient.

How these metabolic defects are linked to clinical manifestations such as possible autism and the severe epileptic developmental disorder, remains to be investigated. We speculate that these phenotypes could be related to role of SLC7A5 in neurons and in the blood brain barrier (Knaus *et al*, 2023; Tarlunganu *et al*, 2016). In the cerebral cortex, SLC7A5 mediated transport of large neutral amino acid and lipid metabolism are interconnected and required for perinatal neuronal excitability and survival (Knaus *et al.*, 2023; Tarlunganu *et al.*, 2016). Consistently, mutations in

Revision Plan

SLC7A5 lead to autism spectrum disorders (Tarlungeanu *et al*, 2016). Hence it is not unlikely that dysregulation of SLC7A5, caused by loss of TXNIP, might affect brain development. To definitely address this point, addition experiments are required.

Additional experiments to clarify the role of the TXNIP and SLC7A5 in brain development would indeed provide important new insight. Yet, at the moment, these experiments are beyond the scope of the current study, since they would require rather complex experiments using mouse models.

Dr. David Teis
Medical University of Innsbruck
Biocenter, Molecular Biochemistry
Innrain 80/82
Innsbruck 6020
Austria

14th Mar 2025

Re: EMBOJ-2025-120740-T
TXNIP mediates LAT1/SLC7A5 endocytosis to reduce amino acid uptake in cells entering quiescence

Dear David,

Thank you for transferring your manuscript on TXNIP-dependent adjustment of amino acid uptake upon quiescence entry from Review Commons to The EMBO Journal. I have read your study as well as the transferred referee reports and your responses to them, and in light of this would be happy to consider a revised version further for EMBO Journal publication. I appreciate the additions and clarifications that have already been implicated at this point, as well as the ongoing work to address specific outstanding issues raised by referee 2. The planned revisions for major concerns 1 and 3 of referee 2 all sound very reasonable and promising; as do the planned interaction characterization/mapping experiments in response to concern #2 - here it would be quite helpful to clearly demonstrate the TXNIP interaction with HECT E3s at native expression levels, using antibodies against endogenous proteins for immunoprecipitation and detection.

When preparing a revised manuscript, please try to adhere to the guidelines listed below and in our Guide to Authors as closely as possible, as this should greatly facilitate our assessment at the time of resubmission - in particular regarding the completion of our author checklist and a dedicated reagents and tools table (both linked below), the inclusion of editable text files and separate, individual figures, the formatting of the references, and the conversion of "supplemental" material into Expanded View and/or Appendix content. Please also note that it is our policy to allow only a single round of (major) revision, making it important to comprehensively answer all criticisms at this point - if this should require more time than our standard three-months deadline, I would be happy to discuss an extension of the revision time, during which our 'scooping protection' (meaning that competing work appearing elsewhere in the meantime will not affect our considerations of your study) would of course remain valid. Please do not hesitate to contact me should you have any further questions at this stage.

Thank you again for the opportunity to consider this study for The EMBO Journal. I look forward to receiving your revision.

With kind regards,

Hartmut

We realize that it is difficult to revise to a specific deadline. In the interest of protecting the conceptual advance provided by the work, we recommend a revision within 3 months (12th Jun 2025). Please discuss the revision progress ahead of this time with the editor if you require more time to complete the revisions. Use the link below to submit your revision:

Link Not Available

Rev_Com_number: RC-2024-02821

New_manu_number: EMBOJ-2025-120740-T

Corr_author: Teis

Title: TXNIP mediates LAT1/SLC7A5 endocytosis to reduce amino acid uptake in cells entering quiescence

Reviewer # 1: *The study is well-executed, and the claims are supported by appropriate experiments. As introduced by the authors in their introduction, ubiquitin-dependent endocytosis of AA transporters has been previously shown in *S. cerevisiae* and TXNIP has previously been identified as a regulator of glucose uptake by promoting endocytosis of GLUT1 and GLUT4. Here, the authors identify the molecular mechanism by which TXNIP promotes the endocytosis, and degradation of amino acid transporters (SLC7A5-SLC7A3) through its interaction with HECT-type ubiquitin ligases. This is an advance in the field and will be of interest for researchers in the fields of quiescence, metabolism and cell biology. Experiments are well designed and important controls have been performed. Overall, the claims and the conclusions are supported by the data.*

Response: We thank the reviewer for the thorough evaluation of our manuscript and for the insightful, constructive comments. In our revision, we have addressed all of the reviewer's suggestions. Additionally, we have included new exciting data demonstrating that TXNIP-mediated downregulation of SLC7A5 - and the resulting reduction in amino acid uptake - is essential for cell survival during prolonged quiescence (new Fig. 7D, EV5B). Hence the TXNIP mediated downregulation of amino acid uptake during quiescence has a cytoprotective function for quiescent cells.

Minor comment 1: *The authors should indicate how often western blot experiments were repeated with similar results. Ideally band quantification (as in Fig. 2b) for the most relevant proteins should be provided for all shown Western blots.*

Response: Thank you for pointing this out. Each Western Blot (WB) experiment has been performed at least 3 times and each WB result for SLC7A5 is complemented by immunofluorescence and/or by FACS analysis, across the manuscript.

We incorporated WB quantifications of SLC7A5 protein levels from Fig. 1C, F, H in Appendix Fig.S1B, EV1B, G; from Fig. 2A and EV2E in Fig.2B and EV2F; from Fig.3B in EV3A, from Fig.4B in EV4B; from Fig.5A, B in Appendix Fig. S4A, C, respectively.

Minor comment 2: *For confocal images no n number of experiments/analyzed cells is stated. Often only 2-3 cells are shown in these images. In some figures, conclusions from these confocal images are additionally supported by cell surface FACS.*

Response: Each immunofluorescence experiment has been performed at least 3 times.

Minor comment 3: *For panels with missing cell surface FACS quantifications, the authors should consider using the existing imaging data to perform quantifications of the membrane signal. In this way the reader can get the right impression of the reproducibility of the phenotype described.*

Response: Each immunofluorescence experiment has been performed at least 3 times. Line-scan quantification of immunofluorescence (IF) of SLC3A2 at the plasma membrane (PM) is now provided for immunofluorescence experiments in Fig. 1D, G, Figure 3C, E in Fig. EV1C, F and EV3B, C and for SLC2A1 in Fig. EV3J, where FACS data was missing. In addition, WB experiments complement the results of each IF experiment.

Minor comment 4: I appreciate that the authors have also investigated SLC2A1 endocytosis in their experimental setup. Interestingly, they found that TXNIP mediated downregulation of SLC7A5-SLC3A2 was not linked to TXNIP mediated SLC2A1 endocytosis. Since the role of TXNIP in glucose metabolism has been studied in more detail in the past, it would be interesting if the authors could further comment on the differences/similarities in the molecular mechanism of glucose and AA transporter downregulation in the discussion.

Response: Thank you for bringing up this point. We now have added the following paragraph to the discussion to speculate about the differences/similarities in the molecular mechanism of glucose and AA transporter downregulation in the discussion:

‘TXNIP was specifically required for the endocytosis of SLC7A5-SLC3A2, but not for the endocytosis of SLC7A11-SLC3A2. Moreover, in RPE1 cells entering quiescence, SLC2A1 (GLUT1) was not downregulated. Hence, it seems that TXNIP can discriminate, in a context dependent manner, between targeting SLC7A5-SLC3A2 or SLC2A1 (GLUT1) or SLC2A4 (GLUT4) for endocytosis. Since AKT-mediated phosphorylation invariably appears to inactivate TXNIP, and dephosphorylation re-activates it, additional mechanism must confer TXNIP selectivity towards these nutrient transporters. We consider it likely, that the exposure of sorting motifs in their cytosolic tails (Qualls-Histed *et al.*, 2023; Waldhart *et al.*, 2017; Wu *et al.*, 2013a) could regulate the binding of activated TXNIP and thus controls selective endocytosis to adapt nutrient uptake. The exposure of these sorting motifs could be dependent on the metabolic context / state of the cell. Indeed, yeast α -arrestins can detect N- or C-terminal acidic sorting motifs in amino acid transporters that are exposed in response to amino acid excess or starvation (Ivashov *et al.*, 2020a) (Guiney *et al.*, 2016). Inspection of the SLC7A5 sequence indicates a possible N-terminal acidic sorting motif (17EEKKEAREK25). Two lysine residues (K19, K25) in this sequence can be ubiquitinated upon protein kinase C (PKC) activation and mTORC1 inhibition (Barthelemy & Andre, 2019; Rosario *et al.*, 2016). The molecular details of how TXNIP interacts with SLC7A5-SLC3A2 remains unclear at the moment.’

Minor comment 5: I would recommend a colour blind-friendly colour palette for the confocal images

Response: Thank you for pointing this out – we have changed the color palette across all figures accordingly.

Reviewer # 2: This study establishes TXNIP as a regulator of LAT1 endocytosis and metabolic homeostasis in quiescence. The integration of KO models and a TXNIP-deficient patient strengthens the findings, though clinical characterization remains underdeveloped

relative to the mechanism reported, and biochemical interactions require endogenous validation. The work expands our understanding of TXNIP beyond association studies, positioning it as a key player in nutrient sensing and metabolic regulation. Addressing the concerns will enhance its relevance across fields - particularly metabolism, cell biology, and disease research. Overall, this is a very interesting study indeed. The use of TXNIP knockout models and a loss-of-function patient variant strengthens the conclusion that TXNIP is required for LAT1 degradation. The functional consequences of TXNIP deficiency (elevated intracellular aa, sustained mTORC1 activation, and accelerated quiescence exit) are well-supported by the data. The major concerns are as follows:

Response: We thank the reviewer for the thorough evaluation of our manuscript and for the insightful, constructive comments. During the revision we have address all major points. Additionally, we have included new exciting data demonstrating that TXNIP-mediated downregulation of SLC7A5 - and the resulting reduction in amino acid uptake - is essential for cell survival during prolonged quiescence (new Fig. 7D EV5B). Hence the TXNIP mediated downregulation of amino acid uptake during quiescence has a cyto-protective function for quiescent cells.

Major concern 1. *The identification of a biallelic TXNIP loss-of-function variant in a patient with metabolic disease and neurological dysfunction is highly significant. However, it is problematic that the manuscript effectively presents a case report but does not explicitly frame it as such, and the clinical details are very superficial (lack of pedigree, genetics, structured disease timeline, differential diagnosis, any histology/scans/photography and broader metabolic profiling - please see best practices for case reports). Although whole-exome sequencing identified the TXNIP variant, it remains unclear whether other genetic or metabolic contributors were systematically excluded. At first glance, the clinical discovery strengthens the physiological significance of the cell biology. However, a discrepancy remains between the clear neurological presentation of the patient (intellectual disability, autism and epilepsy) and the fibroblast-based TXNIP-LAT1 mechanism described in the study. Furthermore, the metabolic phenotype described in this manuscript is significantly more severe than that reported in a previous Swedish study of TXNIP deficiency in humans, where the clinical presentation was milder. This discrepancy suggests that different TXNIP mutations may lead to a spectrum of clinical outcomes, which is highly novel (i.e. metabolic and neurological in terms of loss of function, and carcinogenesis with respect to association studies, reviewed in PMID: 37794178). Of course, this could be influenced by mutation type, genetic background, compensatory mechanisms or environmental factors - it is noteworthy that the previous siblings had mitochondrial dysfunction, and this remains unknown in the present individual. Addressing this variability and discussing potential reasons for the pronounced phenotype observed in this patient would strengthen the manuscript overall. It is noteworthy that LAT1 is highly expressed in brain endothelial cells, which can also adopt a quiescent state (PMID: 33627876), and the authors should expand beyond the single sentence in their discussion. In the absence of the above details, the title and conclusions of Figure 3 and in the discussion greatly overstate causality, implying a direct relationship between TXNIP loss and metabolic dysfunction, despite data from only one patient. This may indeed be the case, but the claims should be carefully revised to reflect an association rather than definitive causation until additional patients are identified. Additionally, while it is assumed that the authors have obtained ethical approval and informed consent, this needs*

to be explicitly stated for transparency, with dedicated details in the methods sections. Addressing these issues will improve the rigor and mechanistic coherence of the study - otherwise it is quite disjointed.

Response: We now provide a more detailed description of the patient's medical history (new Appendix: medical history), including photographic documentation of the patient (new Appendix Figure S2B), and the recent laboratory blood analyses (updated Appendix Table S1) (June 2025):

‘The patient is a boy, born in 2014 as the first child of healthy, consanguineous parents of Turkish origin. More details on his medical history are provided in the appendix (Appendix Medical History). On the seventh day of life, he exhibited floppiness, recurrent hypoglycaemia, and lactic acidosis. A glucose infusion (10 mg/kg/min) stabilized his glucose and lactate concentrations. By day 20, glucose and lactate levels had stabilized with regular feeding but his muscular hypotonia persisted. During infancy, his blood glucose concentrations were within standard range (Appendix Table S1), but the boy experienced recurrent hypoglycaemia in response to metabolic stress, e.g., infections. He exhibited psychomotor developmental delays and, from 18 months of age, experienced increasing epileptic seizures (up to 3 - 4 per month). Currently, he remains metabolically stable but presents with significant developmental delay, muscular hypotonia, and autistic features (Appendix Fig. S2B).

Exome sequencing from peripheral blood of the patient detected a homozygous single nucleotide insertion c.642_643insT in exon 5 of 8 of the *TXNIP* gene. Both parents are healthy heterozygous carriers for this variant. This *TXNIP* variant was not recorded in the population genetic variant database gnomAD that lists *TXNIP* as likely haplosufficient (pLI = 0, LOEUF = 0,709: <https://gnomad.broadinstitute.org> accessed June. 29, 2025). No other (likely) pathogenic variant in any other gene, with known function in metabolism was identified as explanation of the clinical features in the child. Potential pathogenic variants in genes required for mitochondrial functions were also not detected, although they were initially expected to cause the phenotype.

The *TXNIP* variant c.642_643insT caused a frameshift and a premature stop codon after 59 AA (p.Ile215Tyrfs*59), likely causing nonsense-mediated decay (NMD) or the synthesis of a severely truncated *TXNIP* protein (Fig. 3A). Serendipitously, this *TXNIP* variant was similar to the gene-edited version in the RPE1 *TXNIP*^{KO} cells (p.Ile215Thrfs*11).

The patient showed consistent metabolic alterations compatible with an AA transporter deficiency. Blood plasma concentrations of several large neutral amino acids (LNAAs, including L, I, V) were elevated throughout the years 2014 – 2025 (Appendix Table S1).

The increased molar ratio of the LNAAAs (L, I, V) relative to aromatic AAAs (F, Y), resulted in a consistently elevated Fischer's ratio (FR, 2014: FR = 4.46; 2016: FR = 5.38, 2018: FR = 5.90; 2021; FR= 6.98; 2022: FR = 4.23; 2025 FR = 5.36; FR reference range = 2.10 - 4). The methionine levels were not dramatically altered.'

We also provide the following ethical statement:

Ethical statement

All patients' data were extracted from the medical routine records. Written informed consent for molecular genetic studies and publication of data and images was obtained from the legal guardians of the patient. This approach was approved by the ethics committee of the Medical University of Innsbruck (UN4501-MUI). The study was conducted in accordance with the principles of the Declaration of Helsinki.

Furthermore, we have directly compared the functional effects of this TXNIP variant (p.Ile215TyrfsTer59; TXNIP^{mut2}) with the previously identified 'swedish' TXNIP variant (p.Gln58His; p.Gly59; TXNIP^{mut1}) (Katsu-Jimenez et al., 2019). Therefore, we expressed their respective cDNAs in RPE1 TXNIP^{KO} cells to generate an isogenic cellular background. In cells expressing either TXNIP variant, SLC7A5-SLC3A2 degradation was blocked (new Fig. 3D).

Moreover, as requested by the reviewer, we used these cells to compare their mitochondrial respiration (using seahorse analyzer). Cells carrying the bi-allelic TXNIP^{mut1} variant have been reported to have functional mitochondria; however the channeling of glucose and pyruvate into oxidative phosphorylation was inefficient, while malate was accepted as a substrate (Katsu-Jimenez *et al.*, 2019). We confirmed that RPE1 TXNIP^{KO} cells, as well as RPE1 TXNIP^{KO} cells expressing TXNIP^{mut1} but also TXNIP^{mut2} cells exhibited reduced oxygen consumption rates compared to wild-type cells (Figure EV3D). Unfortunately, we could not analyze these cells under quiescence conditions, because the quiescent RPE1 cells detached from the specialized microplates designed for use with Seahorse Analyzers.

We concluded that, TXNIP loss-of-function mutations impaired SLC7A5 downregulation in cells entering quiescence deregulating cellular amino acid levels. In growing RPE1 cells, loss of TXNIP function was associated with reduced oxidative phosphorylation and slightly increased glucose uptake. In human patients, these cellular defects converged into metabolic imbalances that underlie a wide spectrum of clinical manifestations.

Certainly, it will be exciting to better characterize the underlying molecular mechanism. Yet, these studies will require new experimental approaches that go beyond the scope of this manuscript. To address these points, we significantly extended the discussion:

'How these cellular metabolic defects are linked to clinical manifestations such as possible autism and the severe epileptic developmental disorder, remains to be investigated. We

speculate that these phenotypes could be related to roles of SLC7A5 in neurons and in the blood brain barrier (Knaus *et al.*, 2023; Tarlungeanu *et al.*, 2016). In the cerebral cortex, SLC7A5-mediated transport of large neutral amino acid appears interconnected with lipid metabolism and required for perinatal neuronal excitability and survival (Knaus *et al.*, 2023; Tarlungeanu *et al.*, 2016). Consistently, pathogenic variants in SLC7A5 lead to autism spectrum disorders (Tarlungeanu *et al.*, 2016). Hence it is not unlikely that dysregulation of SLC7A5, caused by loss of TXNIP, might affect brain development. To address this point, additional experiments are required.

It is clear that complete loss of TXNIP disrupts metabolic homeostasis on the cellular level, possibly resulting in the failure to coordinate AA and glucose uptake with oxidative phosphorylation across different organs, which leads to metabolic rigidity on the organismal level. Interestingly, the phenotypic spectrum in patients is variable in severity. Thus, other yet unknown genetic and/or non-genetic modifiers may either exacerbate or partly compensate loss of TXNIP function (e.g. diet, differences in medical treatment) and likely play a role in establishing disease phenotypes. These modifiers remain to be identified.'

Major concern 2. *The authors report that TXNIP interacts with HECT E3 ligases to regulate substrate degradation, yet this conclusion is drawn from overexpression-based immunoprecipitation studies, which do not confirm interaction under endogenous conditions. Without direct evidence of TXNIP-HECT E3 binding at native expression levels, this mechanistic link remains unresolved. Given that the authors have already generated antibody-validated TXNIP KO models, endogenous validation should be feasible if the interactions are not super-transient.*

Response: During the revision, we better characterized the interaction of TXNIP with HECT type ubiquitin ligases. Towards this goal, we performed quantitative interactome analysis with TXNIP and the respective PPxY mutants (new Fig. 4E, F, new Fig. EV4E, new Appendix table S2), used endogenous co-immunoprecipitation (new Fig. 4G) and established novel Bead-Immobilized Prey Assay (new Fig. EV4F, new Appendix Fig. S3B). The results of these new experiments extend and strengthen our original conclusion: TXNIP interacts with different HECT-type ubiquitin ligases in a PPxY motif dependent manner, with NEDD4L being the most abundant binding partner, followed by WWP2 and NEDD4. This suggests a ranked interaction profile, with possible redundancies, rather than a strict, highly specific hierarchy.

(1) NEDD4L is the most abundant ubiquitin ligase that interacts with TXNIP in a PPxY dependent manner (new Fig. 4E, F, G). Other HECT-type E3 ubiquitin ligases - such as WWP2, NEDD4, WWP1, ITCH, HECW1, and HECW2 - were also specifically co-immunoprecipitated with TXNIP in a PPxY motif dependent manner (new Fig. 4E, F). Yet, their abundance in the co-immunoprecipitation was significantly lower compared to NEDD4L:

WWP2 was ~8-fold less abundant, NEDD4 ~23-fold less abundant, and WWP1, ITCH, HECW1, and HECW2 were each at least ~35-fold less abundant (new Fig. 4E, F, EV4E). These differences cannot be simply attributed to difference in cellular abundance: ITCH is estimated to be the most abundant of these proteins with $6,5 \times 10^4$ molecules / cell, and NEDD4L ($3,7 \times 10^4$ molecules / cell), WWP2 ($1,8 \times 10^4$ molecules / cell), and NEDD4 ($1,3 \times 10^4$ molecules / cell) are within a similar 2-3 fold abundance range (Cho *et al*, 2022). The interaction of TXNIP with TXN (Thioredoxin) (Nishiyama *et al*, 1999), was independent of the PPxY motifs (new Fig. 4E, F, EV4E).

(2) Consistently, we also co-immunoprecipitated endogenous NEDD4L with ALFA-TXNIP from RPE1 TXNIP^{KO} cells (where ALFA-TXNIP was expressed at close to endogenous levels) (new Fig. 4G).

(3) Finally, we developed a 'Bead-Immobilized Prey Assay (BIPA)' to determine the interaction of TXNIP with the less abundant HECT type ubiquitin ligases (new Fig. EV4F, new Appendix Fig. S3B). We first immobilized equal amounts ALFA-tagged TXNIP, TXNIP^{PPxY331} or TXNIP^{PPxYmut} on ALFA-beads (Appendix Fig. S3B). These TXNIP beads were then incubated with cell lysates from HEK293T cells expressing YFP-tagged HECT type ubiquitin ligases. The binding of the YFP-tagged HECT-type ubiquitin ligases to TXNIP on the beads was analyzed by fluorescence microscopy and quantified (Fig. EV4F and Appendix Fig. S3B). This binding-assay confirmed that TXNIP has the capacity to interact with these HECT type ubiquitin ligases in a PPxY motif dependent manner. However, in contrast to its interaction with endogenous HECT-type ubiquitin ligases, when the ligases were overexpressed, binding was only significantly reduced when both PPxY motifs were mutated (Fig. EV4F and Appendix Fig. S3B).

Major concern 3. *What are the temporal dynamics of TXNIP-associated degradation, and is this process distinct from endosomal microautophagy (as reported in PMID: 30018090)? The authors present convincing, high-quality FACS-based data supporting TXNIP-mediated turnover. If this pathway is mechanistically separate from endosomal microautophagy, it suggests a hierarchy of degradation pathways leading to quiescence. Live cell imaging studies that define the temporal dynamics of this process using the tools the authors have created may reveal the relationship between these processes and refine the broader implications of TXNIP in homeostatic adaptation.*

Response: Thank you for this interesting suggestion. The downregulation of SLC7A5 was temporally not correlated with micro- and macro-autophagy (Mejlvang *et al*, 2018) and appeared to occur later on (Fig. EV1A). Most autophagy adaptors (NBR1, TAX1BP1 and NDP52) and LC3-I/II were degraded prior to SLC7A5 (new Fig. EV1A). Moreover, micro- and macro-autophagy appeared not be affected by loss of TXNIP (new Appendix Fig. S2A).

Minor comment 1. *In the discussion, the authors might briefly speculate on the implications of any functional redundancy that might exist between other arrestins.*

We provided the following statement in the discussion:

‘On the molecular level, a partial functional redundancy among the PPxY motif containing α -arrestins (ARRDC1-4, TXNIP) cannot be excluded. Clustal omega multisequence alignment of TXNIP with human ARRDC1 - 4 revealed between 23% - 44% identity on amino acid level. Moreover, these α -arrestins use their PPxY motifs to interact with a similar set of HECT-type ubiquitin ligases (Lee *et al.*, 2024; Rauch & Martin-Serrano). Therefore, it is not impossible that these α -arrestins are – at least in part – functionally redundant with TXNIP, perhaps even in a cell type specific manner. A similar argument for redundancy could be made for the HECT-type ubiquitin ligases. While NEDD4L was the most abundant interactor of TXNIP, also WWP2 and NEDD4 interacted with TXNIP in a PPxY motif specific manner. Hence different α -arrestins could interact with similar sets of HECT-type ubiquitin ligase thereby forming a robust network to fine-tune nutrient acquisition in a cell type and tissue specific responses to hormonal or metabolic cues.’

Dr. David Teis
Medical University of Innsbruck
Biocenter, Molecular Biochemistry
Innrain 80/82
Innsbruck 6020
Austria

26th Sep 2025

Re: EMBOJ-2025-120740R
TXNIP mediates LAT1/SLC7A5 endocytosis to limit amino acid uptake in cells entering quiescence

Dear David,

Thank you again for submitting your revised Review Commons manuscript, and my sincere apologies for the exceptional delay in its re-evaluation. I had been hoping to once more involve the original referee 2, as many of your revisions had been in response to the major comments this referee had initially raised. However, despite multiple reminders sent by our office, we have still not received his/her re-evaluation. In the meantime, the original referee 1 has checked your new version and responses to the original reports, and since they were fully satisfied (see comments below), I decided to now proceed towards acceptance of the article for The EMBO Journal, as soon as the following editorial and formatting issues have been addressed:

GENERAL:

- Please double-check to make sure to all relevant funding information in the manuscript is congruent with the info entered into our submission system. Currently missing in the submission system is 'Austrian Science Fund (FWF) 10.55776/P30196'
- Please provide suggestions for a short 'blurb' text prefacing and summing up the conceptual aspect of the study in two sentences (max. 250 characters), followed by 3-5 one-sentence 'bullet points' with brief factual statements of key results of the paper; they will form the basis of an editor-written 'Synopsis' accompanying the online version of the article. Please also upload a synopsis image, which can be used as a "visual title" for the synopsis section of your paper. The image should be in PNG or JPG format, and please make sure that it remains in the modest dimensions of (exactly) 550 pixels wide and 300-600 pixels high.

TEXT:

- Please adjust the order as well as the headers of the different manuscript sections: Title page with complete author information, Abstract, Keywords, Introduction, Results, Discussion, Methods, Data Availability, Acknowledgements, Disclosure and Competing Interests Statement, References, Main Figure Legends, Tables, Expanded Figure Legends.
- On the abstract page of the manuscript, please include 4-5 general keyword terms to enhance searchability.
- As we are switching from a free-text author contribution statement towards a more formal statement based on Contributor Role Taxonomy (CRediT) terms, please remove the present Author Contribution section and instead specify each author's contribution(s) directly in the Author Information page of our submission system during upload of the final manuscript. See <https://casrai.org/credit/> for more information.
- Please rename the Competing Interest section into "Disclosure and Competing Interests Statement", in accordance with our updated Guide to Authors (<https://www.embopress.org/competing-interests>)
- Please include a dedicated "Data Availability" section at the end of the Material and Methods; should there no data deposition to public repositories linked to the study, this should still be stated as "This study includes no data deposited in external repositories."
- Please remove the 'Reagents and Tools' table from the main manuscript text, it is sufficient for production that it is uploaded as a separate file.
- Please carefully go through the reference list and make sure that each reference is complete with citation year, journal name, volume/page/locator numbers - some of these informations are currently missing in several entries.

- Please adjust the headers of the Expanded View figures in their legends to "Figure EV1/2/3...", and make sure main and EV figure legends are included at the end of the main manuscript file

DATA:

- Please carefully revise the Appendix PDF according to our author guidelines (www.embopress.org/page/journal/14602075/authorguide#expandedview) - it should start with a title page stating "Appendix for [ms title]" and containing a brief table of contents with respective page numbers. Its contents should be named and called-out as "Appendix Figure S1/2/3..." and "Appendix Table S1/2/3..." throughout (also in the main text, where some of them are still referred to as "Supplementary"). Furthermore, please sort the Appendix contents so that all figures are following one after the other, and also all tables are grouped together. For the outlying section on the patient's medical history, I would suggest giving it a specific name such as "Appendix Patient Information", which should be referred to in the main text at least once.

- During routine pre-acceptance checks, our data editors have raised the following queries regarding figures, data, and legends, which I would ask you to address (ideally using the Track Changes option to facilitate our checking):

1. Please note that the legend for figure 1 is not provided in a sequential (alphabetical) manner. This needs to be rectified.
2. Please note that the exact p values need to be provided in the legends of figures 1A, 2D, 3D, 5D; EV1 C, F
3. Please indicate the statistical test used for data analysis in the legend of figure 4F
4. Please note that the box plots need to be defined in terms of minima, maxima, centre, bounds of box and whiskers, and percentile in the legends of figures 1A, B, E, I; 2B, D; 3D, 5D, 6A, B, D, E, F, G; EV1 C, F, H; EV2 F, G, J, I; EV3 B, C; EV4 C, F
5. Please note that information related to n is missing in the legend of figure 4F
6. Please note that the error bars are not defined in the legends of figures 4E, 7A, EV1 A, D, G; EV2 K; EV3 A; EV4 B, EV5 A
7. Please note that the measure of center for the error bars needs to be defined in the legend of figure EV3 D
8. Please note that the scale bar needs to be defined for figure EV5 B

- Finally, regarding provision of the requested source data, please make sure that the larger files can be deposited in external repositories, and provide the respective accession codes and database URLs both in the manuscript's data availability section and in the source data checklist.

Should you need additional guidance/feedback regarding this final adjustments, please do not hesitate to contact us directly. Thank you again for the opportunity to consider this work for The EMBO Journal, and I look forward to receiving your final version!

With kind regards,

Hartmut

Revision to The EMBO Journal should be submitted online within 90 days, unless an extension has been requested and approved by the editor; please click on the link below to submit the revision online before 25th Dec 2025:

Link Not Available

Referee #1:

I thank the authors for their thorough response to my initial review. They have addressed all of my comments, and the added quantifications make the data even more convincing. As previously stated, I believe that the study is well-executed, the conclusions are supported by the data, and the findings represent an advance in the fields of quiescence, metabolism, and cell biology. I am satisfied with the revisions and recommend the manuscript for publication.

Rev_Com_number: RC-2024-02821
New_manu_number: EMBOJ-2025-120740R
Corr_author: Teis
Title: TXNIP mediates LAT1/SLC7A5 endocytosis to limit amino acid uptake in cells entering quiescence

During the revision we have addressed the following editorial and formatting issues. In addition, we identified and corrected two minor errors in Figure 1b and Figure 6c.

GENERAL:

- Please double-check to make sure to all relevant funding information in the manuscript is congruent with the info entered into our submission system. Currently missing in the submission system is 'Austrian Science Fund (FWF) 10.55776/P30196'

We have added this information in the submission system.

- Please provide suggestions for a short 'blurb' text prefacing and summing up the conceptual aspect of the study in two sentences (max. 250 characters), followed by 3-5 one-sentence 'bullet points' with brief factual statements of key results of the paper; they will form the basis of an editor-written 'Synopsis' accompanying the online version of the article. Please also upload a synopsis image, which can be used as a "visual title" for the synopsis section of your paper. The image should be in PNG or JPG format, and please make sure that it remains in the modest dimensions of (exactly) 550 pixels wide and 300-600 pixels high.

We provide a 'blur' text and a synopsis image.

TEXT:

- Please adjust the order as well as the headers of the different manuscript sections: Title page with complete author information, Abstract, Keywords, Introduction, Results, Discussion, Methods, Data Availability, Acknowledgements, Disclosure and Competing Interests Statement, References, Main Figure Legends, Tables, Expanded Figure Legends.

We have adjusted the order as well as the headers of the different manuscript sections.

- On the abstract page of the manuscript, please include 4-5 general keyword terms to enhance searchability.

On the abstract page of the manuscript, we have included 5 general keywords.

- As we are switching from a free-text author contribution statement towards a more formal statement based on Contributor Role Taxonomy (CRediT) terms, please remove the present Author Contribution section and instead specify each author's contribution(s) directly in the Author Information page of our submission system during upload of the final manuscript. See <https://casrai.org/credit/> for more information.

We have used the submission system to provide author contributions.

- Please rename the Competing Interest section into "Disclosure and Competing Interests Statement", in accordance with our updated Guide to Authors (<https://www.embopress.org/competing-interests>)

We have renamed the section accordingly.

- Please include a dedicated "Data Availability" section at the end of the Material and Methods; should there no data deposition to public repositories linked to the study, this should still be stated as "This study includes no data deposited in external repositories."

We have included the data availability section: the confocal images (15Gb) are available for download on figshare (<https://figshare.com/s/8a55f66a3a17781bcdfd?file=56522243>), and the proteome data was deposited in Pride.

- Please remove the 'Reagents and Tools' table from the main manuscript text, it is sufficient for production that it is uploaded as a separate file.

We remove the 'Reagents and Tools' table from the main manuscript and uploaded it as a separate file.

- Please carefully go through the reference list and make sure that each reference is complete with citation year, journal name, volume/page/locator numbers - some of these informations are currently missing in several entries.

We have identified and removed 3 duplicated references and ensured that all references are complete.

- Please adjust the headers of the Expanded View figures in their legends to "Figure EV1/2/3...", and make sure main and EV figure legends are included at the end of the main manuscript file.

Done.

DATA:

- Please carefully revise the Appendix PDF according to our author guidelines (www.embopress.org/page/journal/14602075/authorguide#expandedview) - it should start with a title page stating "Appendix for [ms title]" and containing a brief table of contents with respective page numbers. Its contents should be named and called-out as "Appendix Figure S1/2/3..." and "Appendix Table S1/2/3..." throughout (also in the main text, where some of them are still referred to as "Supplementary"). Furthermore, please sort the Appendix contents so that all figures are following one after the other, and also all tables are grouped together. For the outlying section on the patient's medical history, I would suggest giving it a specific

name such as "Appendix Patient Information", which should be referred to in the main text at least once.

The Appendix file has been adjusted accordingly.

- During routine pre-acceptance checks, our data editors have raised the following queries regarding figures, data, and legends, which I would ask you to address (ideally using the Track Changes option to facilitate our checking):

1. Please note that the legend for figure 1 is not provided in a sequential (alphabetical) manner. This needs to be rectified.

The order of figure legend 1 was changed from A-J. The changes are highlighted in red.

2. Please note that the exact p values need to be provided in the legends of figures 1A, 2D, 3D, 5D; EV1 C, F

The exact *p*-values were provided in the figure legends. The changes are highlighted in red.

3. Please indicate the statistical test used for data analysis in the legend of figure 4F

The adjusted *p*-values were calculated using the Benjamin-Hochberg method. The changes are highlighted in red.

4. Please note that the box plots need to be defined in terms of minima, maxima, centre, bounds of box and whiskers, and percentile in the legends of figures 1A, B, E, I; 2B, D; 3D, 5D, 6A, B, D, E, F, G; EV1 C, F, H; EV2 F, G, J, I; EV3 B, C; EV4 C, F

The indicated box blots were defined in terms of minima, maxima, centre, bounds of box and whiskers and percentile. Please note that Figure 6G; EV2G are bar charts, showing the respective mean values. We have noted this in the figure legends accordingly. The changes are highlighted in red.

5. Please note that information related to n is missing in the legend of figure 4F -

n=3 was specified. The changes are highlighted in red.

6. Please note that the error bars are not defined in the legends of figures 4E, 7A, EV1 A, D, G; EV2 K; EV3 A; EV4 B, EV5 A

We have defined the error bars are in corresponding figure legends. The changes are highlighted in red.

7. Please note that the measure of center for the error bars needs to be defined in the legend of figure EV3 D

Centre for the error bars was indicated as mean. The changes are highlighted in red.

8. Please note that the scale bar needs to be defined for figure EV5 B

The scale bar in figure EV5B was defined as 400 μ m.

- Finally, regarding provision of the requested source data, please make sure that the larger files can be deposited in external repositories, and provide the respective accession codes and database URLs both in the manuscript's data availability section and in the source data checklist.

We have provided all source data as zip files. The very large imaging data and the proteome data have been uploaded to figshare and pride, respectively.

In addition to the requested changes, we identified and corrected two minor errors in **Figure 1b** and **Figure 6c**.

In **Figure 1b**, a *p*-value was missing; this has now been added.

In **Figure 6c**, we noticed that the loading control for S6 contained the wrong lanes. During figure assembly, the incorrect lanes (3–6) were inadvertently cropped from the correct film instead of the intended lanes (1–4), which were used for quantification (please see figure below). This has now been corrected. The correct source data were provided in the original submission.